# Classically studied coherent structures only paint a partial picture of wall-bounded turbulence

Andrés Cremades [1,2] ✉, Sergio Hoyas [1] & Ricardo Vinuesa [2,3] ✉

For the last 140 years, the mechanisms of transport and dissipation of energy in a turbulent flow have not been completely understood. Previous research has focused on analyzing the so-called coherent structures, organized flow patterns characterized by their spatial coherence, lifespan and significant contribution to momentum and energy transfer. However, the connection between these structures and the flow development is still uncertain. Here, we show a data-driven methodology for objectively identifying high-importance regions. A deep-learning model is trained to predict a future state of the flow and the gradient-SHAP explainability algorithm is used to calculate the importance of each grid point. Finally, high-importance regions are computed using the SHAP data and are compared to the other coherent structures. The SHAP analysis provides an objective way to identify the regions of higher importance, which exhibit different levels of agreement with the classical structures without being completely related to any particular one.

Although turbulent flows have been widely studied for over 140 years[1–4], the nature of turbulence and its energy-transfer mechanisms remain unclear for most of the relevant industrial and scientific flows[5–9]. A complete understanding of turbulence may have worldwide implications, since 15% of the energy consumed worldwide is spent near the surface of vehicles due to turbulent effects[10].

Turbulent-flow dynamics are governed by the Navier–Stokes equations[11,12], a set of non-linear partial differential equations, whose analytical solution for a general flow is probably impossible to obtain. The intricate behavior of these complex fluid flows is due to the multi-scale nature of turbulence[3,13], which also imposes very high computational demands to simulate and study these flows[14]. The technique that numerically integrates the Navier–Stokes equations without any modeling is known as direct numerical simulation, DNS. Unfortunately, DNSs of practical situations are still decades away. It can be estimated that the random-access memory (RAM) needed to simulate a commercial jet is equivalent to that of a month of the internet.

As solving the governing equations is out of reach, many different techniques have appeared during these 140 years[15]. One of the most promising ideas is to understand, model, and finally control turbulence by studying the nonlinear interactions among the different coherent structures of the flow[16]. Historically, these coherent structures were first described experimentally as streamwise streaks[17] and Reynolds-stress events[18]. These two kinds of structures were defined in terms of the transport of kinetic energy and momentum. Later, in 1990[19], a third class was proposed: vortices, which are defined by regions where vorticity was larger than shear. In all cases, the definitions of these structures are based on different physical effects. While the role of these structures in wall-bounded turbulence has been thoroughly documented, the literature lacks a completely objective assessment of the actual importance of these structures at different wall-normal locations.

Once the idea of coherent structure is clear, a second step is to analyze the effects of these structures in the flow. For instance, in the work by Osawa and Jiménez[20] the causal relevance of the flow conditions in wall-bounded turbulence is analyzed by perturbing the fluid and monitoring the effect on its evolution. This methodology recomputes the simulations, modifying the perturbed subdomain and linking the different regions with their impact on the flow evolution.

[1]Instituto Universitario de Matemática Pura y Aplicada, Universitat Politécnica de Valéncia, Valencia, Spain. [2]FLOW, KTH Royal Institute of Technology, Stockholm, Sweden. [3]Department of Aerospace Engineering, University of Michigan, Ann Arbor MI, USA. ✉e-mail: andrescb@kth.se; rvinuesa@umich.edu

Lozano-Durán and Arranz[21] proposed an approach different from that of causality with interventions, based on information theory[22]. This approach, which has been applied to different turbulent flows[23–25], uses the information flux among the flow variables to quantify causality. This methodology avoids the recalculation of the simulation, as it is based on time series that do preserve spatial information.

The present work addresses two different points. On the one hand, the idea is to establish an index that allows the classification of the importance of the different flow regions in the flow evolution. On the other hand, the method should identify the most important regions of the flow independently of any previous definition of turbulent structures.

In a previous study, the intense Reynolds-stress structures were ranked depending on their impact on the evolution of a turbulent channel[26]. This work extends the previous results by calculating the importance of every single grid point in the flow in order to percolate that importance and define new importance-based structures by calculating the Shapley additive explanations (SHAP-based structures). For this purpose, an attribution method[27] based on the extension of the Shapley values[28] to an infinity-player game is required: the so-called gradient SHAP[29]. This algorithm implements the expected-gradients (EG)[30] methodology, a more computationally efficient version of the integrated gradients (IG) proposed by Sundararajan et al.[31].

The methodology employed in this work is summarized in Fig. 1, and can be divided into three main stages. The first one is the development of a surrogate model. To this end, a U-net[32] deep-learning model was trained. The model is trained for a database of 10000

instantaneous flow fields until the relative error of the predictions is approximately 1% of the maximum value for the whole database. The second stage is the use of the previous model for calculating the importance of each input grid point in the prediction of the flow using the gradient-SHAP algorithm. In this sense, the model had to be modified to generate a single output (mean-squared error), which evaluates the accuracy of the prediction. Finally, once the SHAP values are obtained, they are used for defining the high-importance structures (SHAP structures), whose physical properties are evaluated.

## Results

Based on the gradient-SHAP methodology we determine an importance score for each grid point of two turbulent channels, the first one with a friction Reynolds number $Re_\tau = u_\tau h/\nu = 125$ and the second one with $Re_\tau = 550$. These SHAP values identify the most influential points in the causal relationships detected by the deep-learning model. For this reason, the physical meaning of the SHAP values is conditioned by the output of the model. The SHAP values answer the question formulated mathematically for the deep-learning model. In the case presented in this work, the model calculates the mean-squared error of the reconstruction of the flow. Therefore, the SHAP values identify the most important regions of the flow for the evolution of the velocity fluctuation field. The scores evaluate the influence of the grid points in the prediction of the state of the flow after a time interval $\Delta t^+ = \Delta t\, u_\tau^2/\nu = 5$, where $h$ is half height of the channel, $\nu$ is the fluid kinematic viscosity, $u_\tau = \sqrt{\tau_w/\rho}$ is the friction velocity ($\tau_w$ is the wall-shear stress and $\rho$ the fluid density) and $\Delta t$ the time interval between fields[15]. The friction

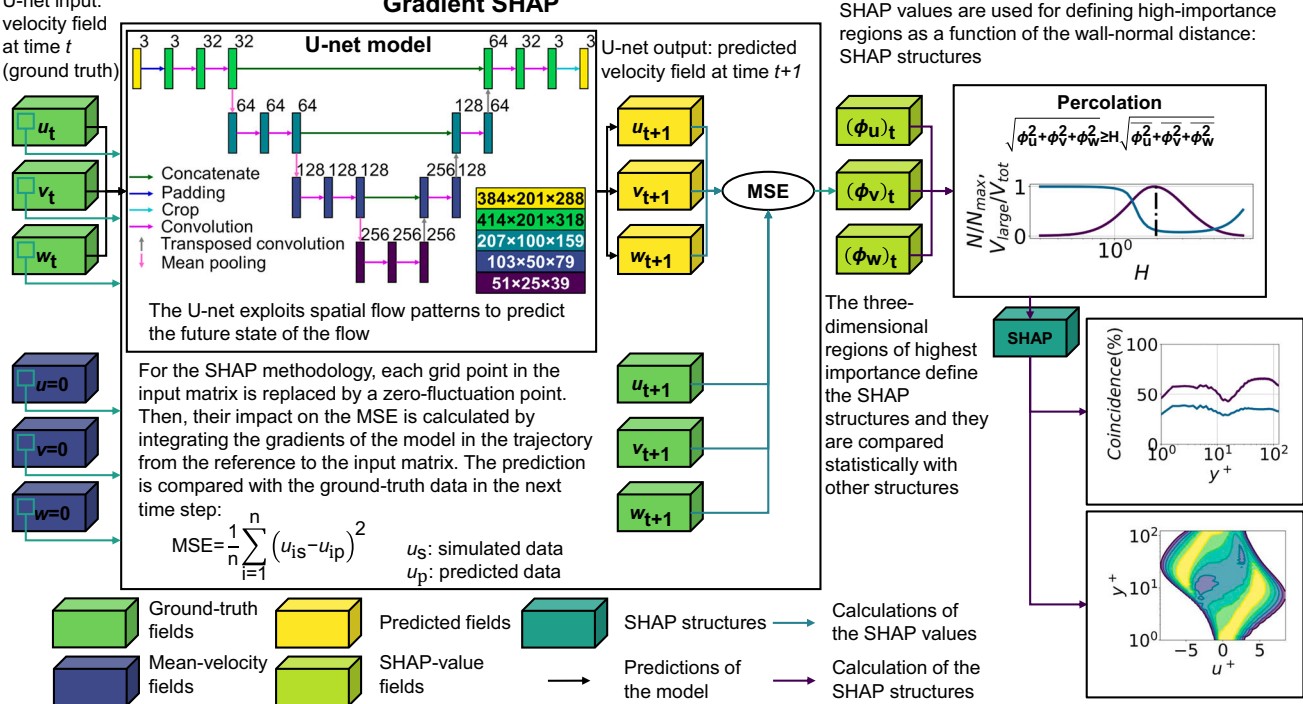

**Fig. 1 | Diagram showing the methodology of the paper.** The methodology is divided into three main stages. The first stage, represented by the black lines, comprises a U-net deep learning (DL) model for predicting the evolution of the flow. The three components of the simulated velocity fluctuation (green blocks, $u_t$ for the streamwise, $v_t$ for the wall-normal and $w_t$ for the spanwise velocity) in an instant $t$ are evolved through the U-net to the next time step $t + 1$ separated a viscous time $t^+ = 5$ (yellow blocks, $u_{t+1}$, $v_{t+1}$ and $w_{t+1}$). In the second stage (blue flow connectors), the importance of each grid node of the field, for each velocity component, is calculated by applying the expected-gradients algorithm an extension of Shapley additive explanations (SHAP values) to an infinite-player game. For evaluating the importance of each input grid point in the prediction, the mean-

squared error of the predicted field is used and the simulated fields in the next step (green blocks, $u_{t+1}$, $v_{t+1}$ and $w_{t+1}$) is defined as the output. The result of this expectation is the so-called SHAP values, which are represented by the light-green blocks, and represent the importance of the streamwise $((\phi_u)_t)$, wall-normal $((\phi_v)_t)$ and spanwise $((\phi_w)_t)$ velocity fluctuations. In the final stage (purple flow connectors), the previously mentioned SHAP values are used to segment the model through a percolation analysis. As a result, the flow field can be segmented into SHAP structures. Once these structures are defined, their physical characteristics are analyzed, and their correlation with other coherent structures is evaluated for the different wall-normal distance $y^+$.

Reynolds number sets the streamwise pressure gradient driving the flow, and thus, it is its main control parameter. The instantaneous velocity vector is represented by $\boldsymbol{U}$ with components $U$, $V$, and $W$ in their streamwise ($x$), wall-normal ($y$), and spanwise ($z$) components. The Reynolds decomposition is applied to the velocity, $U = \overline{U} + u$, where the statistically averaged quantities in $x$, $z$, and $t$ are denoted by an overbar, and the fluctuating quantities are denoted by lowercase letters. Additionally, the root mean square of the fluctuations is expressed with $(\cdot)'$, and the magnitudes in viscous units are denoted by the superscript $+$. Further information on the numerical approach and the simulation characteristics are provided in the Methods section.

Once the gradient-SHAP methodology is applied to the problem, a vector field defining the importance of each region of the space in the evolution of the flow, namely the SHAP values $\phi$, is created. The SHAP values quantify the causal importance of each individual grid point in the evolution of the flow (refer to the supplementary material for a detailed discussion and a comparison with the causal system examples presented by Martínez-Sánchez et al.[33]). As previously mentioned, due to the evolution of the model, the SHAP values determine which grid points are the most influential for the evolution of the flow. Accordingly, the SHAP structures identify the regions of the flow that should be targeted to effectively control its evolution[34]. The SHAP vector provides the importance of the streamwise, wall-normal, and spanwise velocities, $\phi_{\mathrm{u}}$, $\phi_{\mathrm{v}}$, and $\phi_{\mathrm{w}}$, respectively. To establish a compact definition of what is considered a high-importance region, the combined effect of the three components of the vector is taken into account by comparing the absolute value of the vector against the square root of the sum of the square mean of each component as a function of the wall-normal distance:

$$\sqrt{\phi_u^2(x,y,z,t) + \phi_v^2(x,y,z,t) + \phi_w^2(x,y,z,t)} > H\sqrt{\overline{\phi_u^2}(y) + \overline{\phi_v^2}(y) + \overline{\phi_w^2}(y)}, \tag{1}$$

where $H$ is a percolation index set to a value of 2 to maximize the total number of structures[35], as can be observed in the percolation block of Fig. 1. A visual representation showing the SHAP-based structures, colored according to the wall-normal distance, is presented in subfigures a) and b) of Fig. 2.

The definition of the SHAP-based structures is compared against some of the classical definitions of coherent structures widely used in the turbulence community. The first structures to compare with are the intense Reynolds-stress structures or Q events[36]. The structures are presented in subfigures c) and d) of Fig. 2 and are defined as:

$$|u(x,y,z,t) v(x,y,z,t)| > \beta u'(y) v'(y), \tag{2}$$

where $\beta$ is the percolation index to maximize the number of structures. These coherent structures are associated with high momentum transfer and turbulent-kinetic-energy (TKE) production. The second coherent structures are the streamwise streaks[17] of high ($u > 0$) and low ($u < 0$) streamwise velocity:

$$\sqrt{u^2(x,y,z,t) + w^2(x,y,z,t)} > \alpha u_\tau, \tag{3}$$

where $\alpha$ is the percolation parameter. The streamwise streaks are shown in subfigures e) and f) of Fig. 2. The low-velocity streaks result from the lift-up of the low-speed flow near the wall to the higher-velocity regions and they are complemented by the high-speed streaks, which redistribute the energy within the flow. These are regions of high turbulent kinetic energy. Finally, the SHAP-based structures are also compared with the vortices defined by Chong et al.[19]. The definition of these structures is more complex and is presented in the Methods section. In general, the vortices define those regions of the flow where rotation is larger than shear. Vortices appear

close to the ejections[37] and contribute to the near-wall cycle to sustain turbulence involving the streaks. In fact, vortices are a result of the instabilities of the streaks and can be considered dissipative as they carry high levels of enstrophy. The visualization of the vortices is provided in subfigures g) and h) of Fig. 2.

After defining the various coherent structures, their physical properties are analyzed to link the new SHAP-based structures with the Q events, streaks and vortices. Fig. 3 represents the joint probability density function (JPDF) of the inner-scaled streamwise velocity fluctuation and the wall-normal distance of the four different coherent structures. A similar analysis is performed for the wall-normal and the spanwise velocities, and is shown in the Supplementary Material.

Figure 3 shows that the SHAP-based structures exhibit several regions of high importance for both friction Reynolds numbers. The area of positive streamwise fluctuations in the viscous sublayer is in good agreement with the corresponding region of the Q events, which is associated with sweep structures or high-velocity structures moving towards the wall. As presented in Fig. 3, the high-importance high-velocity region close to the wall is similar for both friction Reynolds numbers, despite the higher separation of scales of $Re_\tau = 550$ with respect to $Re_\tau = 125$. In fact, our results show that the sweep events close to the wall constitute high-importance regions independently of the friction Reynolds number, as they transport the high-energy regions towards the wall, where the shear stress is stronger. Another area of importance is observed from $y^+ \approx 30$ to the center of the channel, and this is also in good agreement with a region of high probability density of Q events, corresponding to low-velocity regions moving away from the wall, i.e. ejections. Although the high-importance regions are similar for SHAP structures and Q events at $Re_\tau = 125$, larger differences are observed for $Re_\tau = 550$. While the SHAP structures only detect a region of high importance for low-velocity events, the Q events increase the probability of sweeps. Thus, the SHAP structures can identify the Q events with higher impact for the evolution of the flow, or in other words, the most causal Q events. Note that sweeps and ejections have been widely studied in the literature as very important structures, and therefore the agreement, as well as the differences, with SHAP structures are a very important result. In fact, in a previous study, the ejections and the sweeps were demonstrated to exhibit a similar importance per unit of volume[26] and in the works of Jiménez[16] and Lozano-Durán et al.[36], the ejections and sweeps were highlighted as the most influential Q events. Furthermore, our results show a remarkable agreement between the wall-attached sweeps and the wall-detached ejections with the SHAP structures with positive and negative streamwise velocity fluctuations, respectively. This result is also consistent with the recent research of Osawa and Jiménez[20], which reinforces the idea of the sweeps being the most influential structure in the flow field as it transports the flow close to the wall, where the local shear is higher. At $y^+ \simeq 15$ a strong agreement is observed between the SHAP structures and the streaks for both the positive and the negative streamwise fluctuations. The streaks are streamwise-elongated structures that play an important role in the turbulent flow by redistributing the energy between the low-velocity and the high-velocity regions. Therefore, the regions involving energy transfer between the wall-attached and wall-detached regions have been identified as SHAP structures with high importance for both friction Reynolds numbers. In Fig. 3, a strong relationship between the high-importance regions and the regions of momentum transfer can be observed, as the Q events and streaks exhibit a stronger agreement with the SHAP structures, while the dissipative structures, such as the vortices, show lower agreement. Nevertheless, for $y^+ > 50$ the region of high importance for zero streamwise velocity fluctuation $u \approx 0$ is explained by the presence of vortices and high-dissipation regions. However, although the SHAP structures exhibit high similarities with the previous structures in different regions of the channel, the distributions do not perfectly match any of them. Therefore, SHAP values detect complex

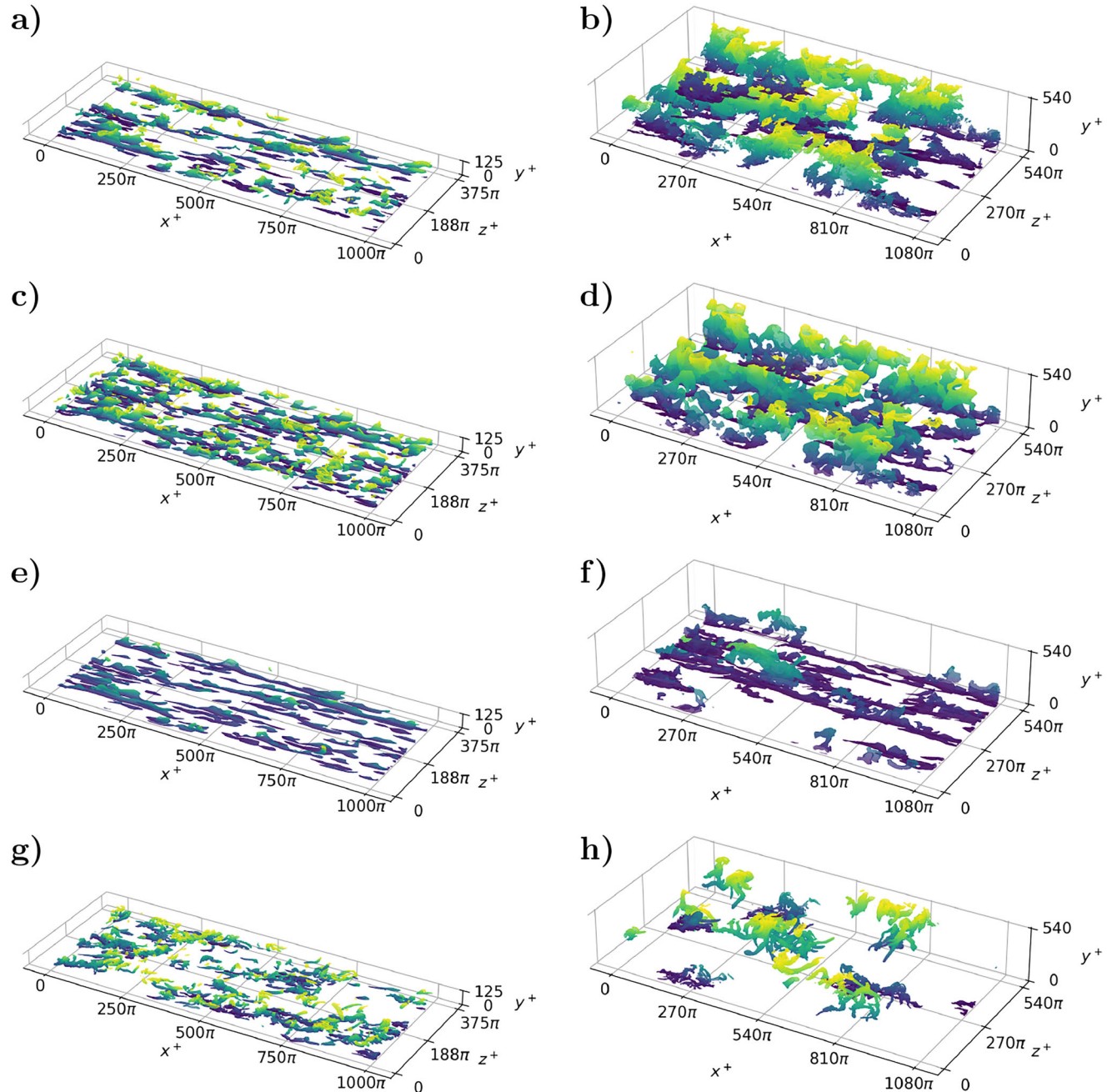

**Fig. 2 | Instantaneous visualization of the various coherent structures in the channel flow.** The figure shows four types of coherent structures (SHAP-based structures, subfigures (**a**) and (**b**), Reynolds-stress structures or Q events, subfigures (**c**) and (**d**), streaks, subfigures (**e**) and (**f**) and vortices, subfigures (**g**) and (**h**) in half of the channel colored by wall-normal distance (purple near the wall and yellow in the mid-plane of the channel) for $Re_\tau = 125$, subfigures (**a**), (**c**), (**e**) and (**g**), and $Re_\tau = 550$, subfigures (**b**), (**d**), (**f**) and (**h**). In the figure, the wall is located at $y^+ = 0$ and the mid-plane of the channel is $y^+ = 125$ or $y^+ = 550$ depending on the case. In addition, we use periodic boundary conditions in $x$ and $z$ for both cases. Note that the same flow field is used for the four panels of each channel. Note that $x^+$, $y^+$ and $z^+$ define the streamwise, spanwise and wall-normal directions respectively.

nonlinear patterns that were not identified by the other definitions and approaches, as could be clearly observed with the Q events far from the wall for $Re_\tau = 550$. For this reason, it is of high importance to analyze the percentage of agreement between the different structures as a function of the wall-normal distance to generate relationships between the properties of the coherent structures and their implications in the evolution of the flow.

The coincidence of pairs of the various structures along the wall-normal direction is presented in Fig. 4. This figure shows a roughly constant agreement between the SHAP structures and the intense Reynolds-stress structures (Qs). This co-occurrence

represents around 60% of the volume of the SHAP structures, and around 40% of the volume of the intense Q events for $Re_\tau = 125$ and around 50% and 30% respectively for $Re_\tau = 550$. This coexistence can be mainly observed in the blue and green regions inside the black contours for the various wall-normal locations shown in Fig. 5. Thus, a significant fraction of the SHAP structures coincides with Q events throughout the channel for both frictions Reynolds numbers, but specially for $Re_\tau = 125$. However, there is a non-negligible percentage, 40% and 50% for each friction Reynolds number respectively of the SHAP structures, which is not correlated with the Reynolds stress. This difference is due to two

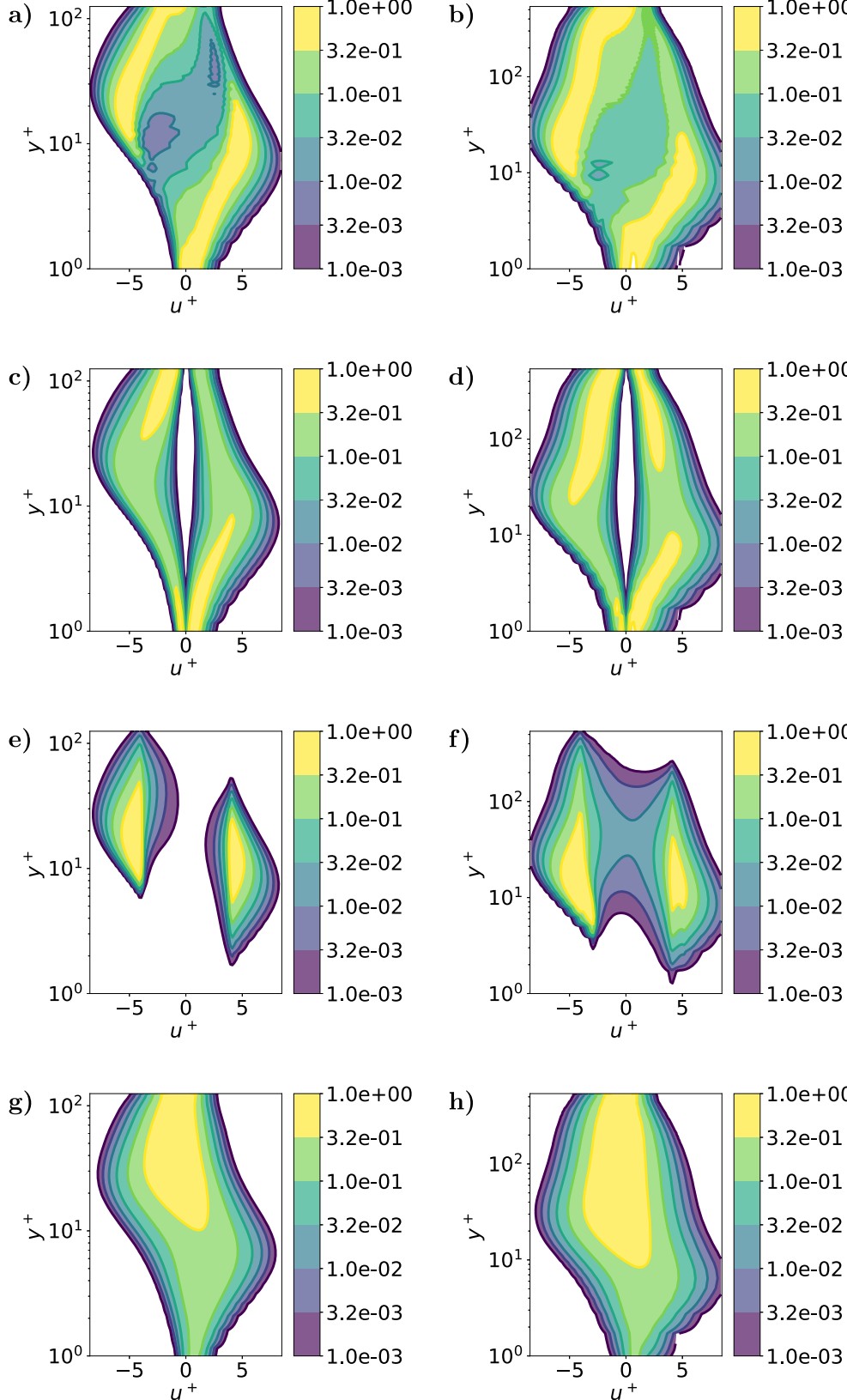

**Fig. 3 | Joint probability density function of the streamwise velocity fluctuation, $u^+$, and the inner-scaled wall-normal distance, $y^+$, for the different types of structures.** The figure presents the SHAP-based structures, subfigures (**a**) and (**b**), Reynolds-stress structures, subfigures (**c**) and (**d**), streaks, subfigures (**e**) and (**f**) and vortices, subfigures (**g**) and (**h**) for $Re_\tau = 125$, subfigures (**a**), (**c**), (**e**) and (**g**), and $Re_\tau = 550$, subfigures (**b**), (**d**), (**f**) and (**h**).

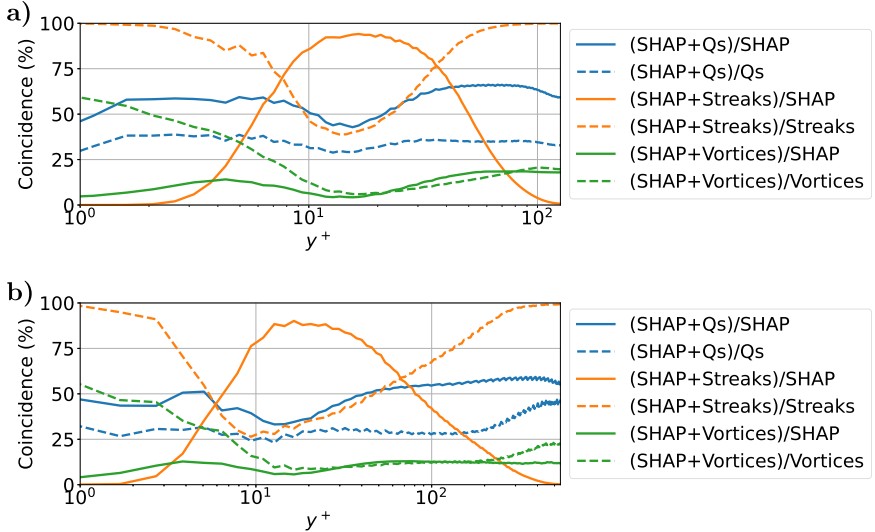

**Fig. 4 | Coincidence between the various types of structures.** Percentage of coincidence of pairs of the following structures: SHAP, Q events, streaks and vortices, relative to the volume of each type of the pair for $Re_\tau = 125$, (**a**), and $Re_\tau = 550$, (**b**), along wall-normal distance $y^+$.

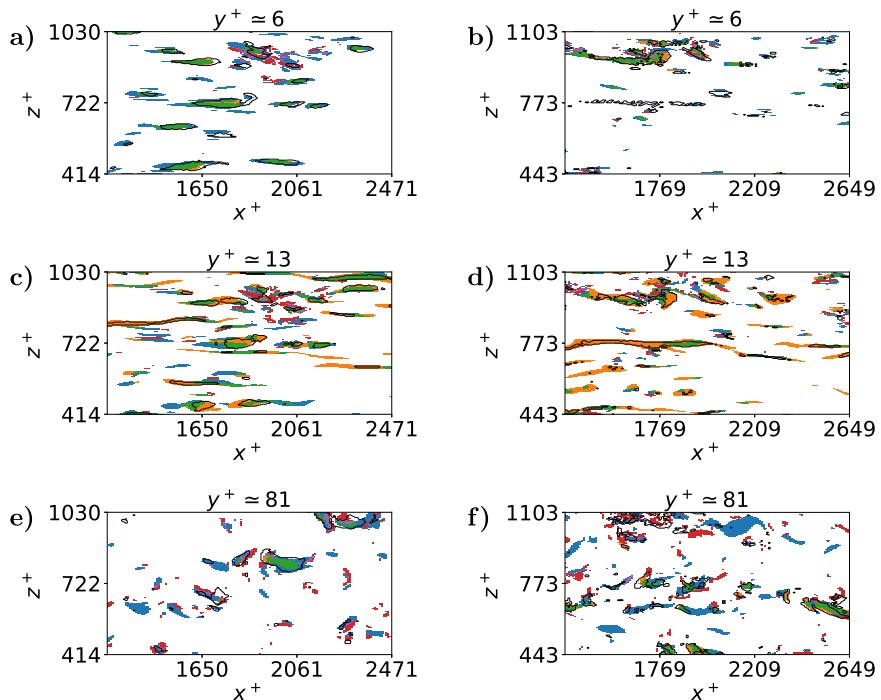

**Fig. 5 | Instantaneous coincidence between the various types of structures at various wall-normal distances, $y^+$.** We show results for $Re_\tau = 125$, subfigures (**a**), (**c**) and (**e**) and $Re_\tau = 550$, subfigures (**b**), (**d**) and (**f**). The colors used for the coincidence between structures follow this code: ▇ Qs\(streaks ∪ vortices), ▇ streaks \(Qs ∪ vortices), ▇ (Qs ∪ streaks)\vortices, ▇ vortices \(Qs ∪ streaks), ▇ (Qs ∪ vortices) \streaks, ▇ (streaks ∪ vortices) \Qs, ▇ Qs ∪ streaks ∪ vortices. In these panels, the SHAP structures are represented by the black solid lines. This figure presents a zoomed-in view of a small region on the three planes, in the streamwise, $x^+$, and spanwise, $z^+$, directions. For a representation of the whole planes the reader is referred to the Supplementary Material.

main reasons: on the one hand, the Q events are strongly related to the wall-normal velocity, while the SHAP structures generally give more importance to the streamwise component (the reader is referred to the Supplementary Material for a deeper analysis). On the other hand, the Q events do not take into account the spanwise fluctuations. Regarding the comparison between SHAP and streaks, for $10 \lesssim y^+ \lesssim 50$ the SHAP structures are contained inside the streaks, with almost 90% agreement in both cases, as can be observed for $y^+ \simeq 13$, where the black contours of the SHAP structures remain inside the large orange or green streaks.

Finally, the agreement between SHAP and vortices is lower and remains below 20% in both cases. Moreover, part of this coincidence is shared with the streaks and the Q events: as indicated by the purple, brown and gray colors inside the black contours. Although the regions containing only vortices are mostly outside the SHAP structures, a fraction of them are detected as highly influential for the evolution of the flow. This explains the agreement between the joint probability density functions of Fig. 3 for SHAP and vortices when $u \simeq 0$ close to the channel center. Therefore, the SHAP regions represent a small fraction of the

vortices, and, although the SHAP values are associated with some high rotation areas, this relationship is not direct.

## Discussion

In the present manuscript, we have presented the capabilities of explainable artificial intelligence (XAI) to estimate the importance of each grid point in a turbulent channel up to a friction Reynolds number $Re_\tau = 550$ for the prediction of its future states. Then, new coherent structures are defined based on these importance scores, namely the SHAP values. The SHAP-based structures have been analyzed from a physical point of view by comparing them against other coherent structures widely studied in the turbulence literature: intense Reynolds-stress structures (Q events)[36], streaks[17] and vortices[19]. Near the wall, a strong correlation between the SHAP structures and the sweeps can be observed, especially in the regions where the sweeps collide with high-velocity streaks. Then, for a wall-normal distance $y^+ \approx 15$, the streaks increase their presence in the flow, agreeing with the SHAP structures. Finally, farther from the wall, the presence of the streaks is lower and the SHAP structures mainly correlate with the ejections, although for $Re_\tau = 550$ the separation of scales is higher and sweeps are also present. Nevertheless, although the vortices increase their agreement with the SHAP structures near the mid-plane of the channel, their importance is still lower than that of the Q events.

For the past 140 years, the wall-bounded-turbulence analysis has focused on the assessment of partial results in certain regions of the flow. The SHAP structures are the first method capable of addressing this question globally and objectively, identifying the regions of highest importance without any preconceived bias. The SHAP values are used to identify these high-importance regions by analyzing the most sensitive input features for a mathematically defined question: which regions minimize the error of the model predictions? Although the SHAP structures partly correlate with the classically studied ones in different regions of the channel, this agreement is not perfect. Since the SHAP structures objectively identify the most important regions of the flow, analyzing the differences between SHAP and other structures can provide an excellent venue to deepen our insight into wall-bounded turbulence, uncovering the most important causal relations in the system. An example of application of the present SHAP framework to another flow case, namely the flow around a wall-mounted square obstacle[23,38], can be found in the Supplementary Material. In this case, the SHAP values highlight the influence of the wall and the obstacle, reducing the importance of the streaks, identifying a similar importance of the Q events and a low importance of the vortices. The definition of causally important regions is a change of paradigm in the analysis of physical problems, and can be replicated in many other fields such as thermal fields or cavitation. Furthermore, it can help to develop much more efficient control strategies[34]. In addition, the methodology has been proved to be scalable with the friction Reynolds number, producing results directly related to the wall-normal distance independently of the Reynolds number. It is also important to highlight that in this work the importance is defined based on a surrogate model which predicts a state of the flow 5 viscous times into the future. The Supplementary Material also presents results for a time interval $\Delta t^+ = 10$, showing high levels of agreement with the analysis for $\Delta t^+ = 5$. Other possible models can be developed, e.g. for predictions with much longer time horizons (characteristic of the outer region) or of the wall-shear stress (where different SHAP structures may be identified). In fact, the SHAP structures for the evolution of the vorticity are presented in the Supplementary Material. The different questions that can be formulated with various models will be addressed in future studies.

## Methods

### Numerical simulation of the turbulent channel

The velocity fields of turbulent channel flow used in this study were calculated by numerically integrating the Navier–Stokes equations using the LISO code. This direct-numerical-simulation (DNS) code has successfully been used for solving some of the largest simulations in the field of wall-bounded turbulence[14,39]. The integration strategy of LISO is based on the spectral method proposed by Kim et al.[40] using a seven-point compact-finite-difference scheme in the wall-normal direction with fourth-order consistency and extended spectral-like resolution[41]. Concerning the temporal discretization, a third-order semi-implicit Runge–Kutta scheme is employed[42]. Constant grid spacings of approximately 8.2 and 4.1 viscous units are used in the streamwise and spanwise directions, respectively for the $Re_\tau = 125$ case, and 8.9 and 4.3 for the $Re_\tau = 550$ case. The wall-normal grid spacing is adjusted along the wall-normal distance to maintain the ratio between the grid spacing and the local isotropic Kolmogorov scale ($\eta$) constant: $\Delta y = 1.5\eta$. The local isotropic Kolmogorov scale is defined as $\eta = \left(\nu^3/\varepsilon\right)^{1/4}$, where $\varepsilon = 2\nu\left(\partial u_i/\partial x_j \cdot \partial u_j/\partial x_i\right)$ is the viscous dissipation rate and $\nu$ the kinematic viscosity.

A turbulent channel with dimensions $8\pi h \times 2h \times 3\pi h$ in the streamwise, wall-normal and spanwise dimensions is simulated for $Re_\tau = 125$ and $2\pi h \times 2h \times \pi h$ for $Re_\tau = 550$, where $h$ is half the distance between the channel walls. These dimensions are selected to ensure that the domain is large enough to include multiple coherent structures of each type in both the streamwise and spanwise directions, as can be observed in Fig. 2. The previous grid spacing and channel size generate a mesh containing $384 \times 201 \times 288$ points in the streamwise, wall-normal and spanwise directions for $Re_\tau = 125$ and $384 \times 251 \times 384$ for $Re_\tau = 550$, which matches the initial and final sizes of the data used in the U-net presented in Fig. 1.

### Explainable deep-learning model

This manuscript focuses on the use of explainable artificial intelligence (XAI) for the quantification of the importance of each grid point in a turbulent channel when predicting a future state of the flow. As presented in Fig. 1, the methodology of the paper can be divided into three main stages: prediction of the evolution of the flow using a deep-learning model, estimation of the importance of the flow field using the gradient-SHAP algorithm based on the expected-gradient methodology[30] and segmentation of the domain in terms of the importance scores of each grid point (SHAP values).

**Prediction of the flow field.** Regarding the deep-learning model, a U-net architecture[32], similar to the one used in Cremades et al.[26], is selected for predicting the future state of the flow. The U-net takes the three-dimensional fields of the 3 velocity fluctuations at an instant $t$ as input and predicts the three fluctuation fields five viscous times into the future (the viscous time is defined in terms of $\nu$ and $u_\tau^2$). Note that, in a previous work[26], we showed that these results were consistent for a range between one to five viscous times. A schematic representation of the U-net can be observed in the central block of Fig. 1. The architecture comprises four layers in the case of $Re_\tau = 125$, where the first layer has a size of $201 \times 288 \times 384$ points, which is padded into the shape $201 \times 318 \times 414$ (blue arrow) in the first operation, exploiting the periodicity of the channel to avoid any possible error in the edges of the convolution, and cropped back (light blue arrows) in the last operation. In the encoder, two three-dimensional convolutions are applied with 32 filters each one; in the decoder, the first three-dimensional convolution with 32 filters is followed by a convolution with 3 filters for the output field adapting the information to the output field. The second, third, and fourth layers exhibit field sizes of $100 \times 159 \times 207$, $50 \times 79 \times 103$, and $25 \times 39 \times 51$, with 64, 128 and 256 filters respectively. The different levels contain two three-dimensional convolutions (violet arrows) in the encoder and one in the decoder. In the encoder, the information is compressed from one level to the following through a three-dimensional mean pooling (pink arrows), and it is expanded in the decoder using transposed three-dimensional convolutions (gray arrows). In addition, the equivalent level in the

encoder and the decoder are connected with concatenations that connect the final field of the encoder level with the output of the transposed convolutions, doubling the number of channels of the tensors. For the $Re_\tau = 550$ case the number of filters is adjusted to 24, 48, 96 and 192 depending on the layer and the sizes of the tensors are modified according to the size of the input field. The U-net architectures are commonly used for image segmentation as they exploit the spatial patterns of the input field and they keep the low- and high-frequency information of the initial field[43–45].

The U-net is trained on a database comprising a total of 10000 instantaneous flow fields of a turbulent channel at a friction Reynolds number of $Re_\tau = 125$ and $Re_\tau = 550$ respectively, using 80% of them for training and 20% for validation until the relative error between the predicted output and the ground truth is approximately 1% in the three velocity components (1.15% in the streamwise, 0.98% in the wall-normal, and 1.08% in the spanwise velocity fluctuations for $Re_\tau = 125$ and 0.74%, 1.15% and 0.88% for $Re_\tau = 550$). Then a total of 8000 fields are used for calculating the SHAP values of the flow. For the training process, a RMSprop gradient-descent algorithm was used[46], with a learning rate of $5 \times 10^{-5}$ and a momentum of 0.9.

**Calculation of the Shapley additive explanations.** In the second stage of the methodology, the gradient-SHAP algorithm[27,30] is used to calculate the importance scores of each of the grid points to predict the output velocity fields. In order to quantify the accuracy of the prediction of the flow, an extra layer is added to the output of the model, as presented in the gray central box of Fig. 1. This layer calculates the mean-squared error (MSE) of the predicted flow compared with the ground truth in the following predicted step. Therefore, a three-dimensional field with three channels (streamwise, wall-normal, and spanwise velocity components) is converted into a single value, which quantifies the error in the predictions. The contribution of each grid point to the MSE is quantified by the local SHAP value:

$$\text{MSE} = \phi_0 + \sum_{j \in (u,v,w)} \sum_{i=0}^{N} \phi_{ji} z_{ji}, \tag{4}$$

where $\phi_0$ is the MSE of the prediction of the reference (non-informative) field, $\phi_{ji}$ is the SHAP value relative to the velocity component $j$ and the grid point $i$ and $z_{ji}$ is a boolean value, 0 if the grid point value is taken from the reference field (non-informative) or 1 if it is taken from the input field.

The gradient SHAP calculates the importance scores of each input grid point using the expected-gradients (EG) methodology. The EG methodology modifies the integrated gradient method[31] to decrease its computational requirements. The methodology used in this paper is based on a slight modification of the expected gradients to calculate the importance of the input features (grid points). The original EG calculate the expectations of the SHAP values relative to a subset of fields used as a reference. In the present methodology, a single reference field is employed for the calculation. This reference or non-informative field is the mean-velocity field. This methodology is defined by the following expression:

$$\phi_i(x_{\text{in}}) = \mathbb{E}_{x_{\text{ref}}, \alpha \sim U(0,1)} \left[ \left(x_{\text{in}_i} - x_{\text{ref}_i}\right) \frac{\partial F\left(x_{\text{ref}} + \alpha\left(x_{\text{in}} - x_{\text{ref}}\right)\right)}{\partial x_{\text{in}_i}} \right], \tag{5}$$

where $\phi_i(x_{\text{in}})$ are the SHAP values (expected gradients) of the feature $i$ from the input field $x_{\text{in}}$, $x_{\text{ref}}$ is the reference field, $F$ the model that calculates the MSE of the predictions and $\alpha$ is a parameter in the range {0, 1} used for interpolating an intermediate field from the reference to the input field, $x_{\text{ref}} + \alpha(x_{\text{in}} - x_{\text{ref}})$. As previously stated, the equation of the expected gradients is modified with respect to the original[30], changing the expectation over a range of reference fields to a single reference field (the mean velocity flow).

Calculating the expected gradients for every grid point generates a field based on the importance of each node of the input field for the prediction. Furthermore, the gradient-SHAP methodology generates spatial noise[30,47,48]. Therefore, in order to smoothen the solution, the periodic boundaries of the turbulent channel are exploited. The channel is moved along these periodic boundaries, calculating 10 estimations of the importance field. These displacements, which do not affect the physics of the flow, and are averaged to remove the noise of the SHAP field. For a deeper understanding of this methodology, an example is provided in the Supplementary Material.

**Segmentation of the domain using Shapley additive explanations.** Finally, after the SHAP field has been calculated, the percolation analysis presented in Equation (1) is computed. The percolation is employed to generate high-importance regions, which we denote SHAP structures. These structures are compared with other coherent structures widely studied in the literature: intense Reynolds-stress structures[36], streaks[17] and vortices[19]. The selection of the percolation analysis is a crucial point in the analysis of the high-importance regions of the flow. For the analysis presented in the main paper, the absolute SHAP value is used to select the high-importance regions for each wall-normal location due to its simplicity and clear physical interpretation: the structures with a higher absolute SHAP value are the ones affecting the predictions the most.

**Calculation of vortices**
The vortices are regions in which the effect of rotation is larger than that of shear. These regions describe closed or spiral patterns[49], in other words, these regions contain the points in which the eigenvalues of the velocity gradient tensor, $\left(A_{ij}\right) = \nabla\boldsymbol{u}$, are complex. The eigenvalues, $\lambda$, are calculated from equation (6):

$$\lambda^3 - P\lambda^2 + Q\lambda - R = 0, \tag{6}$$

where $P$, $Q$ and $R$ are the invariants of the velocity gradient tensor, defined in equations (7), (8) and (9), respectively:

$$P = A_{ii}, \text{ for incompressible flows}: P = 0, \tag{7}$$

$$Q = \frac{1}{2}\left(P^2 - S_{ij}S_{ji} - \Omega_{ij}\Omega_{ji}\right), \tag{8}$$

$$R = \frac{1}{3}\left(-P^3 + 3PQ - S_{ij}S_{jk}S_{ki} - 3\Omega_{ij}\Omega_{jk}S_{ki}\right), \tag{9}$$

where $S_{ij} = (A_{ij} + A_{ji})/2$ is the symmetric part or rate-of-strain tensor and $\Omega_{ij} = (A_{ij} - A_{ji})/2$ is its antisymmetric part or rate-of-rotation tensor. The velocity-gradient tensor is calculated as the sum of both: $A_{ii} = S_{ij} + \Omega_{ij}$.

As previously stated, the vortices are located in the regions in which the eigenvalues are complex. For incompressible flows, the previous condition requires a positive discriminant, $D > 0$:

$$D = Q^3 + \frac{27}{4}R^2 > 0. \tag{10}$$

Finally, a percolation analysis is required to maximize the number of structures in the flow[35]:

$$D(x, y, z, t) > \gamma D'(y) , \tag{11}$$

where $\gamma$ is the percolation index, and $D'(y)$ is the standard deviation of the discriminant as a function of the wall-normal distance.

## Data availability

The minimum representative downsampled data used in this study will be made available open access at: https://github.com/KTH-FlowAI/XAI_turbulentchannel_3d_simplified.git. For the complete database, please contact the authors.

## Code availability

The codes used for this work is available open access at: https://github.com/KTH-FlowAI/XAI_turbulentchannel_3d_simplified.git.

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

## Acknowledgements

The authors acknowledge Adrián Lozano-Durán and Álvaro Martíinez-Sánchez for their support with the validation of the causal nature of the SHAP values. The deep-learning-model training was enabled by resources provided by the National Academic Infrastructure for Super-computing in Sweden (NAISS) at Berzelius (NSC), partially funded by the Swedish Research Council through grant agreement no. 2022-06725. Part of the postprocessing was made in the supercomputer Sirius of the Universitat Politècnica de València. SH has the support of grant PID2021-128676OB-I00 funded by MCIN/AEI/10.13039/501100011033 and by "ERDF A way of making Europe", by the European Union. RV acknowledges the financial support from ERC grant no. 2021-CoG-101043998, DEEPCONTROL. Views and opinions expressed are however those of the author(s) only and do not necessarily reflect those of the European Union or the European Research Council. Neither the European Union nor the granting authority can be held responsible for them.

## Author contributions

A. C.: Conceptualization, Methodology, Software, Validation, Investigation, Writing - Original Draft, Visualization. S. H.: Conceptualization, Data curation, resources, Writing - Original Draft, Funding acquisition. R. V.: Initial idea, conceptualization, project definition, methodology, resources, validation, Writing - Original Draft, Supervision, Project administration, Funding acquisition.

## Funding

## Competing interests

The authors declare no competing interests.
