## [Transparent Peer Review file · Nature Communications]

Classically studied coherent structures only paint a partial picture of wall-bounded turbulence

Corresponding Author: Professor Ricardo Vinuesa

Version 0:

Reviewer comments:

Reviewer #1

(Remarks to the Author)

The paper illustrates a data-driven methodology for identifying high-importance regions in a turbulent flow, using the gradient SHAP method to estimate the importance of each grid point in a turbulent channel for the prediction of its future states. The idea is very well presented and the study compares its outcome against the usage of typical methods (Q events, streaks, and vortices) in the case of wall-bounded turbulence: this analysis is well detailed, very clear and it shows good agreement with established methods, which is satisfying since the deep-learning model does not include prior knowledge of any kind. The only complaint here is the choice of colors in some plots that are not so contrasting.

The work is a practical implementation of the ideas presented in their previous paper "Additive-feature attribution methods: a review on explainable artificial intelligence for fluid dynamics and heat transfer" and the novelty of this study lies in checking if the method can compare and supersede the known approaches commonly used in the field: this is hard to tell from a single comparison, but presented result is promising.

(Remarks on code availability)

The code is working and well commented, presented with a docker that eases the installation

Reviewer #2

(Remarks to the Author)

The authors introduce a method to calculate important regions in turbulent flows, where importance is defined using SHAP-based values of a surrogate model. The approach is applied to calculate relevant structures in a turbulent channel flow at a low Reynolds number and is compared with classical definitions of coherent structures, exhibiting high resemblance.

- Although the application of the SHAP method for structure identification is very interesting, the conclusions drawn from it seem to be very similar to those from classical analyses. This is a valuable result. However, it would be more compelling to showcase the method in a situation where classical analysis fails, in order to extract new physical insights. In my opinion this is the main weakness of the work.

- The definition of 'important' used by the authors is based on SHAP values. However, there is no discussion about the conceptual meaning of this definition and why it is an advantageous choice compared to other definitions in the literature. For example, the definition of Q-events assumes that importance is based on wall-normal momentum transfer, which is critical for maintaining turbulence. I am not suggesting that this definition is better than the SHAP-based definition, but simply pointing out its rationale. It would be important to discuss the rationale behind using SHAP as a definition of importance.

- Related to the comment above, the explanation of SHAP values presented in the methodology does not provide much clarity on the conceptual meaning of the approach or how the method quantifies importance. It would be valuable to include simple examples and/or validation to enhance understanding.

- The paper might be too technical for a general audience. For example, the abstract describes Q-events, which, despite their importance in the field, are not familiar to many experts outside the wall-bounded turbulence community.

- What is the role of the time scale in the U-Net used to predict the flow? The value chosen is 5-plus units, which is comparable to the shortest physical time scale in the flow. For such a short time scale, a model based on linearized Navier–Stokes equations might perform quite well. Either way, there is no guarantee that the important structures identified for 5-plus units are relevant for longer time scales. The work would be strengthened by an analysis of longer time lags.

- I found the choice of Reynolds number ($Re_\tau = 125$) curious, as the smallest Re_τ typically considered in the literature is $Re_\tau = 180$. I assume this is due to computational cost, but I am surprised that the flow does not laminarize at such a low value.

- The indirect connection between turbulence and climate change in the introduction feels tenuous. While turbulence is a key component, there are too many factors involved to make a decisive connection. This seems like a bit of a stretch. The topic of turbulence is important enough without the need for overselling it.

Overall, I see merit in the paper and believe it is a meaningful contribution to the field of turbulence. Nonetheless, it is not entirely clear that it belongs in Nat. Comm. To be honest, I think it would have more impact in a specialized journal, where researchers can better appreciate the technicalities and relevance of the work.

(Remarks on code availability)

Reviewer #3

(Remarks to the Author)
As the attached.

(Remarks on code availability)

Version 1:

Reviewer comments:

Reviewer #1

(Remarks to the Author)

The authors answered the few open questions and also added another application of the method to study the flow around a square wall-mounted obstacle. Everything is well detailed and convincing

(Remarks on code availability)

The code is working and well commented, presented with a docker that eases the installation.

Reviewer #2

(Remarks to the Author)

I'm satisfied with the modifications in the manuscript and I recommend it for publication.

(Remarks on code availability)

REVIEWER COMMENTS

Reviewer #1 (Remarks to the Author):

The paper illustrates a data-driven methodology for identifying high-importance regions in a turbulent flow, using the gradient SHAP method to estimate the importance of each grid point in a turbulent channel for the prediction of its future states. The idea is very well presented and the study compares its outcome against the usage of typical methods (Q events, streaks, and vortices) in the case of wall-bounded turbulence: this analysis is well detailed, very clear and it shows good agreement with established methods, which is satisfying since the deep-learning model does not include prior knowledge of any kind. **The only complaint here is the choice of colors in some plots that are not so contrasting.**

The work is a practical implementation of the ideas presented in their previous paper "Additive-feature attribution methods: a review on explainable artificial intelligence for fluid dynamics and heat transfer" and the novelty of this study lies in checking if the method can compare and supersede the known approaches commonly used in the field: this is hard to tell from a single comparison, but presented result is promising.

We would like to express our gratitude for the reviewer's kind words and constructive feedback. We also fully agree with the reviewer's perspective. While the comparison may be challenging, the primary aim of this paper is to emphasize the distinction between classical and data-driven approaches. This work serves as a starting point for the data-driven analysis of complex flows, and we intend to extend these results in the future by exploring higher Reynolds numbers and different flow characteristics.

We also agree with the comment regarding the color choices. We recognize that maximizing the contrast of the figures is crucial for effectively communicating new ideas. For this reason, we have made efforts to use a sequential colormap (Viridis) wherever possible to enhance figure readability, particularly for individuals with vision impairments. In certain cases, we changed the colormap to qualitative tab10, when it could improve the visualization. Below, we expose the selection of the colormaps for the different plots and we exemplify them.

In this sense, the sequential colormap Viridis is maintained in those figures presenting gradients, so they can be visualized in black and white and for people with vision impairments. These figures are the joint pdfs, the visualization of the structures and the visualization of fields:

For those figures that require discrete colors, such as the lines of coincidence, the bar figures and the visualization of coincidence between structures, the commonly used qualitative colormap `tab10` has been used. The idea behind this selection is improving the visualization of the different categories.

Some examples of how these plots change are presented below:

We hope that we could address this comment to the Reviewer's satisfaction.

Reviewer #1 (Remarks on code availability):

The code is working and well commented, presented with a docker that eases the installation.

We appreciate the kind words of the reviewer.

REVIEWER COMMENTS

Reviewer #2 (Remarks to the Author):

The authors introduce a method to calculate important regions in turbulent flows, where importance is defined using SHAP-based values of a surrogate model. The approach is applied to calculate relevant structures in a turbulent channel flow at a low Reynolds number and is compared with classical definitions of coherent structures, exhibiting high resemblance.

We appreciate the reviewer's feedback and recognize the value of the questions and suggestions provided in enhancing the work. We hope we could address the comments below to the reviewer's satisfaction.

- Although the application of the SHAP method for structure identification is very interesting, the conclusions drawn from it seem to be very similar to those from classical analyses. This is a valuable result. However, it would be more compelling to showcase the method in a situation where classical analysis fails, in order to extract new physical insights. In my opinion this is the main weakness of the work.

We appreciate this insightful comment. We would like to note that, despite some similarities preset in the JPFDs shown in Figure 3 from the original article at $Re_\tau = 125$, the actual agreement between SHAP and Q events was just around 60% (Figure 4 from the original article). The average agreement with vortices was below 20%, and the streaks only had a significant agreement at $y^+=15$, with that agreement quickly dropping elsewhere. This can also be observed in the figure below:

In the revised version of the manuscript we have extended the analysis to $Re_\tau = 550$, with the following levels of coincidence among the various structures:

Interestingly, the average agreement between SHAP and Q events is reduced to 50%, and the low agreement with the vortices is maintained. The streaks maintain good agreement in the near-wall region, which quickly declines elsewhere. This shows that the regions of importance identified with SHAP are different from the classically studied coherent structures, although there are some localized areas of higher agreement.

Following the reviewer's recommendation, we have applied the SHAP analysis to another case, namely the flow around a square wall-mounted obstacle. We expect the classically studied structures to play different roles in this flow due to the significantly diminished importance of the wall.

The following modification was added to the revised manuscript:

- To fill this gap, here we show a data-driven methodology for objectively identifying high-importance regions in a turbulent flow up to a friction Reynolds number of 550.
- Based on the gradient-SHAP methodology we determine an importance score for each grid point of two turbulent channels, the first one with a friction Reynolds number $Re_{\tau} = 125$ and the second one with $Re_{\tau} = 550$.
- Figure 2 has been modified:

Figure 2: Instantaneous visualization of the various coherent structures in the channel flow. The figure shows four types of coherent structures (SHAP-based structures, Reynolds-stress structures or Q events, streaks and vortices from top to bottom) in half of the channel colored by wall-normal distance (purple near the wall and yellow in the mid-plane of the channel) for $Re_\tau = 125$ (left) and $Re_\tau = 550$ (right). In the figure, the wall is located at $y^+ = 0$ and the mid-plane of the channel is $y^+ = 125$ or $y^+ = 550$ depending on the case. In addition, we use periodic boundary conditions in x and z for both cases. Note that the same flow field is used for the four panels of each channel.

- The structures are presented in the second row of Figure 2 and are defined as:
- where α is the percolation parameter. The streamwise streaks are shown in the third row of Figure 2.
- The visualization of the vortices is provided in the bottom row of Figure 2.
- Figure 3 has been modified:

Figure 3: Joint probability density function of the streamwise velocity fluctuation and the inner-scaled wall-normal distance for the different types of structures. The figure presents the SHAP-based structures, Reynolds-stress structures, streaks and vortices from top to bottom for $Re_\tau = 125$ (left) and $Re_\tau = 550$ (right). The histogram distribution contours are defined in the logarithmic scale, representing the most likely areas with yellow colors and the most unlikely ones with purple regions.

- Figure \ref{fig:fig_3} shows that the SHAP-based structures exhibit several regions of high importance for both friction Reynolds numbers.
- As presented in Figure \ref{fig:fig_3}, the high-importance high-velocity region close to the wall is similar for both friction Reynolds numbers, despite the

higher separation of scales of $\text{Re}_\tau=550$ with respect to $\text{Re}_\tau=125$. In fact, our results show that the sweep events close to the wall constitute high-importance regions independently of the friction Reynolds number, as they transport the high-energy regions towards the wall, where the shear stress is stronger.

- Although the high-importance regions are similar for SHAP structures and Q events at $\text{Re}_\tau = 125$, larger differences are observed for $\text{Re}_\tau = 550$ due to the separation of scales. While the SHAP structures only detect a region of high importance for low-velocity events, the Q events increase the probability of sweeps.
- Note that sweeps and ejections have been widely studied in the literature as very important structures, and therefore the agreement, as well as the differences, with SHAP structures are a very important result.
- Therefore, the regions involving energy transfer between the wall-attached and wall-detached regions have been identified as SHAP structures with high importance for both friction Reynolds numbers.
- Therefore, SHAP values detect complex nonlinear patterns that were not identified by the other definitions and approaches, as could be clearly observed with the Q events far from the wall for $\text{Re}_\tau=550$.
- Figure 4 has been modified:

Figure 4: Coincidence between the various types of structures. Percentage of coincidence of pairs of the following structures: SHAP, Q events, streaks and vortices, relative to the volume of each type of the pair for $\text{Re}_\tau = 125$ (top) and $\text{Re}_\tau = 550$ (bottom).

- This co-occurrence represents around 60% of the volume of the SHAP structures, and around 40% of the volume of the intense Q events for $\text{Re}_\tau = 125$ and around 50% and 30% respectively for $\text{Re}_\tau = 550$. This coexistence can be mainly observed in the blue and green regions inside the black contours for the various wall-normal locations shown in Figure

\ref{fig:fig_5}. Thus, a significant fraction of the SHAP structures coincides with Q events throughout the channel for both friction Reynolds numbers, but specially for $\text{Re}_\tau = 125$. However, there is a non-negligible percentage, 40% and 50% for each friction Reynolds number respectively of the SHAP structures, which is not correlated with the Reynolds stress.

- Regarding the comparison between SHAP and streaks, for $\text{Re}_\tau \approx 125$ the SHAP structures are contained inside the streaks, with almost 90% agreement in both cases, as can be observed for $\text{Re}_\tau \approx 125$, where the black contours of the SHAP structures remain inside the large orange or green streaks. Finally, the agreement between SHAP and vortices is lower and remains below 20% in both cases. Moreover, part of this coincidence is shared with the streaks and the Q events: as indicated by the purple, brown and gray colors inside the black contours.
- In the present manuscript, we have presented the capabilities of explainable artificial intelligence (XAI) to estimate the importance of each grid point in a turbulent channel up to a friction Reynolds number $\text{Re}_\tau = 550$ for the prediction of its future states.
- Finally, farther from the wall, the presence of the streaks is lower and the SHAP structures mainly correlate with the ejections, although for $\text{Re}_\tau = 550$ the separation of scales is higher and sweeps are also present.
- An example of application of the present SHAP framework to another flow case, namely the flow around a wall-mounted square obstacle \cite{martinez2023,yousif2023deep}, can be found in the Supplementary Material. In this case, the SHAP values highlight the influence of the wall and the obstacle, reducing the importance of the streaks Re_τ , identifying a similar importance of the Q events and a low importance of the vortices.
- In addition, the methodology has been proved to be scalable with the friction Reynolds number, producing results directly related to the wall-normal distance independently of the Reynolds number.
- Constant grid spacings of approximately 8.2 and 4.1 viscous units are used in the streamwise and spanwise directions, respectively for the $\text{Re}_\tau = 125$ case, and 8.9 and 4.3 for the $\text{Re}_\tau = 550$ case.
- A turbulent channel with dimensions $8\pi h \times 2h \times 3\pi h$ in the streamwise, wall-normal and spanwise dimensions is simulated for $\text{Re}_\tau = 125$ and $2\pi h \times 2h \times \pi h$ for $\text{Re}_\tau = 550$, where h is half the distance between the channel walls.
- The previous grid spacing and channel size generate a mesh containing $384 \times 201 \times 288$ points in the streamwise, wall-normal and spanwise directions for $\text{Re}_\tau = 125$ and $384 \times 251 \times 384$ for $\text{Re}_\tau = 550$, which matches the initial and final sizes of the data used in the U-net presented in Figure \ref{fig:fig_1}.
- The architecture comprises four layers in the case of $\text{Re}_\tau = 125$, where the first layer has a size of $201 \times 288 \times 384$ points, which is padded into the shape $201 \times 318 \times 414$ (blue arrow) in the first

operation, exploiting the periodicity of the channel to avoid any possible error in the edges of the convolution, and cropped back (light blue arrows) in the last operation.

- For the $\text{Re}_\tau=550$ case the number of filters is adjusted to 24, 48, 96 and 192 depending on the layer and the sizes of the tensors are modified according to the size of the input field.
- The U-net is trained on a database comprising a total of 10000 instantaneous flow fields of a turbulent channel at a friction Reynolds number of $\text{Re}_\tau = 125$ and $\text{Re}_\tau = 550$ respectively, using 80% of them for training and 20% for validation until the relative error between the predicted output and the ground truth is approximately 1% in the three velocity components (1.15% in the streamwise, 0.98% in the wall-normal, and 1.08% in the spanwise velocity fluctuations for $\text{Re}_\tau = 125$ and 0.74%, 1.15% and 0.88% for $\text{Re}_\tau = 550$).

And the following section was added to the Supplementary Material:

- Supplementary Figure 9 has been modified:

Supplementary Figure 9: Joint probability density function of the three velocity components of the SHAP structures for the different wall-normal distances. The figure shows the distribution of the three velocity fluctuations, namely the streamwise, wall-normal, and spanwise components from left to right for $\text{Re}_\tau = 125$ (top) and $\text{Re}_\tau = 550$ (bottom).

- Note that the SHAP values exhibit similar distributions for both friction Reynolds numbers.
- Supplementary Figure 10 has been modified:

Supplementary Figure 10: Joint probability density function of the three velocity components of the intense Reynolds stress structures for the different wall-normal distances. The figure shows the distribution of the three velocity fluctuations, namely the streamwise, wall-normal, and spanwise components from left to right for $Re_\tau = 125$ (top) and $Re_\tau = 550$ (bottom).

- However, although in the case of $Re_\tau = 550$ a high probability of sweeps far from the wall is observed, these do not correspond to high-importance regions, which remain similar for both friction Reynolds numbers.
- The percentage of agreement between the SHAP structures and the intense Reynolds-stress structures, streaks and vortices was presented with a visualization of the structures at three different wall-normal distances in Figure 5 from the main paper. This section focuses on extending the visualization of the structures at different wall-normal distances for the whole channel size, Supplementary Figures 4, to Figure 4e.
- Supplementary Figure 11 has been modified:

Supplementary Figure 11: Joint probability density function of the three velocity components of the streaks for the different wall-normal distances. The figure shows the distribution of the three velocity fluctuations, namely the stream-wise, wall-normal, and spanwise components from left to right for $Re_\tau = 125$ (top) and $Re_\tau = 550$ (bottom).

- Supplementary Figure \ref{fig:sup_fig_1} shows that the important structures are located near $y^+ \approx 15$. This wall-normal distance matches the distribution of the streaks for both friction Reynolds numbers.
- Figure 12 has been modified:

Supplementary Figure 12: Joint probability density function of the three velocity components of the vortices for the different wall-normal distances. The figure shows the distribution of the three velocity fluctuations, namely the stream-wise, wall-normal, and spanwise components from left to right for $Re_\tau = 125$ (top) and $Re_\tau = 550$ (bottom).

- This fact agrees with the region of the joint probability density function of the SHAP structures (Supplementary Figure \ref{fig:sup_fig_1}), where they exhibit a non-negligible probability of structures in regions of low-velocity fluctuations for both friction Reynolds numbers.

- For a friction Reynolds number of $\text{Re}_\tau=125$, at $y^+\approx 3$ most of the SHAP structures are composed by intense Reynolds-stress structures or Q events, with a small presence of streaks and vortices, see Supplementary Figures \ref{fig:sup_fig_4} and \ref{fig:sup_fig_4b}.
- As the wall-normal distance is increased to $y^+\approx 6$, the streaks gain importance and there is a strong agreement between the SHAP structures and the regions in which Q events and streaks collide; note that this trend can also be observed for $y^+\approx 35$. Then, for a wall-normal distance $y^+\approx 13$ the SHAP structures are located inside the streaks, mostly where they match the Q events. For larger wall-normal distances, the SHAP structures are mostly located in regions of intense Reynolds stress, $y^+\approx 81$ and $y^+\approx 110$, and the vortices gain importance as the wall-normal distance increases.
- For a friction Reynolds number of $\text{Re}_\tau=550$, the coincidence between the coherent structures for different wall-normal distances is presented in Supplementary Figures \ref{fig:sup_fig_4c}, \ref{fig:sup_fig_4d} and \ref{fig:sup_fig_4e}. Near the wall, the intense Q events coincide with the SHAP structures. This visualization is consistent with the high-importance sweeps in Figure~3 of the main paper. As the wall-normal distance is increased, the Q events and the streaks collide and match the SHAP structures. Then, for a wall-normal distance $y^+\approx 13$, the streaks increase and the SHAP structures are included within them. After this point, the streaks become weaker and the SHAP structures are located in the regions in which the Q events and streaks are coincident. For $y^+>100$ the presence of Q events and vortices increases, matching the SHAP structures part of the Q events, in agreement with the ideas presented on the main text: the SHAP values are in better agreement with the ejections far from the wall despite the increased presence of sweeps.
- Supplementary Figure 13 has been modified:

Supplementary Figure 13: Instantaneous coincidence between Q events, streaks, and vortices for six wall-normal locations below $y^+ = 15$ for $Re_\tau = 125$. The colors used for the coincidence between structures follow this code: $Q_s \setminus (\text{streaks} \cup \text{vortices})$, $\text{streaks} \setminus (Q_s \cup \text{vortices})$, $(Q_s \cup \text{streaks}) \setminus \text{vortices}$, $\text{vortices} \setminus (Q_s \cup \text{streaks})$, $(Q_s \cup \text{vortices}) \setminus \text{streaks}$, $(\text{streaks} \cup \text{vortices}) \setminus Q_s$, $Q_s \cup \text{streaks} \cup \text{vortices}$. The SHAP structures are represented by the black solid lines. Note that the dashed lines indicate the domain used in Figure 5 of the main article.

- Supplementary Figure 14 has been modified:

Supplementary Figure 14: Instantaneous coincidence between Q events, streaks, and vortices for three wall-normal locations above $y^+ = 15$ for $Re_\tau = 125$. The colors used for the coincidence between structures follow this code: $\blacksquare Q_s \setminus (\text{streaks} \cup \text{vortices})$, $\blacksquare \text{streaks} \setminus (Q_s \cup \text{vortices})$, $\blacksquare (Q_s \cup \text{streaks}) \setminus \text{vortices}$, $\blacksquare \text{vortices} \setminus (Q_s \cup \text{streaks})$, $\blacksquare (Q_s \cup \text{vortices}) \setminus \text{streaks}$, $\blacksquare (\text{streaks} \cup \text{vortices}) \setminus Q_s$, $\blacksquare Q_s \cup \text{streaks} \cup \text{vortices}$. The SHAP structures are represented by the black solid lines. Note that the dashed lines indicate the domain used in Figure 5 of the main article.

- Supplementary Figure 15 has been modified:

Supplementary Figure 15: Instantaneous coincidence between Q events, streaks, and vortices for six wall-normal locations below $y^+ = 15$ for $Re_\tau = 550$. The colors used for the coincidence between structures follow this code: $\blacksquare Qs \setminus (\text{streaks} \cup \text{vortices})$, $\blacksquare \text{streaks} \setminus (Qs \cup \text{vortices})$, $\blacksquare (Qs \cup \text{streaks}) \setminus \text{vortices}$, $\blacksquare \text{vortices} \setminus (Qs \cup \text{streaks})$, $\blacksquare (Qs \cup \text{vortices}) \setminus \text{streaks}$, $\blacksquare (\text{streaks} \cup \text{vortices}) \setminus Qs$, $\blacksquare Qs \cup \text{streaks} \cup \text{vortices}$. The SHAP structures are represented by the black solid lines. Note that the dashed lines indicate the domain used in Figure 5 of the main article.

- Supplementary Figure 16 has been modified:

Supplementary Figure 16: Instantaneous coincidence between Q events, streaks, and vortices for three wall-normal locations above $y^+ = 15$ for $Re_\tau = 550$. The colors used for the coincidence between structures follow this code: $\blacksquare Q_s \setminus (\text{streaks} \cup \text{vortices})$, $\blacksquare \text{streaks} \setminus (Q_s \cup \text{vortices})$, $\blacksquare (Q_s \cup \text{streaks}) \setminus \text{vortices}$, $\blacksquare \text{vortices} \setminus (Q_s \cup \text{streaks})$, $\blacksquare (Q_s \cup \text{vortices}) \setminus \text{streaks}$, $\blacksquare (\text{streaks} \cup \text{vortices}) \setminus Q_s$, $\blacksquare Q_s \cup \text{streaks} \cup \text{vortices}$. The SHAP structures are represented by the black solid lines. Note that the dashed lines indicate the domain used in Figure 5 of the main article.

- Supplementary Figure 17 has been modified:

Supplementary Figure 17: Instantaneous coincidence between Q events, streaks, and vortices for three wall-normal locations above $y^+ = 15$ for $\text{Re}_\tau = 550$. The colors used for the coincidence between structures follow this code: $Q_s \setminus (\text{streaks} \cup \text{vortices})$, $\text{streaks} \setminus (Q_s \cup \text{vortices})$, $(Q_s \cup \text{streaks}) \setminus \text{vortices}$, $\text{vortices} \setminus (Q_s \cup \text{streaks})$, $(Q_s \cup \text{vortices}) \setminus \text{streaks}$, $(\text{streaks} \cup \text{vortices}) \setminus Q_s$, $Q_s \cup \text{streaks} \cup \text{vortices}$. The SHAP structures are represented by the black solid lines. Note that the dashed lines indicate the domain used in Figure 5 of the main article.

- Section in the supplementary material:

SHAP structures in the flow around a square wall-mounted obstacle

In the present study, the SHAP structures have been calculated for a turbulent channel flow at two different friction Reynolds numbers: $\text{Re}_\tau = 125$ and $\text{Re}_\tau = 550$. However, in order to illustrate the adaptability of the present methodology to any type of flow, the SHAP structures in the flow around a square wall-mounted obstacle are also analyzed. The analyzed database, described in detail in Refs. \$\sim\$ cite{martinez2023,yousif2023deep}, was obtained by means of a direct

numerical simulation (DNS) at a Reynolds number $\text{Re}_h = u_0 h / \nu = 2000$, where h is the height of the obstacle, u_0 the freestream velocity and ν the kinematic viscosity. We analyze a region of the domain on the leeward side of after the obstacle with size $2.86 h \times 2 h \times 1.24 h$. The obstacle has a cross-section $0.25 h \times 0.25 h$. The methodology of the main paper is reproduced for the obstacle flow, detecting the various coherent structures and evaluating their coincidence. It can be observed that, up to the obstacle height (for $y < h$), the agreement between SHAP structures and Q events is slightly lower than that at $\text{Re}_\tau = 125$ (with a moderate increase in the shear layers at $y > h$). Regarding the streaks, the coincidence is much lower than in the case of the channel, since the main dynamics are within the wake, with the wall having a less important role. Finally, the vortices exhibit a low level of agreement with the SHAP structures as in the channel case. This example highlights that the SHAP framework can identify the most important features in different flows, regardless of the roles played by the classical structures.

Supplementary Figure 20: Coincidence of the various coherent structures in the flow around a square wall-mounted obstacle [10, 11]. Percentage of coincidence of pairs of the following structures: SHAP, Q events, streaks and vortices, relative to the volume of each type of the pair.

We would like to note that both in the channel at higher Reynolds number and the wall-mounted obstacle, the SHAP leads to importance-based structures different from the classical ones, further strengthening its usage as a method to obtain insight into flow physics.

- The definition of 'important' used by the authors is based on SHAP values. However, there is no discussion about the conceptual meaning of this definition and why it is an advantageous choice compared to other definitions in the literature. For example, the definition of Q-events assumes that importance is based on wall-normal momentum transfer, which is critical for maintaining turbulence. I am not suggesting that this definition is better than the SHAP-based definition, but simply pointing out its rationale. It would be important to discuss the rationale behind using SHAP as a definition of importance.

We believe that this is a crucial point in the analysis, and we greatly appreciate the reviewer's suggestion. The central question in this paper is: *Which structures have the greatest influence on the evolution of the velocity field?* This question inherently shapes how the structures are defined. In this work, we use an explainable-deep-learning approach to

gain insight into flow evolution. The structures are defined based on their importance for the reconstruction of the flow. Additionally, we would like to provide a more in-depth discussion on why we consider the definition of SHAP structures to be particularly advantageous.

SHAP structures are based on the calculation of the SHapley Additive exPlanations (SHAP values, Lundberg and Lee 2017). These values are importance scores that highlight the most important regions of the flow for the prediction of the following time step. The SHAP values are based on the deep-learning model (surrogate model), which detects the causal implication of the input field for the evolution of the flow (output field). Thus, the SHAP values, as discussed in detail in our answer to the next comment (where we also added a new section in the Supplementary Material), detect those regions with a higher causal effect (Martínez-Sánchez et al., 2024).

Furthermore, we have recently compared different control strategies in turbulent channel flow aimed at reducing skin friction (Beneitez et al., 2025). In this work, we developed methods based on deep reinforcement learning aimed at minimizing the presence of Q events, streaks, and SHAP-based structures (among others). Interestingly, this work showed that the highest drag reduction is obtained precisely when the SHAP structures (which identify the most important causal connections in the flow) are minimized. The following modifications were added to the manuscript to further support this point:

- **The SHAP values quantify the causal importance of each individual grid point in the evolution of the flow (refer to the supplementary material for a detailed discussion and a comparison with the causal system examples presented by \cite{martinez2024}). As previously mentioned, due to the evolution of the model, the SHAP values determine which grid points are the most influential for the evolution of the flow. Accordingly, the SHAP structures identify the regions of the flow that should be targeted to effectively control its evolution~\cite{beneitez2025improving}.**
- **Thus, the SHAP structures can identify the Q events with higher impact for the evolution of the flow, or in other words, the most causal Q events.**
- **Since the SHAP structures objectively identify the most important regions of the flow, analyzing the differences between SHAP and other structures can provide an excellent venue to deepen our insight into wall-bounded turbulence, uncovering the most important causal relations in the system.**
- **The definition of causally important regions is a change of paradigm in the analysis of physical problems, and can be replicated in many other fields such as thermal fields or cavitation. Furthermore, it can help to develop much more efficient control strategies~\cite{beneitez2025improving}.**

Lundberg, S. and Lee, S. (2017). A unified approach to interpreting model predictions. arXiv preprint arXiv:1705.07874.

Martínez-Sánchez, Á., Arranz, G., & Lozano-Durán, A. (2024). Decomposing causality into its synergistic, unique, and redundant components. *Nature Communications*, 15(1), 9296.

Beneitez, M., Cremades, A., Guastoni, L., & Vinuesa, R. (2025). Improving turbulence control through explainable deep learning. *Preprint arXiv:2504.02354*.

- Related to the comment above, the explanation of SHAP values presented in the methodology does not provide much clarity on the conceptual meaning of the approach or how the method quantifies importance. It would be valuable to include simple examples and/or validation to enhance understanding.

We thank the Reviewer for this comment. We have added a section in the Supplementary Material with simple examples containing clearly identified causal relations, and we used the present SHAP framework to identify them. The new section is reproduced below:

Validation of the causal nature of the SHAP values

SHAP values are applied to the evolution of turbulent flows, identifying key regions that influence their development. The causal implications of the SHAP values are exploited, unveiling the cause and effect relationships inside the flow. In this section, we present a series of causal validation tests, adapted from \citet{martinez2024}, to assess the effectiveness of the explainable deep learning methodology. Each system consists of three variables, Q_1 , Q_2 and Q_3 , whose values depend on their states at the previous time steps and a noise level W_1 , W_2 and W_3 respectively:

$$\left. \begin{aligned} Q_1^{t+1} &= f_{Q_1} (Q_1^t, Q_2^t, Q_3^t) + g_{W_1} (W_1^t) \\ Q_2^{t+1} &= f_{Q_2} (Q_1^t, Q_2^t, Q_3^t) + g_{W_2} (W_2^t) \\ Q_3^{t+1} &= f_{Q_3} (Q_1^t, Q_2^t, Q_3^t) + g_{W_3} (W_3^t) \end{aligned} \right\} \quad (1)$$

Next, a deep-learning model, f , is trained to predict the evolution of the variables Q_1 , Q_2 and Q_3 : $\left[Q_1^{t+1}, Q_2^{t+1}, Q_3^{t+1} \right] = f \left(\left[Q_1^t, Q_2^t, Q_3^t \right] \right)$. The model f is a fully connected neural network with 9 hidden layers of 8 neurons each and every an output layer with 3 neurons. The model is trained on a database of 200,000 samples, with 80% used for training and 20% for testing. Finally, the SHAP values are applied to assess the influence of the variables Q_1 , Q_2 and Q_3 at time t on their evolution at time $t+1$. These values are computed over the test database and averaged to identify which input variable has the most significant impact on the predictions of the models.

The first case is a system with a mediator variable, where an intermediate variable transmits the information between the other two variables. In this system, the variable

Q_1 depends on Q_2 and Q_2 depends on Q_3 . The model is defined as follows:

$$\left. \begin{aligned} Q_1^{t+1} &= \sin(Q_2^t) + 0.001W_1^t \\ Q_2^{t+1} &= \cos(Q_3^t) + 0.01W_2^t \\ Q_3^{t+1} &= 0.5Q_3^t + 0.1W_3^t \end{aligned} \right\} \quad (2)$$

The mean SHAP values averaged over the test data, are presented in Supplementary Figure \ref{fig:barshap3}. In the figure, each color represents the contribution of the input variables Q_1^t , Q_2^t and Q_3^t to the evolution of the output variables: Q_1^{t+1} (blue), Q_2^{t+1} (orange) and Q_3^{t+1} (green). The SHAP values confirm that Q_2^t is the only variable influencing Q_1^{t+1} , while Q_3^t plays the most significant role in predicting both Q_2^{t+1} and Q_3^{t+1} , as defined by equation (\ref{eq:mediator}).

Supplementary Figure 1: SHAP evaluation of the mediator system. Mean SHAP value of the input variables Q_1^t , Q_2^t and Q_3^t for the model defined in equation (2), to predict their evolution Q_1^{t+1} in blue, Q_2^{t+1} in orange and Q_3^{t+1} in green.

The second case is a system with a cofounder variable. In this system, a single variable generates the other two, meaning that both variables Q_1 and Q_2 depend on Q_3 . The model is defined as follows:

$$\left. \begin{aligned} Q_1^{t+1} &= \sin(Q_1^t + Q_3^t) + 0.001W_1^t \\ Q_2^{t+1} &= \cos(Q_2^t - Q_3^t) + 0.001W_2^t \\ Q_3^{t+1} &= 0.5Q_3^t + 0.1W_3^t \end{aligned} \right\} \quad (3)$$

The results for the cofounder system, shown in Figure \ref{fig:barshap4}, demonstrate that the SHAP values can effectively capture the shared influence of Q_1 and Q_3 on the prediction of Q_1 . They also reflect the influence of Q_2 and Q_3 on the prediction of Q_2 . The predictions for Q_1 primarily depend on its own previous value, with a smaller contribution from Q_3 . In contrast, the cofounder variable Q_3 has a stronger influence in the prediction of Q_2 as indicated by the orange bars. This idea can be justified by analyzing the evolution of the temporal signals, Figure \ref{fig:evo4}. In this figure, the strong self-dependency of the variable Q_1 is evidenced as the high frequency of the signal Q_3 produces relatively small perturbations on the previous state of Q_1 . However, the signal Q_2 presents a higher frequency which is mostly condition by the previous state of Q_3 . This idea also evidences the capacity of the SHAP values not only to determine the causality between variables but also the intensity of the cause-effect relationships. Finally, Q_3 only depends on its previous value.

Supplementary Figure 2: SHAP evaluation of the cofounder system. Mean SHAP value of the input variables Q_1^t , Q_2^t and Q_3^t for the model defined in equation (3), to predict their evolution Q_1^{t+1} in blue, Q_2^{t+1} in orange and Q_3^{t+1} in green.

Supplementary Figure 3: Temporal evolution of the cofounder variables. The temporal evolution is presented for 30 consecutive time steps for the variables Q_1 in blue, Q_2 in orange and Q_3 in green. The time sampling is visualized by the position of the markers.

In the collider system, one variable depends on the other two. Specifically, the value of the variable Q_1^{t+1} is determined by the states of Q_2 and Q_3 . The model is defined as follows:

$$\left. \begin{aligned} Q_1^{t+1} &= \sin(Q_2^t Q_3^t) + 0.001W_1^t \\ Q_2^{t+1} &= 0.5Q_2^t + 0.1W_2^t \\ Q_3^{t+1} &= 0.5Q_3^t + 0.1W_3^t \end{aligned} \right\} \quad (4)$$

The mean SHAP values presented in Figure \ref{fig:barshap5} indicate that Q_1 is equally influenced by Q_2 and Q_3 , while Q_2 and Q_3 depend solely on their own previous value.

Supplementary Figure 4: SHAP evaluation of the collider system. Mean SHAP value of the input variables Q_1^t , Q_2^t and Q_3^t for the model defined in equation (4), to predict their evolution Q_1^{t+1} in blue, Q_2^{t+1} in orange and Q_3^{t+1} in green.

Finally, the redundant collider system is presented. In this case, two variables are identical, with Q_2 and Q_3 representing the same variable. The variable Q_1 depends on both Q_2 and Q_3 , despite them being identical. As shown in Figure \ref{fig:barshap6}, the SHAP values cannot differentiate them.

$$\left. \begin{aligned} Q_1^{t+1} &= 0.3Q_1^t + \sin(Q_2^t Q_3^t) + 0.001W_1^t \\ Q_2^{t+1} &= 0.5Q_2^t + 0.1W_2^t \\ Q_3^{t+1} &= Q_2^{t+1} \end{aligned} \right\} \quad (5)$$

Additionally, the SHAP values reveal that the influence of the sinus on the prediction of Q_1 is stronger than the effect of its previous state. The influence of the sinus in the evolution of the variable Q_1 is visualized in Figure \ref{fig:evo6}, where its variation follows Q_2 and Q_3 .

Supplementary Figure 5: SHAP evaluation of the redundant collider system. Mean SHAP value of the input variables Q_1^t , Q_2^t and Q_3^t for the model defined in equation (5), to predict their evolution Q_1^{t+1} in blue, Q_2^{t+1} in orange and Q_3^{t+1} in green.

Supplementary Figure 6: Temporal evolution of the redundant collider variables. The temporal evolution is presented for 30 consecutive time steps for the variables Q_1 in blue, Q_2 in orange and Q_3 in green. The time sampling is visualized by the position of the markers.

SHAP values effectively capture the causal contribution of input variables in a dynamic system. Deep-learning models used to predict the evolution of a dynamic system establish causal relationships between inputs and outputs. Applying SHAP values to these models reveals these relationships by identifying the variables that have the greatest influence on the evolution of the system's state. In the turbulent channel case, grid points with higher SHAP values are those that more strongly influence the prediction of the flow's next state—in other words, they play a greater causal role in its evolution.

- The paper might be too technical for a general audience. For example, the abstract describes Q-events, which, despite their importance in the field, are not familiar to many experts outside the wall-bounded turbulence community.

We thank the reviewer for pointing this out. To make the concepts more accessible, we have modified the abstract as follows:

- **Previous research has focused on analyzing the so-called coherent structures of the flow, or in other words, those regions of the flow intense enough in terms of energy production and transport, turbulent kinetic energy or rotation, to evolve on their own. However, the connection between these classically studied structures and the flow development is still uncertain. In a previous analysis, the importance of the different intense Reynolds stress structures was quantified through a data-driven methodology, showing that the calculated importance did not perfectly agree with the definition of the structures.**

In addition, in the main text we have also added explanations of the streaks and the vortices to clarify these concepts for a wider audience:

- **These are regions of high turbulent kinetic energy.**
- **In general, the vortices define those regions of the flow where rotation is larger than shear. Vortices appear close to the ejections \cite{loz14time} and contribute to the proposed near-wall cycle to sustain turbulence involving the streaks. In fact, vortices are a result of the instabilities of the streaks and can be considered dissipative as they carry high levels of enstrophy.**

- What is the role of the time scale in the U-Net used to predict the flow? The value chosen is 5-plus units, which is comparable to the shortest physical time scale in the flow. For such a short time scale, a model based on linearized Navier–Stokes equations might perform quite well. Either way, there is no guarantee that the important structures identified for 5-plus units are relevant for longer time scales. The work would be strengthened by an analysis of longer time lags.

Following the reviewer's recommendation, we have extended the analysis to a time interval of $\Delta t^+ = 10$. As can be shown in the modifications below, the results are essentially the same as the ones with $\Delta t^+ = 5$, a fact that indicates that these importance-based structures exhibit robust properties and have high importance for the near-wall dynamics. The following text was added to the main paper:

- The Supplementary Material also presents results for a time interval $\Delta t^+ = 10$, showing high levels of agreement with the analysis for $\Delta t^+ = 5$.

And a new section was added to the Supplementary Material:

SHAP structures for longer time horizons

In this section, SHAP structures are analyzed at $Re_{\tau} = 125$ for a longer time interval between input and output: $\Delta t^+ = 10$. The corresponding joint probability density functions are shown in Supplementary Figure \ref{fig:sup_fig_dt10_1}. These distributions for $\Delta t^+ = 10$ exhibit a strong similarity with those obtained for $\Delta t^+ = 5$, as illustrated in Supplementary Figure \ref{fig:sup_fig_1}. As with the shorter time step, the SHAP analysis at $\Delta t^+ = 10$ identifies regions near the wall with high velocity, and regions farther from the wall with low velocity, as most influential. Notably, for $\Delta t^+ = 10$, high-importance ejections are absent from the channel center, indicating that increasing the time interval shifts the importance toward regions with maximal velocity fluctuations, specifically around $y^+ \approx 15$.

Supplementary Figure 18: Joint probability density function of the three velocity components of the SHAP structures for the different wall-normal distances for a time horizon $\Delta t^+ = 10$. The figure shows the distribution of the three velocity fluctuations, namely the streamwise, wall-normal, and spanwise components from left to right for $Re_{\tau} = 125$.

Regarding the overlap between SHAP structures and traditional coherent structures, the results are very similar to the ones obtained for $\Delta t^+ = 5$, with a small reduction in agreement, particularly with the streaks, which account for a smaller proportion of the SHAP values. This observation reinforces the interpretation that SHAP values isolate the most influential regions of the flow rather than simply reproducing known structures.

Supplementary Figure 19: Coincidence of the coherent structures for $\Delta t^+ = 10$. Percentage of coincidence of pairs of the following structures: SHAP, Q events, streaks and vortices, relative to the volume of each type of the pair for a turbulent channel at $Re_\tau = 125$.

Furthermore, one could consider longer time horizons and even other output functions (such as the skin friction), and these would probably yield different SHAP structures. This framework allows us to identify importance-based structures given a particular objective function. In this work this objective is the near-wall dynamics, but many other options are possible, and they will surely motivate future studies. The following text was added to the main text to reflect this point:

- **Other possible models can be developed, e.g. for predictions with much longer time horizons (characteristic of the outer region) or of the wall-shear stress (where different SHAP structures may be identified). The different questions that can be formulated with various models will be addressed in future studies.**

- I found the choice of Reynolds number ($Re_\tau = 125$) curious, as the smallest Re_τ typically considered in the literature is $Re_\tau = 180$. I assume this is due to computational cost, but I am surprised that the flow does not laminarize at such a low value.

This is a very good point by the reviewer. We initially studied the flow at a low Reynolds number due to computational constraints, and the flow always remained turbulent partly thanks to the very large computational box we employed. We have significantly improved the efficiency of the algorithm, and have now been able to repeat the analysis at a significantly higher $Re_\tau = 550$. The conclusions are very similar to the ones at $Re_\tau = 125$, a fact that further supports the robustness and scalability of the analysis presented here.

The following text was added to the main paper:

- **To fill this gap, here we show a data-driven methodology for objectively identifying high-importance regions in a turbulent flow up to a friction Reynolds number of 550.**
- **Based on the gradient-SHAP methodology we determine an importance score for each grid point of two turbulent channels, the first one with a friction**

Reynolds number $Re_\tau = u_\tau h / \nu = 125$ and the second one with $Re_\tau = 550$.

- Figure 2 has been modified:

Figure 2: Instantaneous visualization of the various coherent structures in the channel flow. The figure shows four types of coherent structures (SHAP-based structures, Reynolds-stress structures or Q events, streaks and vortices from top to bottom) in half of the channel colored by wall-normal distance (purple near the wall and yellow in the mid-plane of the channel) for $Re_\tau = 125$ (left) and $Re_\tau = 550$ (right). In the figure, the wall is located at $y^+ = 0$ and the mid-plane of the channel is $y^+ = 125$ or $y^+ = 550$ depending on the case. In addition, we use periodic boundary conditions in x and z for both cases. Note that the same flow field is used for the four panels of each channel.

- The structures are presented in the second row of Figure 2 and are defined as:
- where α is the percolation parameter. The streamwise streaks are shown in the third row of Figure 2.
- The visualization of the vortices is provided in the bottom row of Figure 2.
- Figure 3 has been modified:

Figure 3: Joint probability density function of the streamwise velocity fluctuation and the inner-scaled wall-normal distance for the different types of structures. The figure presents the SHAP-based structures, Reynolds-stress structures, streaks and vortices from top to bottom for $Re_\tau = 125$ (left) and $Re_\tau = 550$ (right). The histogram distribution contours are defined in the logarithmic scale, representing the most likely areas with yellow colors and the most unlikely ones with purple regions.

- Figure \ref{fig:fig_3} shows that the SHAP-based structures exhibit several regions of high importance for both friction Reynolds numbers.
- As presented in Figure \ref{fig:fig_3}, the high-importance high-velocity region close to the wall is similar for both friction Reynolds numbers, despite the

higher separation of scales of $\text{Re}_\tau=550$ with respect to $\text{Re}_\tau=125$. In fact, our results show that the sweep events close to the wall constitute high-importance regions independently of the friction Reynolds number, as they transport the high-energy regions towards the wall, where the shear stress is stronger.

- Although the high-importance regions are similar for SHAP structures and Q events at $\text{Re}_\tau = 125$, larger differences are observed for $\text{Re}_\tau = 550$ due to the separation of scales. While the SHAP structures only detect a region of high importance for low-velocity events, the Q events increase the probability of sweeps.
- Note that sweeps and ejections have been widely studied in the literature as very important structures, and therefore the agreement, as well as the differences, with SHAP structures are a very important result.
- Therefore, the regions involving energy transfer between the wall-attached and wall-detached regions have been identified as SHAP structures with high importance for both friction Reynolds numbers.
- Therefore, SHAP values detect complex nonlinear patterns that were not identified by the other definitions and approaches, as could be clearly observed with the Q events far from the wall for $\text{Re}_\tau=550$.
- Figure 4 has been modified:

Figure 4: Coincidence between the various types of structures. Percentage of coincidence of pairs of the following structures: SHAP, Q events, streaks and vortices, relative to the volume of each type of the pair for $\text{Re}_\tau = 125$ (top) and $\text{Re}_\tau = 550$ (bottom).

- This co-occurrence represents around 60% of the volume of the SHAP structures, and around 40% of the volume of the intense Q events for $\text{Re}_\tau = 125$ and around 50% and 30% respectively for $\text{Re}_\tau = 550$. This coexistence can be mainly observed in the blue and green regions inside the black contours for the various wall-normal locations shown in Figure

\ref{fig:fig_5}. Thus, a significant fraction of the SHAP structures coincides with Q events throughout the channel for both friction Reynolds numbers, but specially for $\text{Re}_\tau = 125$. However, there is a non-negligible percentage, 40% and 50% for each friction Reynolds number respectively of the SHAP structures, which is not correlated with the Reynolds stress.

- Regarding the comparison between SHAP and streaks, for $10 \lesssim y^+ \lesssim 50$ the SHAP structures are contained inside the streaks, with almost 90% agreement in both cases, as can be observed for $y^+ \lesssim 13$, where the black contours of the SHAP structures remain inside the large orange or green streaks. Finally, the agreement between SHAP and vortices is lower and remains below 20% in both cases. Moreover, part of this coincidence is shared with the streaks and the Q events: as indicated by the purple, brown and gray colors inside the black contours.
- In the present manuscript, we have presented the capabilities of explainable artificial intelligence (XAI) to estimate the importance of each grid point in a turbulent channel \revo{up to a friction Reynolds number $\text{Re}_\tau = 550$ } for the prediction of its future states.
- Finally, farther from the wall, the presence of the streaks is lower and the SHAP structures mainly correlate with the ejections, although for $\text{Re}_\tau = 550$ the separation of scales is higher and sweeps are also present.
- In addition, the methodology has been proved to be scalable with the friction Reynolds number, producing results directly related to the wall-normal distance independently of the Reynolds number.
- Constant grid spacings of approximately 8.2 and 4.1 viscous units are used in the streamwise and spanwise directions, respectively for the $\text{Re}_\tau = 125$ case, and 8.9 and 4.3 for the $\text{Re}_\tau = 550$ case.
- A turbulent channel with dimensions $8\pi h \times 2h \times 3\pi h$ in the streamwise, wall-normal and spanwise dimensions is simulated for $\text{Re}_\tau = 125$ and $2\pi h \times 2h \times \pi h$ for $\text{Re}_\tau = 550$, where h is half the distance between the channel walls.
- The previous grid spacing and channel size generate a mesh containing $384 \times 201 \times 288$ points in the streamwise, wall-normal and spanwise directions for $\text{Re}_\tau = 125$ and $384 \times 251 \times 384$ for $\text{Re}_\tau = 550$, which matches the initial and final sizes of the data used in the U-net presented in Figure~\ref{fig:fig_1}.
- The architecture comprises four layers in the case of $\text{Re}_\tau = 125$, where the first layer has a size of $201 \times 288 \times 384$ points, which is padded into the shape $201 \times 318 \times 414$ (blue arrow) in the first operation, exploiting the periodicity of the channel to avoid any possible error in the edges of the convolution, and cropped back (light blue arrows) in the last operation.
- For the $\text{Re}_\tau = 550$ case the number of filters is adjusted to 24, 48, 96 and 192 depending on the layer and the sizes of the tensors are modified according to the size of the input field.

- The U-net is trained on a database comprising a total of 10000 instantaneous flow fields of a turbulent channel at a friction Reynolds number of $\text{Re}_\tau = 125$ and $\text{Re}_\tau = 550$ respectively, using 80% of them for training and 20% for validation until the relative error between the predicted output and the ground truth is approximately 1% in the three velocity components (1.15% in the streamwise, 0.98% in the wall-normal, and 1.08% in the spanwise velocity fluctuations for $\text{Re}_\tau = 125$ and 0.74%, 1.15% and 0.88% for $\text{Re}_\tau = 550$).

And the following section was added to the Supplementary Material:

- Supplementary Figure 9 has been modified:

Supplementary Figure 9: Joint probability density function of the three velocity components of the SHAP structures for the different wall-normal distances. The figure shows the distribution of the three velocity fluctuations, namely the streamwise, wall-normal, and spanwise components from left to right for $\text{Re}_\tau = 125$ (top) and $\text{Re}_\tau = 550$ (bottom).

- Note that the SHAP values exhibit similar distributions for both friction Reynolds numbers.
- Supplementary Figure 10 has been modified:

Supplementary Figure 10: Joint probability density function of the three velocity components of the intense Reynolds stress structures for the different wall-normal distances. The figure shows the distribution of the three velocity fluctuations, namely the streamwise, wall-normal, and spanwise components from left to right for $Re_\tau = 125$ (top) and $Re_\tau = 550$ (bottom).

- However, although in the case of $Re_\tau = 550$ a high probability of sweeps far from the wall is observed, these do not correspond to high-importance regions, which remain similar for both friction Reynolds numbers.
- The percentage of agreement between the SHAP structures and the intense Reynolds-stress structures, streaks and vortices was presented with a visualization of the structures at three different wall-normal distances in Figure 5 from the main paper. This section focuses on extending the visualization of the structures at different wall-normal distances for the whole channel size, Supplementary Figures 4, to Figure 4e.
- Supplementary Figure 11 has been modified:

Supplementary Figure 11: Joint probability density function of the three velocity components of the streaks for the different wall-normal distances. The figure shows the distribution of the three velocity fluctuations, namely the streamwise, wall-normal, and spanwise components from left to right for $Re_\tau = 125$ (top) and $Re_\tau = 550$ (bottom).

- Supplementary Figure \ref{fig:sup_fig_1} shows that the important structures are located near $y^+ \approx 15$. This wall-normal distance matches the distribution of the streaks for both friction Reynolds numbers.
- Figure 12 has been modified:

Supplementary Figure 12: Joint probability density function of the three velocity components of the vortices for the different wall-normal distances. The figure shows the distribution of the three velocity fluctuations, namely the streamwise, wall-normal, and spanwise components from left to right for $Re_\tau = 125$ (top) and $Re_\tau = 550$ (bottom).

- This fact agrees with the region of the joint probability density function of the SHAP structures (Supplementary Figure \ref{fig:sup_fig_1}), where they exhibit a non-negligible probability of structures in regions of low-velocity fluctuations for both friction Reynolds numbers.

- For a friction Reynolds number of $\text{Re}_\tau=125$, at $y^+\approx 3$ most of the SHAP structures are composed by intense Reynolds-stress structures or Q events, with a small presence of streaks and vortices, see Supplementary Figures \ref{fig:sup_fig_4} and \ref{fig:sup_fig_4b}.
- As the wall-normal distance is increased to $y^+\approx 6$, the streaks gain importance and there is a strong agreement between the SHAP structures and the regions in which Q events and streaks collide; note that this trend can also be observed for $y^+\approx 35$. Then, for a wall-normal distance $y^+\approx 13$ the SHAP structures are located inside the streaks, mostly where they match the Q events. For larger wall-normal distances, the SHAP structures are mostly located in regions of intense Reynolds stress, $y^+\approx 81$ and $y^+\approx 110$, and the vortices gain importance as the wall-normal distance increases.
- For a friction Reynolds number of $\text{Re}_\tau=550$, the coincidence between the coherent structures for different wall-normal distances is presented in Supplementary Figures \ref{fig:sup_fig_4c}, \ref{fig:sup_fig_4d} and \ref{fig:sup_fig_4e}. Near the wall, the intense Q events coincide with the SHAP structures. This visualization is consistent with the high-importance sweeps in Figure~3 of the main paper. As the wall-normal distance is increased, the Q events and the streaks collide and match the SHAP structures. Then, for a wall-normal distance $y^+\approx 13$, the streaks increase and the SHAP structures are included within them. After this point, the streaks become weaker and the SHAP structures are located in the regions in which the Q events and streaks are coincident. For $y^+>100$ the presence of Q events and vortices increases, matching the SHAP structures part of the Q events, in agreement with the ideas presented on the main text: the SHAP values are in better agreement with the ejections far from the wall despite the increased presence of sweeps.
- Supplementary Figure 13 has been modified:

Supplementary Figure 13: Instantaneous coincidence between Q events, streaks, and vortices for six wall-normal locations below $y^+ = 15$ for $Re_\tau = 125$. The colors used for the coincidence between structures follow this code: $Q_s \setminus (\text{streaks} \cup \text{vortices})$, $\text{streaks} \setminus (Q_s \cup \text{vortices})$, $(Q_s \cup \text{streaks}) \setminus \text{vortices}$, $\text{vortices} \setminus (Q_s \cup \text{streaks})$, $(Q_s \cup \text{vortices}) \setminus \text{streaks}$, $(\text{streaks} \cup \text{vortices}) \setminus Q_s$, $Q_s \cup \text{streaks} \cup \text{vortices}$. The SHAP structures are represented by the black solid lines. Note that the dashed lines indicate the domain used in Figure 5 of the main article.

- Supplementary Figure 14 has been modified:

Supplementary Figure 14: Instantaneous coincidence between Q events, streaks, and vortices for three wall-normal locations above $y^+ = 15$ for $Re_\tau = 125$. The colors used for the coincidence between structures follow this code: $Q_s \setminus (\text{streaks} \cup \text{vortices})$, $\text{streaks} \setminus (Q_s \cup \text{vortices})$, $(Q_s \cup \text{streaks}) \setminus \text{vortices}$, $\text{vortices} \setminus (Q_s \cup \text{streaks})$, $(Q_s \cup \text{vortices}) \setminus \text{streaks}$, $(\text{streaks} \cup \text{vortices}) \setminus Q_s$, $Q_s \cup \text{streaks} \cup \text{vortices}$. The SHAP structures are represented by the black solid lines. Note that the dashed lines indicate the domain used in Figure 5 of the main article.

- Supplementary Figure 15 has been modified:

Supplementary Figure 15: Instantaneous coincidence between Q events, streaks, and vortices for six wall-normal locations below $y^+ = 15$ for $Re_\tau = 550$. The colors used for the coincidence between structures follow this code: $\blacksquare Qs \setminus (\text{streaks} \cup \text{vortices})$, $\blacksquare \text{streaks} \setminus (Qs \cup \text{vortices})$, $\blacksquare (Qs \cup \text{streaks}) \setminus \text{vortices}$, $\blacksquare \text{vortices} \setminus (Qs \cup \text{streaks})$, $\blacksquare (Qs \cup \text{vortices}) \setminus \text{streaks}$, $\blacksquare (\text{streaks} \cup \text{vortices}) \setminus Qs$, $\blacksquare Qs \cup \text{streaks} \cup \text{vortices}$. The SHAP structures are represented by the black solid lines. Note that the dashed lines indicate the domain used in Figure 5 of the main article.

- Supplementary Figure 16 has been modified:

Supplementary Figure 16: Instantaneous coincidence between Q events, streaks, and vortices for three wall-normal locations above $y^+ = 15$ for $Re_\tau = 550$. The colors used for the coincidence between structures follow this code: $\blacksquare Q_s \setminus (\text{streaks} \cup \text{vortices})$, $\blacksquare \text{streaks} \setminus (Q_s \cup \text{vortices})$, $\blacksquare (Q_s \cup \text{streaks}) \setminus \text{vortices}$, $\blacksquare \text{vortices} \setminus (Q_s \cup \text{streaks})$, $\blacksquare (Q_s \cup \text{vortices}) \setminus \text{streaks}$, $\blacksquare (\text{streaks} \cup \text{vortices}) \setminus Q_s$, $\blacksquare Q_s \cup \text{streaks} \cup \text{vortices}$. The SHAP structures are represented by the black solid lines. Note that the dashed lines indicate the domain used in Figure 5 of the main article.

- Supplementary Figure 17 has been modified:

Supplementary Figure 17: Instantaneous coincidence between Q events, streaks, and vortices for three wall-normal locations above $y^+ = 15$ for $Re_\tau = 550$. The colors used for the coincidence between structures follow this code: $Q_s \setminus (\text{streaks} \cup \text{vortices})$, $\text{streaks} \setminus (Q_s \cup \text{vortices})$, $(Q_s \cup \text{streaks}) \setminus \text{vortices}$, $\text{vortices} \setminus (Q_s \cup \text{streaks})$, $(Q_s \cup \text{vortices}) \setminus \text{streaks}$, $(\text{streaks} \cup \text{vortices}) \setminus Q_s$, $Q_s \cup \text{streaks} \cup \text{vortices}$. The SHAP structures are represented by the black solid lines. Note that the dashed lines indicate the domain used in Figure 5 of the main article.

In fact, the following figure provides a direct comparison between both Reynolds numbers, showing the high degree of agreement between the SHAP distributions in both cases.

Figure: Joint probability density function of SHAP structures as a function of the wall-normal distance and the streamwise velocity fluctuation for the (dashed) $Re_\tau = 125$ and (solid) $Re_\tau = 550$ turbulent channel flows.

- The indirect connection between turbulence and climate change in the introduction feels tenuous. While turbulence is a key component, there are too many factors involved to make a decisive connection. This seems like a bit of a stretch. The topic of turbulence is important enough without the need for overselling it.

We understand the reviewer's point and have made the following modification to the introduction:

A complete understanding of turbulence may have worldwide implications, since 15% of the energy consumed worldwide is spent near the surface of vehicles due to turbulent effects~\cite{Jimenez2013}.

Furthermore, we implemented the following modification in the abstract to address this point:

The dissipation of energy due to turbulence is significant, and understanding turbulence physics has wide implications for energy efficiency.

Overall, I see merit in the paper and believe it is a meaningful contribution to the field of turbulence. Nonetheless, it is not entirely clear that it belongs in Nat. Comm. To be honest, I think it would have more impact in a specialized journal, where researchers can better appreciate the technicalities and relevance of the work.

We appreciate the reviewer's kind words and positive assessment of our work.

The decision of submitting the paper to Nature Communications was based on multiple reasons. The first reason is the fact that the present paper is the continuation of a study already published in Nature Communications:

Cremades, A., Hoyas, S., Deshpande, R., Quintero, P., Lellep, M., Lee, W. J., ... & Vinuesa, R. (2024). Identifying regions of importance in wall-bounded turbulence through explainable deep learning. *Nature Communications*, 15(1), 3864.

This article has had a large impact in 1 year: over 11,000 accesses, 35 citations and great online attention (97th percentile or all tracked articles of a similar age in all journals and 88th percentile in Nature Communications). Therefore, we believe that the readers of Nature Communications will be interested in the topic. Furthermore, Nature Communications can reach a broad audience, including researchers working on other complex and non-linear problems that could benefit from the advances presented in this paper. In order to reflect this idea in the main text of the paper, the following sentence has been added:

The definition of causally important regions is a change of paradigm in the analysis of physical problems, and can be replicated in many other fields such as thermal fields or cavitation. Furthermore, it can help to develop much more efficient control strategies~\cite{beneitez2025improving}.

The article by Beneitez et al. (2025) (also mentioned in a previous response) shows that using the methodology presented here to identify the most important structures in the flow can guide the most efficient control strategies to reduce turbulent drag. This may lead to a completely new paradigm in analysis and optimization of high-dimensional chaotic systems, with broad implications in a wide range of areas, further justifying the relevance of Nature Communications for this study.

Beneitez, M., Cremades, A., Guastoni, L., & Vinuesa, R. (2025). Improving turbulence control through explainable deep learning. *arXiv preprint arXiv:2504.02354*.

IREVIEWER COMMENTS

Reviewer #3 (Remarks to the Author):

This paper introduces data-driven turbulence structure detection using the SHAP algorithm. They detected the highest importance in channel turbulence. However, since the channel flow dataset has a very low Reynolds number, $Re_\tau = 125$, the length scale separation between the large-scale motion and the smallest (vertical) one is insufficient.

First and foremost, we would like to thank the reviewer for their valuable feedback. This is a very good point by the reviewer. We initially studied the flow at a low Reynolds number due to computational constraints, and the flow always remained turbulent partly thanks to the very large computational box we employed. We have significantly improved the efficiency of the algorithm, and have now been able to repeat the analysis at a significantly higher $Re_\tau = 550$. The conclusions are very similar to the ones at $Re_\tau = 125$, a fact that further supports the robustness and scalability of the analysis presented here.

The following text was added to the main paper:

- To fill this gap, here we show a data-driven methodology for objectively identifying high-importance regions in a turbulent flow up to a friction Reynolds number of 550.
- Based on the gradient-SHAP methodology we determine an importance score for each grid point of two turbulent channels, the first one with a friction Reynolds number $Re_\tau = 125$ and the second one with $Re_\tau = 550$.
- Figure 2 has been modified:

Figure 2: Instantaneous visualization of the various coherent structures in the channel flow. The figure shows four types of coherent structures (SHAP-based structures, Reynolds-stress structures or Q events, streaks and vortices from top to bottom) in half of the channel colored by wall-normal distance (purple near the wall and yellow in the mid-plane of the channel) for $Re_\tau = 125$ (left) and $Re_\tau = 550$ (right). In the figure, the wall is located at $y^+ = 0$ and the mid-plane of the channel is $y^+ = 125$ or $y^+ = 550$ depending on the case. In addition, we use periodic boundary conditions in x and z for both cases. Note that the same flow field is used for the four panels of each channel.

- The structures are presented in the second row of Figure 2 and are defined as:
- where α is the percolation parameter. The streamwise streaks are shown in the third row of Figure 2.
- The visualization of the vortices is provided in the bottom row of Figure 2.
- Figure 3 has been modified:

Figure 3: Joint probability density function of the streamwise velocity fluctuation and the inner-scaled wall-normal distance for the different types of structures. The figure presents the SHAP-based structures, Reynolds-stress structures, streaks and vortices from top to bottom for $Re_\tau = 125$ (left) and $Re_\tau = 550$ (right). The histogram distribution contours are defined in the logarithmic scale, representing the most likely areas with yellow colors and the most unlikely ones with purple regions.

- Figure \ref{fig:fig_3} shows that the SHAP-based structures exhibit several regions of high importance for both friction Reynolds numbers.
- As presented in Figure \ref{fig:fig_3}, the high-importance high-velocity region close to the wall is similar for both friction Reynolds numbers, despite the

higher separation of scales of $\text{Re}_\tau=550$ with respect to $\text{Re}_\tau=125$. In fact, our results show that the sweep events close to the wall constitute high-importance regions independently of the friction Reynolds number, as they transport the high-energy regions towards the wall, where the shear stress is stronger.

- Although the high-importance regions are similar for SHAP structures and Q events at $\text{Re}_\tau = 125$, larger differences are observed for $\text{Re}_\tau = 550$ due to the separation of scales. While the SHAP structures only detect a region of high importance for low-velocity events, the Q events increase the probability of sweeps.
- Note that sweeps and ejections have been widely studied in the literature as very important structures, and therefore the agreement, as well as the differences, with SHAP structures are a very important result.
- Therefore, the regions involving energy transfer between the wall-attached and wall-detached regions have been identified as SHAP structures with high importance for both friction Reynolds numbers.
- Therefore, SHAP values detect complex nonlinear patterns that were not identified by the other definitions and approaches, as could be clearly observed with the Q events far from the wall for $\text{Re}_\tau=550$.
- Figure 4 has been modified:

Figure 4: Coincidence between the various types of structures. Percentage of coincidence of pairs of the following structures: SHAP, Q events, streaks and vortices, relative to the volume of each type of the pair for $\text{Re}_\tau = 125$ (top) and $\text{Re}_\tau = 550$ (bottom).

- This co-occurrence represents around 60% of the volume of the SHAP structures, and around 40% of the volume of the intense Q events for $\text{Re}_\tau = 125$ and around 50% and 30% respectively for $\text{Re}_\tau = 550$. This coexistence can be mainly observed in the blue and green regions inside the black contours for the various wall-normal locations shown in Figure

\ref{fig:fig_5}. Thus, a significant fraction of the SHAP structures coincides with Q events throughout the channel for both friction Reynolds numbers, but specially for $\text{Re}_\tau = 125$. However, there is a non-negligible percentage, 40% and 50% for each friction Reynolds number respectively of the SHAP structures, which is not correlated with the Reynolds stress.

- Regarding the comparison between SHAP and streaks, for $\text{Re}_\tau \approx 10$ the SHAP structures are contained inside the streaks, with almost 90% agreement in both cases, as can be observed for $\text{Re}_\tau \leq 13$, where the black contours of the SHAP structures remain inside the large orange or green streaks. Finally, the agreement between SHAP and vortices is lower and remains below 20% in both cases. Moreover, part of this coincidence is shared with the streaks and the Q events: as indicated by the purple, brown and gray colors inside the black contours.
- In the present manuscript, we have presented the capabilities of explainable artificial intelligence (XAI) to estimate the importance of each grid point in a turbulent channel (up to a friction Reynolds number $\text{Re}_\tau = 550$) for the prediction of its future states.
- Finally, farther from the wall, the presence of the streaks is lower and the SHAP structures mainly correlate with the ejections, although for $\text{Re}_\tau = 550$ the separation of scales is higher and sweeps are also present.
- In addition, the methodology has been proved to be scalable with the friction Reynolds number, producing results directly related to the wall-normal distance independently of the Reynolds number.
- Constant grid spacings of approximately 8.2 and 4.1 viscous units are used in the streamwise and spanwise directions, respectively for the $\text{Re}_\tau = 125$ case, and 8.9 and 4.3 for the $\text{Re}_\tau = 550$ case.
- A turbulent channel with dimensions $8\pi h \times 2h \times 3\pi h$ in the streamwise, wall-normal and spanwise dimensions is simulated for $\text{Re}_\tau = 125$ and $2\pi h \times 2h \times \pi h$ for $\text{Re}_\tau = 550$, where h is half the distance between the channel walls.
- The previous grid spacing and channel size generate a mesh containing $384 \times 201 \times 288$ points in the streamwise, wall-normal and spanwise directions for $\text{Re}_\tau = 125$ and $384 \times 251 \times 384$ for $\text{Re}_\tau = 550$, which matches the initial and final sizes of the data used in the U-net presented in Figure~\ref{fig:fig_1}.
- The architecture comprises four layers in the case of $\text{Re}_\tau = 125$, where the first layer has a size of $201 \times 288 \times 384$ points, which is padded into the shape $201 \times 318 \times 414$ (blue arrow) in the first operation, exploiting the periodicity of the channel to avoid any possible error in the edges of the convolution, and cropped back (light blue arrows) in the last operation.
- For the $\text{Re}_\tau = 550$ case the number of filters is adjusted to 24, 48, 96 and 192 depending on the layer and the sizes of the tensors are modified according to the size of the input field.

- The U-net is trained on a database comprising a total of 10000 instantaneous flow fields of a turbulent channel at a friction Reynolds number of $\text{Re}_\tau = 125$ and $\text{Re}_\tau = 550$ respectively, using 80% of them for training and 20% for validation until the relative error between the predicted output and the ground truth is approximately 1% in the three velocity components (1.15% in the streamwise, 0.98% in the wall-normal, and 1.08% in the spanwise velocity fluctuations for $\text{Re}_\tau = 125$ and 0.74%, 1.15% and 0.88% for $\text{Re}_\tau = 550$).

And the following modifications were implemented in the Supplementary Material:

- Supplementary Figure 9 has been modified:

Supplementary Figure 9: Joint probability density function of the three velocity components of the SHAP structures for the different wall-normal distances. The figure shows the distribution of the three velocity fluctuations, namely the streamwise, wall-normal, and spanwise components from left to right for $\text{Re}_\tau = 125$ (top) and $\text{Re}_\tau = 550$ (bottom).

- Note that the SHAP values exhibit similar distributions for both friction Reynolds numbers.
- Supplementary Figure 10 has been modified:

Supplementary Figure 10: Joint probability density function of the three velocity components of the intense Reynolds stress structures for the different wall-normal distances. The figure shows the distribution of the three velocity fluctuations, namely the streamwise, wall-normal, and spanwise components from left to right for $Re_\tau = 125$ (top) and $Re_\tau = 550$ (bottom).

- However, although in the case of $Re_\tau = 550$ a high probability of sweeps far from the wall is observed, these do not correspond to high-importance regions, which remain similar for both friction Reynolds numbers.
- The percentage of agreement between the SHAP structures and the intense Reynolds-stress structures, streaks and vortices was presented with a visualization of the structures at three different wall-normal distances in Figure 5 from the main paper. This section focuses on extending the visualization of the structures at different wall-normal distances for the whole channel size, Supplementary Figures 4, to Figure 4e.
- Supplementary Figure 11 has been modified:

Supplementary Figure 11: Joint probability density function of the three velocity components of the streaks for the different wall-normal distances. The figure shows the distribution of the three velocity fluctuations, namely the streamwise, wall-normal, and spanwise components from left to right for $Re_\tau = 125$ (top) and $Re_\tau = 550$ (bottom).

- Supplementary Figure \ref{fig:sup_fig_1} shows that the important structures are located near $y^+ \approx 15$. This wall-normal distance matches the distribution of the streaks for both friction Reynolds numbers.
- Figure 12 has been modified:

Supplementary Figure 12: Joint probability density function of the three velocity components of the vortices for the different wall-normal distances. The figure shows the distribution of the three velocity fluctuations, namely the streamwise, wall-normal, and spanwise components from left to right for $Re_\tau = 125$ (top) and $Re_\tau = 550$ (bottom).

- This fact agrees with the region of the joint probability density function of the SHAP structures (Supplementary Figure \ref{fig:sup_fig_1}), where they exhibit a non-negligible probability of structures in regions of low-velocity fluctuations for both friction Reynolds numbers.

- For a friction Reynolds number of $\text{Re}_\tau=125$, at $y^+\approx 3$ most of the SHAP structures are composed by intense Reynolds-stress structures or Q events, with a small presence of streaks and vortices, see Supplementary Figures \ref{fig:sup_fig_4} and \ref{fig:sup_fig_4b}.
- As the wall-normal distance is increased to $y^+\approx 6$, the streaks gain importance and there is a strong agreement between the SHAP structures and the regions in which Q events and streaks collide; note that this trend can also be observed for $y^+\approx 35$. Then, for a wall-normal distance $y^+\approx 13$ the SHAP structures are located inside the streaks, mostly where they match the Q events. For larger wall-normal distances, the SHAP structures are mostly located in regions of intense Reynolds stress, $y^+\approx 81$ and $y^+\approx 110$, and the vortices gain importance as the wall-normal distance increases.
- For a friction Reynolds number of $\text{Re}_\tau=550$, the coincidence between the coherent structures for different wall-normal distances is presented in Supplementary Figures \ref{fig:sup_fig_4c}, \ref{fig:sup_fig_4d} and \ref{fig:sup_fig_4e}. Near the wall, the intense Q events coincide with the SHAP structures. This visualization is consistent with the high-importance sweeps in Figure~3 of the main paper. As the wall-normal distance is increased, the Q events and the streaks collide and match the SHAP structures. Then, for a wall-normal distance $y^+\approx 13$, the streaks increase and the SHAP structures are included within them. After this point, the streaks become weaker and the SHAP structures are located in the regions in which the Q events and streaks are coincident. For $y^+>100$ the presence of Q events and vortices increases, matching the SHAP structures part of the Q events, in agreement with the ideas presented on the main text: the SHAP values are in better agreement with the ejections far from the wall despite the increased presence of sweeps.
- Supplementary Figure 13 has been modified:

Supplementary Figure 13: Instantaneous coincidence between Q events, streaks, and vortices for six wall-normal locations below $y^+ = 15$ for $Re_\tau = 125$. The colors used for the coincidence between structures follow this code: $Q_s \setminus (\text{streaks} \cup \text{vortices})$, $\text{streaks} \setminus (Q_s \cup \text{vortices})$, $(Q_s \cup \text{streaks}) \setminus \text{vortices}$, $\text{vortices} \setminus (Q_s \cup \text{streaks})$, $(Q_s \cup \text{vortices}) \setminus \text{streaks}$, $(\text{streaks} \cup \text{vortices}) \setminus Q_s$, $Q_s \cup \text{streaks} \cup \text{vortices}$. The SHAP structures are represented by the black solid lines. Note that the dashed lines indicate the domain used in Figure 5 of the main article.

- Supplementary Figure 14 has been modified:

Supplementary Figure 14: Instantaneous coincidence between Q events, streaks, and vortices for three wall-normal locations above $y^+ = 15$ for $Re_\tau = 125$. The colors used for the coincidence between structures follow this code: $\blacksquare Q_s \setminus (\text{streaks} \cup \text{vortices})$, $\blacksquare \text{streaks} \setminus (Q_s \cup \text{vortices})$, $\blacksquare (Q_s \cup \text{streaks}) \setminus \text{vortices}$, $\blacksquare \text{vortices} \setminus (Q_s \cup \text{streaks})$, $\blacksquare (Q_s \cup \text{vortices}) \setminus \text{streaks}$, $\blacksquare (\text{streaks} \cup \text{vortices}) \setminus Q_s$, $\blacksquare Q_s \cup \text{streaks} \cup \text{vortices}$. The SHAP structures are represented by the black solid lines. Note that the dashed lines indicate the domain used in Figure 5 of the main article.

- Supplementary Figure 15 has been modified:

Supplementary Figure 15: Instantaneous coincidence between Q events, streaks, and vortices for six wall-normal locations below $y^+ = 15$ for $Re_\tau = 550$. The colors used for the coincidence between structures follow this code: $\blacksquare Qs \setminus (\text{streaks} \cup \text{vortices})$, $\blacksquare \text{streaks} \setminus (Qs \cup \text{vortices})$, $\blacksquare (Qs \cup \text{streaks}) \setminus \text{vortices}$, $\blacksquare \text{vortices} \setminus (Qs \cup \text{streaks})$, $\blacksquare (Qs \cup \text{vortices}) \setminus \text{streaks}$, $\blacksquare (\text{streaks} \cup \text{vortices}) \setminus Qs$, $\blacksquare Qs \cup \text{streaks} \cup \text{vortices}$. The SHAP structures are represented by the black solid lines. Note that the dashed lines indicate the domain used in Figure 5 of the main article.

- Supplementary Figure 16 has been modified:

Supplementary Figure 16: Instantaneous coincidence between Q events, streaks, and vortices for three wall-normal locations above $y^+ = 15$ for $Re_\tau = 550$. The colors used for the coincidence between structures follow this code: $\blacksquare Q_s \setminus (\text{streaks} \cup \text{vortices})$, $\blacksquare \text{streaks} \setminus (Q_s \cup \text{vortices})$, $\blacksquare (Q_s \cup \text{streaks}) \setminus \text{vortices}$, $\blacksquare \text{vortices} \setminus (Q_s \cup \text{streaks})$, $\blacksquare (Q_s \cup \text{vortices}) \setminus \text{streaks}$, $\blacksquare (\text{streaks} \cup \text{vortices}) \setminus Q_s$, $\blacksquare Q_s \cup \text{streaks} \cup \text{vortices}$. The SHAP structures are represented by the black solid lines. Note that the dashed lines indicate the domain used in Figure 5 of the main article.

- Supplementary Figure 17 has been modified:

Supplementary Figure 17: Instantaneous coincidence between Q events, streaks, and vortices for three wall-normal locations above $y^+ = 15$ for $Re_\tau = 550$. The colors used for the coincidence between structures follow this code: $Qs \setminus (\text{streaks} \cup \text{vortices})$, $\text{streaks} \setminus (Qs \cup \text{vortices})$, $(Qs \cup \text{streaks}) \setminus \text{vortices}$, $\text{vortices} \setminus (Qs \cup \text{streaks})$, $(Qs \cup \text{vortices}) \setminus \text{streaks}$, $(\text{streaks} \cup \text{vortices}) \setminus Qs$, $Qs \cup \text{streaks} \cup \text{vortices}$. The SHAP structures are represented by the black solid lines. Note that the dashed lines indicate the domain used in Figure 5 of the main article.

In fact, the following figure provides a direct comparison between both Reynolds numbers, showing the high degree of agreement between the SHAP distributions in both cases.

Figure: Joint probability density function of SHAP structures as a function of the wall-normal distance and the streamwise velocity fluctuation for the (dashed) $Re_\tau = 125$ and (solid) $Re_\tau = 550$ turbulent channel flows.

I cannot find the novelty of using the machine learning (ML) technique, which costs a lot for the learning process and is difficult to apply for higher-Re simulation datasets. Also, the main result of Figure 3 represents the SHAP algorithm, which, in my interpretation, shows the velocity fluctuation scale (i.e. large-scale fluid motion). Since this paper only uses low-Re data,

Regarding the novelty of the machine learning (ML) technique, we would like to note that this paper does not focus on the architecture used for predicting flow evolution or the accuracy of the predictions. Instead, we employ an explainable framework to uncover the complex and non-linear causal relationships within a turbulent flow. The key novelty of this work lies in defining the regions of the flow (SHAP structures) having the greatest influence on the causal correlations within the velocity field. To justify this idea, we have added a new section in the Supplementary Material to explain the causal implications of the explainable-deep-learning methodology:

Validation of the causal nature of the SHAP values

SHAP values are applied to the evolution of turbulent flows, identifying key regions that influence their development. The causal implications of the SHAP values are exploited, unveiling the cause and effect relationships inside the flow. In this section, we present a series of causal validation tests, adapted from \citet{martinez2024}, to assess the effectiveness of the explainable deep learning methodology. Each system consists of three variables, Q_1 , Q_2 and Q_3 , whose values depend on their

states at the previous time steps and a noise level W_1 , W_2 and W_3 respectively:

$$\left. \begin{aligned} Q_1^{t+1} &= f_{Q_1}(Q_1^t, Q_2^t, Q_3^t) + g_{W_1}(W_1^t) \\ Q_2^{t+1} &= f_{Q_2}(Q_1^t, Q_2^t, Q_3^t) + g_{W_2}(W_2^t) \\ Q_3^{t+1} &= f_{Q_3}(Q_1^t, Q_2^t, Q_3^t) + g_{W_3}(W_3^t) \end{aligned} \right\} \quad (1)$$

Next, a deep-learning model, f , is trained to predict the evolution of the variables Q_1 , Q_2 and Q_3 : $\left[Q_1^{t+1}, Q_2^{t+1}, Q_3^{t+1} \right] = f \left(\left[Q_1^t, Q_2^t, Q_3^t \right] \right)$. The model f is a fully connected neural network with 9 hidden layers of 8 neurons each and every an output layer with 3 neurons. The model is trained on a database of 200,000 samples, with 80% used for training and 20% for testing. Finally, the SHAP values are applied to assess the influence of the variables Q_1 , Q_2 and Q_3 at time t on their evolution at time $t+1$. These values are computed over the test database and averaged to identify which input variable has the most significant impact on the predictions of the models.

The first case is a system with a mediator variable, where an intermediate variable transmits the information between the other two variables. In this system, the variable Q_1 depends on Q_2 and Q_2 depends on Q_3 . The model is defined as follows:

$$\left. \begin{aligned} Q_1^{t+1} &= \sin(Q_2^t) + 0.001W_1^t \\ Q_2^{t+1} &= \cos(Q_3^t) + 0.01W_2^t \\ Q_3^{t+1} &= 0.5Q_3^t + 0.1W_3^t \end{aligned} \right\} \quad (2)$$

The mean SHAP values averaged over the test data, are presented in Supplementary Figure \ref{fig:barshap3}. In the figure, each color represents the contribution of the input variables Q_1^t , Q_2^t and Q_3^t to the evolution of the output variables: Q_1^{t+1} (blue), Q_2^{t+1} (orange) and Q_3^{t+1} (green). The SHAP values confirm that Q_2^t is the only variable influencing Q_1^{t+1} , while Q_3^t plays the most significant role in predicting both Q_2^{t+1} and Q_3^{t+1} , as defined by equation (\ref{eq:mediator}).

Supplementary Figure 1: SHAP evaluation of the mediator system. Mean SHAP value of the input variables Q_1^t , Q_2^t and Q_3^t for the model defined in equation (2), to predict their evolution Q_1^{t+1} in blue, Q_2^{t+1} in orange and Q_3^{t+1} in green.

The second case is a system with a cofounder variable. In this system, a single variable generates the other two, meaning that both variables Q_1 and Q_2 depend on Q_3 . The model is defined as follows:

$$\left. \begin{aligned} Q_1^{t+1} &= \sin(Q_1^t + Q_3^t) + 0.001W_1^t \\ Q_2^{t+1} &= \cos(Q_2^t - Q_3^t) + 0.001W_2^t \\ Q_3^{t+1} &= 0.5Q_3^t + 0.1W_3^t \end{aligned} \right\} \quad (3)$$

The results for the cofounder system, shown in Figure \ref{fig:barshap4}, demonstrate that the SHAP values can effectively capture the shared influence of Q_1 and Q_3 on the prediction of Q_1 . They also reflect the influence of Q_2 and Q_3 on the prediction of Q_2 . The predictions for Q_1 primarily depend on its own previous value, with a smaller contribution from Q_3 . In contrast, the cofounder variable Q_3 has a stronger influence in the prediction of Q_2 as indicated by the orange bars. This idea can be justified by analyzing the evolution of the temporal signals, Figure \ref{fig:evo4}. In this figure, the strong self-dependency of the variable Q_1 is evidenced as the high frequency of the signal Q_3 produces relatively small perturbations on the previous state of Q_1 . However, the signal Q_2 presents a higher frequency which is mostly condition by the previous state of Q_3 . This idea also evidences the capacity of the SHAP values not only to determine the causality between variables but also the intensity of the cause-effect relationships. Finally, Q_3 only depends on its previous value.

Supplementary Figure 2: SHAP evaluation of the cofounder system. Mean SHAP value of the input variables Q_1^t , Q_2^t and Q_3^t for the model defined in equation (3), to predict their evolution Q_1^{t+1} in blue, Q_2^{t+1} in orange and Q_3^{t+1} in green.

Supplementary Figure 3: Temporal evolution of the cofounder variables. The temporal evolution is presented for 30 consecutive time steps for the variables Q_1 in blue, Q_2 in orange and Q_3 in green. The time sampling is visualized by the position of the markers.

In the collider system, one variable depends on the other two. Specifically, the value of the variable Q_1^{t+1} is determined by the states of Q_2 and Q_3 . The model is defined as follows:

$$\left. \begin{aligned} Q_1^{t+1} &= \sin(Q_2^t Q_3^t) + 0.001W_1^t \\ Q_2^{t+1} &= 0.5Q_2^t + 0.1W_2^t \\ Q_3^{t+1} &= 0.5Q_3^t + 0.1W_3^t \end{aligned} \right\} \quad (4)$$

The mean SHAP values presented in Figure \ref{fig:barshap5} indicate that Q_1 is equally influenced by Q_2 and Q_3 , while Q_2 and Q_3 depend solely on their own previous value.

Supplementary Figure 4: SHAP evaluation of the collider system. Mean SHAP value of the input variables Q_1^t , Q_2^t and Q_3^t for the model defined in equation (4), to predict their evolution Q_1^{t+1} in blue, Q_2^{t+1} in orange and Q_3^{t+1} in green.

Finally, the redundant collider system is presented. In this case, two variables are identical, with Q_2 and Q_3 representing the same variable. The variable Q_1 depends on both Q_2 and Q_3 , despite them being identical. As shown in Figure \ref{fig:barshap6}, the SHAP values cannot differentiate them.

$$\left. \begin{aligned} Q_1^{t+1} &= 0.3Q_1^t + \sin(Q_2^t Q_3^t) + 0.001W_1^t \\ Q_2^{t+1} &= 0.5Q_2^t + 0.1W_2^t \\ Q_3^{t+1} &= Q_2^{t+1} \end{aligned} \right\} \quad (5)$$

Additionally, the SHAP values reveal that the influence of the sinus on the prediction of Q_1 is stronger than the effect of its previous state. The influence of the sinus in the evolution of the variable Q_1 is visualized in Figure \ref{fig:evo6}, where its variation follows Q_2 and Q_3 .

Supplementary Figure 5: SHAP evaluation of the redundant collider system. Mean SHAP value of the input variables Q_1^t , Q_2^t and Q_3^t for the model defined in equation (5), to predict their evolution Q_1^{t+1} in blue, Q_2^{t+1} in orange and Q_3^{t+1} in green.

Supplementary Figure 6: Temporal evolution of the redundant collider variables. The temporal evolution is presented for 30 consecutive time steps for the variables Q_1 in blue, Q_2 in orange and Q_3 in green. The time sampling is visualized by the position of the markers.

SHAP values effectively capture the causal contribution of input variables in a dynamic system. Deep-learning models used to predict the evolution of a dynamic system establish causal relationships between inputs and outputs. Applying SHAP

values to these models reveals these relationships by identifying the variables that have the greatest influence on the evolution of the system's state. In the turbulent channel case, grid points with higher SHAP values are those that more strongly influence the prediction of the flow's next state—in other words, they play a greater causal role in its evolution.

Furthermore, the following modifications have been implemented to further justify the relevance of the proposed method:

- The SHAP values quantify the causal importance of each individual grid point in the evolution of the flow (refer to the supplementary material for a detailed discussion and a comparison with the causal system examples presented by \cite{martinez2024}). As previously mentioned, due to the evolution of the model, the SHAP values determine which grid points are the most influential for the evolution of the flow. Accordingly, the SHAP structures identify the regions of the flow that should be targeted to effectively control its evolution~\cite{beneitez2025improving}.
- Thus, the SHAP structures can identify the Q events with higher impact for the evolution of the flow, or in other words, the most causal Q events.
- Since the SHAP structures objectively identify the most important regions of the flow, analyzing the differences between SHAP and other structures can provide an excellent venue to deepen our insight into wall-bounded turbulence, uncovering the most important causal relations in the system.
- The definition of causally important regions is a change of paradigm in the analysis of physical problems, and can be replicated in many other fields such as thermal fields or cavitation. Furthermore, it can help to develop much more efficient control strategies~\cite{beneitez2025improving}.

Additionally, the friction Reynolds number has been increased up to a value of 550, ensuring the separation of scales.

Regarding Figure 3, we present the probability of encountering a streamwise velocity fluctuation across different types of coherent structures, conditioned on the wall-normal distance. This plot encompasses all scales of velocity fluctuations detected within the structures, thereby capturing the full range of turbulent flow scales. Although we understand the doubts that might have arisen in the case of $Re_\tau = 125$ where the SHAP structures mainly appeared for high values of the streamwise velocity, the agreement with the results of $Re_\tau = 550$ demonstrate the capacity of the methodology to detect high-importance regions. This idea can be inferred from the modification of Figure 3. As shown below, although the Q events exhibit high probability of containing high streamwise velocity regions, the SHAP structures do not highlight these regions as important. Therefore, SHAP values not only detect velocity fluctuation scales, but they highlight the regions with a higher causal contribution for the future evolution of the flow. Our answer to the previous comment provides more details on the results at $Re_\tau = 550$.

Figure: Joint probability density function of the SHAP structures (left) and Q events (right) for a $Re_\tau = 550$.

I cannot address whether the new algorithm is superior to the existing structure detection method.

The methodology proposed in this paper offers several advantages over existing structure-detection methods. Explainable-machine-learning models inherently capture causal correlations. Therefore, when analyzing SHAP values, we are not merely examining an instantaneous flow magnitude (such as Reynolds stress in Q events) but rather identifying the regions that have the greatest influence on the future evolution of the velocity field.

This approach is crucial for turbulent-flow control, as effective control strategies require anticipating the flow's evolution to maximize the impact of control actions. In fact, we would like to add here the reference of a preprint paper in which we used a SHAP-based control strategy which outperformed the reduction of skin friction with other traditional methods:

Beneitez, M., Cremades, A., Guastoni, L., & Vinuesa, R. (2025). Improving turbulence control through explainable deep learning. *arXiv preprint arXiv:2504.02354*.

In order to highlight this idea in the main text, we have added the following sentence:

The definition of causally important regions is a change of paradigm in the analysis of physical problems, and can be replicated in many other fields such as thermal fields or cavitation. Furthermore, it can help to develop much more efficient control strategies~\cite{beneitez2025improving}.

Consequently, the SHAP framework not only identifies completely new structures purely based on importance, but can also guide control and optimization strategies by using the SHAP field as a reward to minimize/maximize.

At least Figure 3(a) represents just the sum of the u-streak structure near the wall ($y^+ \approx 12$), and above the Reynolds stress ($u'v'$) structures, then it is hardly said that the ML technique uncovers new turbulence structures even at this low rate of Reynolds number.

In fact, while the SHAP structures in Figure 3 may appear statistically similar to the sum of Q events and streaks, a closer examination of their instantaneous overlap in Figure 4 reveals significant differences. This analysis demonstrates that SHAP structures are not merely a combination of Q events and streaks, although some resemblance exists. In addition, the new case of application, at a higher Reynolds number, evidences larger differences between the SHAP structures, the Q events and the streaks. In the figures below, the joint probability density function of the streamwise velocity and the wall-normal distance for the SHAP structures (left), the Q events (center) and the streaks (right) for a turbulent channel at a friction Reynolds number $Re_\tau = 550$ is presented. In the figures the Q events show a high probability of high streamwise velocity regions at a wall-normal distance above 100 wall units. Nevertheless, this probability does not match the SHAP structures at the same wall-normal distance. Additionally, streaks and SHAP structures only match around $y^+ \approx 15$. Both ideas evidence that SHAP values do not represent Q events, streaks or both at the same time, but they represent those regions of the flow with higher causal implications, which might or might not match any other classical structure, as it has been presented in the supplementary material and it would be discussed later in this document. The modifications concerning the higher Reynolds number and the causal implication of the SHAP values were presented in the previous answers.

Figure: Joint probability density function of the streamwise velocity and the wall-normal distance for SHAP structures (left), Q events (center) and streaks (right).

In addition, an additional case of application is added to the paper, showing the identification of high-importance structures in the flow around a square wall-mounted obstacle. The following sentences have been added to the main text:

- An example of application of the present SHAP framework to another flow case, namely the flow around a wall-mounted square obstacle~\cite{martinez2023,yousif2023deep}, can be found in the Supplementary Material. In this case, the SHAP values highlight the influence of the wall and the obstacle, reducing the importance of the streaks $\backslash\text{rev}Rv\{$, identifying a similar importance of the Q events and a low importance of the vortices.

And the following section to the Supplementary Material:

SHAP structures in the flow around a square wall-mounted obstacle

In the present study, the SHAP structures have been calculated for a turbulent channel flow at two different friction Reynolds numbers: $\text{Re}_{\tau}=125$ and $\text{Re}_{\tau}=550$. However, in order to illustrate the adaptability of the present methodology to any type of flow, the SHAP structures in the flow around a square wall-mounted obstacle are also analyzed. The analyzed database, described in detail in Refs.~\cite{martinez2023,yousif2023deep}, was obtained by means of a direct numerical simulation (DNS) at a Reynolds number $\text{Re}_h = u_0 h/\nu = 2000$, where h is the height of the obstacle, u_0 the freestream velocity and ν the kinematic viscosity. We analyze a region of the domain on the leeward side of after the obstacle with size $2.86 h \times 2 h \times 1.24 h$. The obstacle has a cross-section $0.25h \times 0.25h$. The methodology of the main paper is reproduced for the obstacle flow, detecting the various coherent structures and evaluating their coincidence. It can be observed that, up to the obstacle height (for $y < h$), the agreement between SHAP structures and Q events is slightly lower than that at $\text{Re}_{\tau}=125$ (with a moderate increase in the shear layers at $y > h$). Regarding the streaks, the coincidence is much lower than in the case of the channel, since the main dynamics are within the wake, with the wall having a less important role. Finally, the vortices exhibit a low level of agreement with the SHAP structures as in the channel case. This example highlights that the SHAP framework can identify the most important features in different flows, regardless of the roles played by the classical structures.

Supplementary Figure 20: Coincidence of the various coherent structures in the flow around a square wall-mounted obstacle [10, 11]. Percentage of coincidence of pairs of the following structures: SHAP, Q events, streaks and vortices, relative to the volume of each type of the pair.

Below are specific minor comments: 1. How did the authors choose the time interval of $\Delta t^+ = 5$ for the high-importance region of SHAP, and what kind of physics is focused on within the short time interval? Considering the chaotic nature of turbulence, the highly important turbulence region should not be limited to such a short (viscous) time-interval projection, which may relate to the dissipation scale.

Following the reviewer's recommendation, we have extended the analysis to a time interval of $\Delta t^+ = 10$. As can be shown in the modifications below, the results are essentially the same as the ones with $\Delta t^+ = 5$, a fact that indicates that these importance-based structures exhibit robust properties and have high importance for the near-wall dynamics. The following text was added to the main paper:

- The Supplementary Material also presents results for a time interval $\Delta t^+ = 10$, showing high levels of agreement with the analysis for $\Delta t^+ = 5$.

And the following section was added to the Supplementary Material:

SHAP structures for longer time horizons

In this section, SHAP structures are analyzed at $Re_{\tau} = 125$ for a longer time interval between input and output: $\Delta t^+ = 10$. The corresponding joint probability density functions are shown in Supplementary Figure \ref{fig:sup_fig_dt10_1}. These distributions for $\Delta t^+ = 10$ exhibit a strong similarity with those obtained for $\Delta t^+ = 5$, as illustrated in Supplementary Figure \ref{fig:sup_fig_1}. As with the shorter time step, the SHAP analysis at $\Delta t^+ = 10$ identifies regions near the wall with high velocity, and regions farther from the wall with low velocity, as most influential. Notably, for $\Delta t^+ = 10$, high-importance ejections are absent from the channel center, indicating that increasing the time interval shifts the importance toward regions with maximal velocity fluctuations, specifically around $y^+ \approx 15$.

Supplementary Figure 18: Joint probability density function of the three velocity components of the SHAP structures for the different wall-normal distances for a time horizon $\Delta t^+ = 10$. The figure shows the distribution of the three velocity fluctuations, namely the streamwise, wall-normal, and spanwise components from left to right for $Re_\tau = 125$.

Regarding the overlap between SHAP structures and traditional coherent structures, the results are very similar to the ones obtained for $\Delta t^+ = 5$, with a small reduction in agreement, particularly with the streaks, which account for a smaller proportion of the SHAP values. This observation reinforces the interpretation that SHAP values isolate the most influential regions of the flow rather than simply reproducing known structures.

Supplementary Figure 19: Coincidence of the coherent structures for $\Delta t^+ = 10$. Percentage of coincidence of pairs of the following structures: SHAP, Q events, streaks and vortices, relative to the volume of each type of the pair for a turbulent channel at $Re_\tau = 125$.

Furthermore, one could consider longer time horizons and even other output functions (such as the skin friction), and these would probably yield different SHAP structures. This framework allows us to identify importance-based structures given a particular objective function. In this work this objective is the near-wall dynamics, but many other options are possible, and they will surely motivate future studies. The following text was added to the main text to reflect this point:

- Other possible models can be developed, e.g. for predictions with much longer time horizons (characteristic of the outer region) or of the wall-shear stress (where different SHAP structures may be identified). The different questions that can be formulated with various models will be addressed in future studies.

2. Another critical factor might be the choice of the U-net input in the SHAP algorithm. What will happen when we use the vorticity field as the input?

This is an interesting suggestion by the Reviewer. In fact, the proposed explainability framework enables answering questions regarding the most important underlying mechanisms for any input and output, as long as one can train a model yielding sufficiently good predictions. In this case, we created a model where the input and the output are the vorticity fluctuations, with the same time interval as that of the velocity fluctuations. The analysis reveals completely different physical phenomena, as discussed below. This can open the door to a wide range of possible analysis to identify novel mechanisms in high-order chaotic systems.

The following sentence was added to the discussion of the main paper:

In fact, the SHAP structures for the evolution of the vorticity are presented in the Supplementary Material.

Furthermore, the following modifications were added to the Supplementary Material:

The present analysis focuses on the evolution of vorticity in a turbulent channel flow at $\text{Re}_\tau = 125$. Here, vorticity fluctuations are temporally evolved using a U-net model, following the approach shown in Figure 1 of the main paper. SHAP values are then employed to quantify the importance of each grid point in the predictions of the model. Supplementary Figure \ref{fig:sup_fig_vor} presents the normalized joint probability density function (PDF) of the velocity fluctuation components and wall-normal distance within the SHAP-identified structures. These joint PDFs reveal that vorticity transport is primarily concentrated in ejection-like structures near the wall and sweep-like structures farther from the wall. The former are associated with vorticity generation due to wall friction and lift-off effects, while the latter contribute through the downward rotational motion of the flow toward the wall.

Supplementary Figure 20: Joint probability density function of the three velocity components of the SHAP structures for the evolution of the vorticity as a function of the wall-normal distance. The figure shows the distribution of the three velocity fluctuations, namely the streamwise, wall-normal, and spanwise components from left to right for $Re_\tau = 125$.

To investigate the instantaneous overlap between SHAP structures associated with vorticity and other coherent structures, a spatio-temporal coincidence analysis is performed. Supplementary Figure \ref{fig:sup_fig_coincvor} shows the volumetric overlap between the SHAP-identified vorticity structures and the other flow structures, including Q events, streaks, vortices, and SHAP structures for velocity. The SHAP structures for vorticity display a low level of agreement, with less than 20% volumetric coincidence with the other coherent structures. This result highlights the power of the explainable-deep-learning framework to identify different types of relevant phenomena in high-dimensional chaotic systems.

Supplementary Figure 21: Coincidence of the coherent structures. Percentage of coincidence of pairs of the following structures: SHAP for the vorticity, Q events, streaks, vortices and SHAP for the velocity, relative to the volume of each type of the pair for a turbulent channel at $Re_\tau = 125$.

3. What is the physical meaning of the SHAP vector, and how can we relate the squared-averaged vectors (eq.(1)) to the Reynolds-averaged Navier-Stokes equations?

The physical meaning of the SHAP vector depends on the specific question posed to the model. In this study, the model output is the mean-squared error of the velocity reconstruction at a future time step. Therefore, the SHAP values represent the error introduced when the information at a specific grid point is removed. In other words, they identify the most influential points in the causal relationships detected by the machine-learning model for the evolution of the velocity fluctuations. In order to clarify their meaning the following sentences has been added to the text:

- **These SHAP values identify the most influential points in the causal relationships detected by the deep-learning model. For this reason, the physical meaning of the SHAP values is conditioned by the output of the model. The SHAP values answer the question formulated mathematically for the deep-learning model. In the case presented in this work, the model calculates the mean-squared error of the reconstruction of the flow. Therefore, the SHAP values identify the most important regions of the flow for the evolution of the velocity fluctuation field.**
- **As previously mentioned, due to the evolution of the model, the SHAP values determine which grid points are the most influential for the evolution of the flow.**
- **The SHAP values are used to identify these high-importance regions by analyzing the most sensitive input features for a mathematically defined question: which regions minimize the error of the model predictions?**

4. I guess, authors means eq.(3): $|u| > \alpha u$.

Since we are working with a three-dimensional problem, we aim to account for all the turbulent kinetic energy parallel to the wall. For this reason, we define the streaks by also incorporating the spanwise component.

REVIEWER COMMENTS

Reviewer #1 (Remarks to the Author):

The paper illustrates a data-driven methodology for identifying high-importance regions in a turbulent flow, using the gradient SHAP method to estimate the importance of each grid point in a turbulent channel for the prediction of its future states. The idea is very well presented and the study compares its outcome against the usage of typical methods (Q events, streaks, and vortices) in the case of wall-bounded turbulence: this analysis is well detailed, very clear and it shows good agreement with established methods, which is satisfying since the deep-learning model does not include prior knowledge of any kind. **The only complaint here is the choice of colors in some plots that are not so contrasting.**

The work is a practical implementation of the ideas presented in their previous paper "Additive-feature attribution methods: a review on explainable artificial intelligence for fluid dynamics and heat transfer" and the novelty of this study lies in checking if the method can compare and supersede the known approaches commonly used in the field: this is hard to tell from a single comparison, but presented result is promising.

We would like to express our gratitude for the reviewer's kind words and constructive feedback. We also fully agree with the reviewer's perspective. While the comparison may be challenging, the primary aim of this paper is to emphasize the distinction between classical and data-driven approaches. This work serves as a starting point for the data-driven analysis of complex flows, and we intend to extend these results in the future by exploring higher Reynolds numbers and different flow characteristics.

We also agree with the comment regarding the color choices. We recognize that maximizing the contrast of the figures is crucial for effectively communicating new ideas. For this reason, we have made efforts to use a sequential colormap (Viridis) wherever possible to enhance figure readability, particularly for individuals with vision impairments. In certain cases, we changed the colormap to qualitative tab10, when it could improve the visualization. Below, we expose the selection of the colormaps for the different plots and we exemplify them.

In this sense, the sequential colormap Viridis is maintained in those figures presenting gradients, so they can be visualized in black and white and for people with vision impairments. These figures are the joint pdfs, the visualization of the structures and the visualization of fields:

For those figures that require discrete colors, such as the lines of coincidence, the bar figures and the visualization of coincidence between structures, the commonly used qualitative colormap `tab10` has been used. The idea behind this selection is improving the visualization of the different categories.

Some examples of how these plots change are presented below:

We hope that we could address this comment to the Reviewer's satisfaction.

Reviewer #1 (Remarks on code availability):

The code is working and well commented, presented with a docker that eases the installation.

We appreciate the kind words of the reviewer.

REVIEWER COMMENTS

Reviewer #2 (Remarks to the Author):

The authors introduce a method to calculate important regions in turbulent flows, where importance is defined using SHAP-based values of a surrogate model. The approach is applied to calculate relevant structures in a turbulent channel flow at a low Reynolds number and is compared with classical definitions of coherent structures, exhibiting high resemblance.

We appreciate the reviewer's feedback and recognize the value of the questions and suggestions provided in enhancing the work. We hope we could address the comments below to the reviewer's satisfaction.

- Although the application of the SHAP method for structure identification is very interesting, the conclusions drawn from it seem to be very similar to those from classical analyses. This is a valuable result. However, it would be more compelling to showcase the method in a situation where classical analysis fails, in order to extract new physical insights. In my opinion this is the main weakness of the work.

We appreciate this insightful comment. We would like to note that, despite some similarities preset in the JPDFs shown in Figure 3 from the original article at $Re_\tau = 125$, the actual agreement between SHAP and Q events was just around 60% (Figure 4 from the original article). The average agreement with vortices was below 20%, and the streaks only had a significant agreement at $y^+=15$, with that agreement quickly dropping elsewhere. This can also be observed in the figure below:

In the revised version of the manuscript we have extended the analysis to $Re_\tau = 550$, with the following levels of coincidence among the various structures:

Interestingly, the average agreement between SHAP and Q events is reduced to 50%, and the low agreement with the vortices is maintained. The streaks maintain good agreement in the near-wall region, which quickly declines elsewhere. This shows that the regions of importance identified with SHAP are different from the classically studied coherent structures, although there are some localized areas of higher agreement.

Following the reviewer's recommendation, we have applied the SHAP analysis to another case, namely the flow around a square wall-mounted obstacle. We expect the classically studied structures to play different roles in this flow due to the significantly diminished importance of the wall.

The following modification was added to the revised manuscript:

- To fill this gap, here we show a data-driven methodology for objectively identifying high-importance regions in a turbulent flow up to a friction Reynolds number of 550.
- Based on the gradient-SHAP methodology we determine an importance score for each grid point of two turbulent channels, the first one with a friction Reynolds number $Re_{\tau} = 125$ and the second one with $Re_{\tau} = 550$.
- Figure 2 has been modified:

Figure 2: Instantaneous visualization of the various coherent structures in the channel flow. The figure shows four types of coherent structures (SHAP-based structures, Reynolds-stress structures or Q events, streaks and vortices from top to bottom) in half of the channel colored by wall-normal distance (purple near the wall and yellow in the mid-plane of the channel) for $Re_\tau = 125$ (left) and $Re_\tau = 550$ (right). In the figure, the wall is located at $y^+ = 0$ and the mid-plane of the channel is $y^+ = 125$ or $y^+ = 550$ depending on the case. In addition, we use periodic boundary conditions in x and z for both cases. Note that the same flow field is used for the four panels of each channel.

- The structures are presented in the second row of Figure 2 and are defined as:
- where α is the percolation parameter. The streamwise streaks are shown in the third row of Figure 2.
- The visualization of the vortices is provided in the bottom row of Figure 2.
- Figure 3 has been modified:

Figure 3: Joint probability density function of the streamwise velocity fluctuation and the inner-scaled wall-normal distance for the different types of structures. The figure presents the SHAP-based structures, Reynolds-stress structures, streaks and vortices from top to bottom for $Re_\tau = 125$ (left) and $Re_\tau = 550$ (right). The histogram distribution contours are defined in the logarithmic scale, representing the most likely areas with yellow colors and the most unlikely ones with purple regions.

- Figure \ref{fig:fig_3} shows that the SHAP-based structures exhibit several regions of high importance for both friction Reynolds numbers.
- As presented in Figure \ref{fig:fig_3}, the high-importance high-velocity region close to the wall is similar for both friction Reynolds numbers, despite the

higher separation of scales of $\text{Re}_\tau=550$ with respect to $\text{Re}_\tau=125$. In fact, our results show that the sweep events close to the wall constitute high-importance regions independently of the friction Reynolds number, as they transport the high-energy regions towards the wall, where the shear stress is stronger.

- Although the high-importance regions are similar for SHAP structures and Q events at $\text{Re}_\tau = 125$, larger differences are observed for $\text{Re}_\tau = 550$ due to the separation of scales. While the SHAP structures only detect a region of high importance for low-velocity events, the Q events increase the probability of sweeps.
- Note that sweeps and ejections have been widely studied in the literature as very important structures, and therefore the agreement, as well as the differences, with SHAP structures are a very important result.
- Therefore, the regions involving energy transfer between the wall-attached and wall-detached regions have been identified as SHAP structures with high importance for both friction Reynolds numbers.
- Therefore, SHAP values detect complex nonlinear patterns that were not identified by the other definitions and approaches, as could be clearly observed with the Q events far from the wall for $\text{Re}_\tau=550$.
- Figure 4 has been modified:

Figure 4: Coincidence between the various types of structures. Percentage of coincidence of pairs of the following structures: SHAP, Q events, streaks and vortices, relative to the volume of each type of the pair for $\text{Re}_\tau = 125$ (top) and $\text{Re}_\tau = 550$ (bottom).

- This co-occurrence represents around 60% of the volume of the SHAP structures, and around 40% of the volume of the intense Q events for $\text{Re}_\tau = 125$ and around 50% and 30% respectively for $\text{Re}_\tau = 550$. This coexistence can be mainly observed in the blue and green regions inside the black contours for the various wall-normal locations shown in Figure

\ref{fig:fig_5}. Thus, a significant fraction of the SHAP structures coincides with Q events throughout the channel for both friction Reynolds numbers, but specially for $\text{Re}_\tau = 125$. However, there is a non-negligible percentage, 40% and 50% for each friction Reynolds number respectively of the SHAP structures, which is not correlated with the Reynolds stress.

- Regarding the comparison between SHAP and streaks, for $\text{Re}_\tau \approx 125$ the SHAP structures are contained inside the streaks, with almost 90% agreement in both cases, as can be observed for $\text{Re}_\tau \approx 125$, where the black contours of the SHAP structures remain inside the large orange or green streaks. Finally, the agreement between SHAP and vortices is lower and remains below 20% in both cases. Moreover, part of this coincidence is shared with the streaks and the Q events: as indicated by the purple, brown and gray colors inside the black contours.
- In the present manuscript, we have presented the capabilities of explainable artificial intelligence (XAI) to estimate the importance of each grid point in a turbulent channel up to a friction Reynolds number $\text{Re}_\tau = 550$ for the prediction of its future states.
- Finally, farther from the wall, the presence of the streaks is lower and the SHAP structures mainly correlate with the ejections, although for $\text{Re}_\tau = 550$ the separation of scales is higher and sweeps are also present.
- An example of application of the present SHAP framework to another flow case, namely the flow around a wall-mounted square obstacle~\cite{martinez2023,yousif2023deep}, can be found in the Supplementary Material. In this case, the SHAP values highlight the influence of the wall and the obstacle, reducing the importance of the streaks \revRv{, identifying a similar importance of the Q events and a low importance of the vortices.
- In addition, the methodology has been proved to be scalable with the friction Reynolds number, producing results directly related to the wall-normal distance independently of the Reynolds number.
- Constant grid spacings of approximately 8.2 and 4.1 viscous units are used in the streamwise and spanwise directions, respectively for the $\text{Re}_\tau=125$ case, and 8.9 and 4.3 for the $\text{Re}_\tau=550$ case.
- A turbulent channel with dimensions $8\pi h \times 2h \times 3\pi h$ in the streamwise, wall-normal and spanwise dimensions is simulated for $\text{Re}_\tau=125$ and $2\pi h \times 2h \times \pi h$ for $\text{Re}_\tau=550$, where h is half the distance between the channel walls.
- The previous grid spacing and channel size generate a mesh containing $384 \times 201 \times 288$ points in the streamwise, wall-normal and spanwise directions for $\text{Re}_\tau=125$ and $384 \times 251 \times 384$ for $\text{Re}_\tau=550$, which matches the initial and final sizes of the data used in the U-net presented in Figure~\ref{fig:fig_1}.
- The architecture comprises four layers in the case of $\text{Re}_\tau=125$, where the first layer has a size of $201 \times 288 \times 384$ points, which is padded into the shape $201 \times 318 \times 414$ (blue arrow) in the first

operation, exploiting the periodicity of the channel to avoid any possible error in the edges of the convolution, and cropped back (light blue arrows) in the last operation.

- For the $\text{Re}_\tau=550$ case the number of filters is adjusted to 24, 48, 96 and 192 depending on the layer and the sizes of the tensors are modified according to the size of the input field.
- The U-net is trained on a database comprising a total of 10000 instantaneous flow fields of a turbulent channel at a friction Reynolds number of $\text{Re}_\tau = 125$ and $\text{Re}_\tau = 550$ respectively, using 80% of them for training and 20% for validation until the relative error between the predicted output and the ground truth is approximately 1% in the three velocity components (1.15% in the streamwise, 0.98% in the wall-normal, and 1.08% in the spanwise velocity fluctuations for $\text{Re}_\tau = 125$ and 0.74%, 1.15% and 0.88% for $\text{Re}_\tau = 550$).

And the following section was added to the Supplementary Material:

- Supplementary Figure 9 has been modified:

Supplementary Figure 9: Joint probability density function of the three velocity components of the SHAP structures for the different wall-normal distances. The figure shows the distribution of the three velocity fluctuations, namely the streamwise, wall-normal, and spanwise components from left to right for $\text{Re}_\tau = 125$ (top) and $\text{Re}_\tau = 550$ (bottom).

- Note that the SHAP values exhibit similar distributions for both friction Reynolds numbers.
- Supplementary Figure 10 has been modified:

Supplementary Figure 10: Joint probability density function of the three velocity components of the intense Reynolds stress structures for the different wall-normal distances. The figure shows the distribution of the three velocity fluctuations, namely the streamwise, wall-normal, and spanwise components from left to right for $Re_\tau = 125$ (top) and $Re_\tau = 550$ (bottom).

- However, although in the case of $Re_\tau = 550$ a high probability of sweeps far from the wall is observed, these do not correspond to high-importance regions, which remain similar for both friction Reynolds numbers.
- The percentage of agreement between the SHAP structures and the intense Reynolds-stress structures, streaks and vortices was presented with a visualization of the structures at three different wall-normal distances in Figure 5 from the main paper. This section focuses on extending the visualization of the structures at different wall-normal distances for the whole channel size, Supplementary Figures 4, to Figure 4e.
- Supplementary Figure 11 has been modified:

Supplementary Figure 11: Joint probability density function of the three velocity components of the streaks for the different wall-normal distances. The figure shows the distribution of the three velocity fluctuations, namely the stream-wise, wall-normal, and spanwise components from left to right for $Re_\tau = 125$ (top) and $Re_\tau = 550$ (bottom).

- Supplementary Figure \ref{fig:sup_fig_1} shows that the important structures are located near $y^+ \approx 15$. This wall-normal distance matches the distribution of the streaks for both friction Reynolds numbers.
- Figure 12 has been modified:

Supplementary Figure 12: Joint probability density function of the three velocity components of the vortices for the different wall-normal distances. The figure shows the distribution of the three velocity fluctuations, namely the stream-wise, wall-normal, and spanwise components from left to right for $Re_\tau = 125$ (top) and $Re_\tau = 550$ (bottom).

- This fact agrees with the region of the joint probability density function of the SHAP structures (Supplementary Figure \ref{fig:sup_fig_1}), where they exhibit a non-negligible probability of structures in regions of low-velocity fluctuations for both friction Reynolds numbers.

- For a friction Reynolds number of $\text{Re}_\tau=125$, at $y^+\approx 3$ most of the SHAP structures are composed by intense Reynolds-stress structures or Q events, with a small presence of streaks and vortices, see Supplementary Figures \ref{fig:sup_fig_4} and \ref{fig:sup_fig_4b}.
- As the wall-normal distance is increased to $y^+\approx 6$, the streaks gain importance and there is a strong agreement between the SHAP structures and the regions in which Q events and streaks collide; note that this trend can also be observed for $y^+\approx 35$. Then, for a wall-normal distance $y^+\approx 13$ the SHAP structures are located inside the streaks, mostly where they match the Q events. For larger wall-normal distances, the SHAP structures are mostly located in regions of intense Reynolds stress, $y^+\approx 81$ and $y^+\approx 110$, and the vortices gain importance as the wall-normal distance increases.
- For a friction Reynolds number of $\text{Re}_\tau=550$, the coincidence between the coherent structures for different wall-normal distances is presented in Supplementary Figures \ref{fig:sup_fig_4c}, \ref{fig:sup_fig_4d} and \ref{fig:sup_fig_4e}. Near the wall, the intense Q events coincide with the SHAP structures. This visualization is consistent with the high-importance sweeps in Figure~3 of the main paper. As the wall-normal distance is increased, the Q events and the streaks collide and match the SHAP structures. Then, for a wall-normal distance $y^+\approx 13$, the streaks increase and the SHAP structures are included within them. After this point, the streaks become weaker and the SHAP structures are located in the regions in which the Q events and streaks are coincident. For $y^+>100$ the presence of Q events and vortices increases, matching the SHAP structures part of the Q events, in agreement with the ideas presented on the main text: the SHAP values are in better agreement with the ejections far from the wall despite the increased presence of sweeps.
- Supplementary Figure 13 has been modified:

Supplementary Figure 13: Instantaneous coincidence between Q events, streaks, and vortices for six wall-normal locations below $y^+ = 15$ for $Re_\tau = 125$. The colors used for the coincidence between structures follow this code: $Q_s \setminus (\text{streaks} \cup \text{vortices})$, $\text{streaks} \setminus (Q_s \cup \text{vortices})$, $(Q_s \cup \text{streaks}) \setminus \text{vortices}$, $\text{vortices} \setminus (Q_s \cup \text{streaks})$, $(Q_s \cup \text{vortices}) \setminus \text{streaks}$, $(\text{streaks} \cup \text{vortices}) \setminus Q_s$, $Q_s \cup \text{streaks} \cup \text{vortices}$. The SHAP structures are represented by the black solid lines. Note that the dashed lines indicate the domain used in Figure 5 of the main article.

- Supplementary Figure 14 has been modified:

Supplementary Figure 14: Instantaneous coincidence between Q events, streaks, and vortices for three wall-normal locations above $y^+ = 15$ for $Re_\tau = 125$. The colors used for the coincidence between structures follow this code: \blacksquare $Qs \setminus (\text{streaks} \cup \text{vortices})$, \blacksquare $\text{streaks} \setminus (Qs \cup \text{vortices})$, \blacksquare $(Qs \cup \text{streaks}) \setminus \text{vortices}$, \blacksquare $\text{vortices} \setminus (Qs \cup \text{streaks})$, \blacksquare $(Qs \cup \text{vortices}) \setminus \text{streaks}$, \blacksquare $(\text{streaks} \cup \text{vortices}) \setminus Qs$, \blacksquare $Qs \cup \text{streaks} \cup \text{vortices}$. The SHAP structures are represented by the black solid lines. Note that the dashed lines indicate the domain used in Figure 5 of the main article.

- Supplementary Figure 15 has been modified:

Supplementary Figure 15: Instantaneous coincidence between Q events, streaks, and vortices for six wall-normal locations below $y^+ = 15$ for $Re_\tau = 550$. The colors used for the coincidence between structures follow this code: $\blacksquare Qs \setminus (\text{streaks} \cup \text{vortices})$, $\blacksquare \text{streaks} \setminus (Qs \cup \text{vortices})$, $\blacksquare (Qs \cup \text{streaks}) \setminus \text{vortices}$, $\blacksquare \text{vortices} \setminus (Qs \cup \text{streaks})$, $\blacksquare (Qs \cup \text{vortices}) \setminus \text{streaks}$, $\blacksquare (\text{streaks} \cup \text{vortices}) \setminus Qs$, $\blacksquare Qs \cup \text{streaks} \cup \text{vortices}$. The SHAP structures are represented by the black solid lines. Note that the dashed lines indicate the domain used in Figure 5 of the main article.

- Supplementary Figure 16 has been modified:

Supplementary Figure 16: Instantaneous coincidence between Q events, streaks, and vortices for three wall-normal locations above $y^+ = 15$ for $Re_\tau = 550$. The colors used for the coincidence between structures follow this code: \blacksquare $Q_s \setminus (\text{streaks} \cup \text{vortices})$, \blacksquare $\text{streaks} \setminus (Q_s \cup \text{vortices})$, \blacksquare $(Q_s \cup \text{streaks}) \setminus \text{vortices}$, \blacksquare $\text{vortices} \setminus (Q_s \cup \text{streaks})$, \blacksquare $(Q_s \cup \text{vortices}) \setminus \text{streaks}$, \blacksquare $(\text{streaks} \cup \text{vortices}) \setminus Q_s$, \blacksquare $Q_s \cup \text{streaks} \cup \text{vortices}$. The SHAP structures are represented by the black solid lines. Note that the dashed lines indicate the domain used in Figure 5 of the main article.

- Supplementary Figure 17 has been modified:

Supplementary Figure 17: Instantaneous coincidence between Q events, streaks, and vortices for three wall-normal locations above $y^+ = 15$ for $\text{Re}_\tau = 550$. The colors used for the coincidence between structures follow this code: $Q_s \setminus (\text{streaks} \cup \text{vortices})$, $\text{streaks} \setminus (Q_s \cup \text{vortices})$, $(Q_s \cup \text{streaks}) \setminus \text{vortices}$, $\text{vortices} \setminus (Q_s \cup \text{streaks})$, $(Q_s \cup \text{vortices}) \setminus \text{streaks}$, $(\text{streaks} \cup \text{vortices}) \setminus Q_s$, $Q_s \cup \text{streaks} \cup \text{vortices}$. The SHAP structures are represented by the black solid lines. Note that the dashed lines indicate the domain used in Figure 5 of the main article.

- Section in the supplementary material:

SHAP structures in the flow around a square wall-mounted obstacle

In the present study, the SHAP structures have been calculated for a turbulent channel flow at two different friction Reynolds numbers: $\text{Re}_\tau = 125$ and $\text{Re}_\tau = 550$. However, in order to illustrate the adaptability of the present methodology to any type of flow, the SHAP structures in the flow around a square wall-mounted obstacle are also analyzed. The analyzed database, described in detail in Refs. \$\sim\$ cite{martinez2023,yousif2023deep}, was obtained by means of a direct

numerical simulation (DNS) at a Reynolds number $\text{Re}_h = u_0 h / \nu = 2000$, where h is the height of the obstacle, u_0 the freestream velocity and ν the kinematic viscosity. We analyze a region of the domain on the leeward side of after the obstacle with size $2.86 h \times 2 h \times 1.24 h$. The obstacle has a cross-section $0.25 h \times 0.25 h$. The methodology of the main paper is reproduced for the obstacle flow, detecting the various coherent structures and evaluating their coincidence. It can be observed that, up to the obstacle height (for $y < h$), the agreement between SHAP structures and Q events is slightly lower than that at $\text{Re}_\tau = 125$ (with a moderate increase in the shear layers at $y > h$). Regarding the streaks, the coincidence is much lower than in the case of the channel, since the main dynamics are within the wake, with the wall having a less important role. Finally, the vortices exhibit a low level of agreement with the SHAP structures as in the channel case. This example highlights that the SHAP framework can identify the most important features in different flows, regardless of the roles played by the classical structures.

Supplementary Figure 20: Coincidence of the various coherent structures in the flow around a square wall-mounted obstacle [10, 11]. Percentage of coincidence of pairs of the following structures: SHAP, Q events, streaks and vortices, relative to the volume of each type of the pair.

We would like to note that both in the channel at higher Reynolds number and the wall-mounted obstacle, the SHAP leads to importance-based structures different from the classical ones, further strengthening its usage as a method to obtain insight into flow physics.

- The definition of 'important' used by the authors is based on SHAP values. However, there is no discussion about the conceptual meaning of this definition and why it is an advantageous choice compared to other definitions in the literature. For example, the definition of Q-events assumes that importance is based on wall-normal momentum transfer, which is critical for maintaining turbulence. I am not suggesting that this definition is better than the SHAP-based definition, but simply pointing out its rationale. It would be important to discuss the rationale behind using SHAP as a definition of importance.

We believe that this is a crucial point in the analysis, and we greatly appreciate the reviewer's suggestion. The central question in this paper is: *Which structures have the greatest influence on the evolution of the velocity field?* This question inherently shapes how the structures are defined. In this work, we use an explainable-deep-learning approach to

gain insight into flow evolution. The structures are defined based on their importance for the reconstruction of the flow. Additionally, we would like to provide a more in-depth discussion on why we consider the definition of SHAP structures to be particularly advantageous.

SHAP structures are based on the calculation of the SHapley Additive exPlanations (SHAP values, Lundberg and Lee 2017). These values are importance scores that highlight the most important regions of the flow for the prediction of the following time step. The SHAP values are based on the deep-learning model (surrogate model), which detects the causal implication of the input field for the evolution of the flow (output field). Thus, the SHAP values, as discussed in detail in our answer to the next comment (where we also added a new section in the Supplementary Material), detect those regions with a higher causal effect (Martínez-Sánchez et al., 2024).

Furthermore, we have recently compared different control strategies in turbulent channel flow aimed at reducing skin friction (Beneitez et al., 2025). In this work, we developed methods based on deep reinforcement learning aimed at minimizing the presence of Q events, streaks, and SHAP-based structures (among others). Interestingly, this work showed that the highest drag reduction is obtained precisely when the SHAP structures (which identify the most important causal connections in the flow) are minimized. The following modifications were added to the manuscript to further support this point:

- **The SHAP values quantify the causal importance of each individual grid point in the evolution of the flow (refer to the supplementary material for a detailed discussion and a comparison with the causal system examples presented by \cite{martinez2024}). As previously mentioned, due to the evolution of the model, the SHAP values determine which grid points are the most influential for the evolution of the flow. Accordingly, the SHAP structures identify the regions of the flow that should be targeted to effectively control its evolution~\cite{beneitez2025improving}.**
- **Thus, the SHAP structures can identify the Q events with higher impact for the evolution of the flow, or in other words, the most causal Q events.**
- **Since the SHAP structures objectively identify the most important regions of the flow, analyzing the differences between SHAP and other structures can provide an excellent venue to deepen our insight into wall-bounded turbulence, uncovering the most important causal relations in the system.**
- **The definition of causally important regions is a change of paradigm in the analysis of physical problems, and can be replicated in many other fields such as thermal fields or cavitation. Furthermore, it can help to develop much more efficient control strategies~\cite{beneitez2025improving}.**

Lundberg, S. and Lee, S. (2017). A unified approach to interpreting model predictions. arXiv preprint arXiv:1705.07874.

Martínez-Sánchez, Á., Arranz, G., & Lozano-Durán, A. (2024). Decomposing causality into its synergistic, unique, and redundant components. *Nature Communications*, 15(1), 9296.

Beneitez, M., Cremades, A., Guastoni, L., & Vinuesa, R. (2025). Improving turbulence control through explainable deep learning. *Preprint arXiv:2504.02354*.

- Related to the comment above, the explanation of SHAP values presented in the methodology does not provide much clarity on the conceptual meaning of the approach or how the method quantifies importance. It would be valuable to include simple examples and/or validation to enhance understanding.

We thank the Reviewer for this comment. We have added a section in the Supplementary Material with simple examples containing clearly identified causal relations, and we used the present SHAP framework to identify them. The new section is reproduced below:

Validation of the causal nature of the SHAP values

SHAP values are applied to the evolution of turbulent flows, identifying key regions that influence their development. The causal implications of the SHAP values are exploited, unveiling the cause and effect relationships inside the flow. In this section, we present a series of causal validation tests, adapted from \citet{martinez2024}, to assess the effectiveness of the explainable deep learning methodology. Each system consists of three variables, Q_1 , Q_2 and Q_3 , whose values depend on their states at the previous time steps and a noise level W_1 , W_2 and W_3 respectively:

$$\left. \begin{aligned} Q_1^{t+1} &= f_{Q_1} (Q_1^t, Q_2^t, Q_3^t) + g_{W_1} (W_1^t) \\ Q_2^{t+1} &= f_{Q_2} (Q_1^t, Q_2^t, Q_3^t) + g_{W_2} (W_2^t) \\ Q_3^{t+1} &= f_{Q_3} (Q_1^t, Q_2^t, Q_3^t) + g_{W_3} (W_3^t) \end{aligned} \right\} \quad (1)$$

Next, a deep-learning model, f , is trained to predict the evolution of the variables Q_1 , Q_2 and Q_3 : $\left[Q_1^{t+1}, Q_2^{t+1}, Q_3^{t+1} \right] = f \left(\left[Q_1^t, Q_2^t, Q_3^t \right] \right)$. The model f is a fully connected neural network with 9 hidden layers of 8 neurons each and every an output layer with 3 neurons. The model is trained on a database of 200,000 samples, with 80% used for training and 20% for testing. Finally, the SHAP values are applied to assess the influence of the variables Q_1 , Q_2 and Q_3 at time t on their evolution at time $t+1$. These values are computed over the test database and averaged to identify which input variable has the most significant impact on the predictions of the models.

The first case is a system with a mediator variable, where an intermediate variable transmits the information between the other two variables. In this system, the variable

Q_1^t depends on Q_2^t and Q_2^t depends on Q_3^t . The model is defined as follows:

$$\left. \begin{aligned} Q_1^{t+1} &= \sin(Q_2^t) + 0.001W_1^t \\ Q_2^{t+1} &= \cos(Q_3^t) + 0.01W_2^t \\ Q_3^{t+1} &= 0.5Q_3^t + 0.1W_3^t \end{aligned} \right\} \quad (2)$$

The mean SHAP values averaged over the test data, are presented in Supplementary Figure \ref{fig:barshap3}. In the figure, each color represents the contribution of the input variables Q_1^t , Q_2^t and Q_3^t to the evolution of the output variables: Q_1^{t+1} (blue), Q_2^{t+1} (orange) and Q_3^{t+1} (green). The SHAP values confirm that Q_2^t is the only variable influencing Q_1^{t+1} , while Q_3^t plays the most significant role in predicting both Q_2^{t+1} and Q_3^{t+1} , as defined by equation (\ref{eq:mediator}).

Supplementary Figure 1: SHAP evaluation of the mediator system. Mean SHAP value of the input variables Q_1^t , Q_2^t and Q_3^t for the model defined in equation (2), to predict their evolution Q_1^{t+1} in blue, Q_2^{t+1} in orange and Q_3^{t+1} in green.

The second case is a system with a cofounder variable. In this system, a single variable generates the other two, meaning that both variables Q_1 and Q_2 depend on Q_3 . The model is defined as follows:

$$\left. \begin{aligned} Q_1^{t+1} &= \sin(Q_1^t + Q_3^t) + 0.001W_1^t \\ Q_2^{t+1} &= \cos(Q_2^t - Q_3^t) + 0.001W_2^t \\ Q_3^{t+1} &= 0.5Q_3^t + 0.1W_3^t \end{aligned} \right\} \quad (3)$$

The results for the cofounder system, shown in Figure \ref{fig:barshap4}, demonstrate that the SHAP values can effectively capture the shared influence of Q_1 and Q_3 on the prediction of Q_1 . They also reflect the influence of Q_2 and Q_3 on the prediction of Q_2 . The predictions for Q_1 primarily depend on its own previous value, with a smaller contribution from Q_3 . In contrast, the cofounder variable Q_3 has a stronger influence in the prediction of Q_2 as indicated by the orange bars. This idea can be justified by analyzing the evolution of the temporal signals, Figure \ref{fig:evo4}. In this figure, the strong self-dependency of the variable Q_1 is evidenced as the high frequency of the signal Q_3 produces relatively small perturbations on the previous state of Q_1 . However, the signal Q_2 presents a higher frequency which is mostly condition by the previous state of Q_3 . This idea also evidences the capacity of the SHAP values not only to determine the causality between variables but also the intensity of the cause-effect relationships. Finally, Q_3 only depends on its previous value.

Supplementary Figure 2: SHAP evaluation of the cofounder system. Mean SHAP value of the input variables Q_1^t , Q_2^t and Q_3^t for the model defined in equation (3), to predict their evolution Q_1^{t+1} in blue, Q_2^{t+1} in orange and Q_3^{t+1} in green.

Supplementary Figure 3: Temporal evolution of the cofounder variables. The temporal evolution is presented for 30 consecutive time steps for the variables Q_1 in blue, Q_2 in orange and Q_3 in green. The time sampling is visualized by the position of the markers.

In the collider system, one variable depends on the other two. Specifically, the value of the variable Q_1^{t+1} is determined by the states of Q_2 and Q_3 . The model is defined as follows:

$$\left. \begin{aligned} Q_1^{t+1} &= \sin(Q_2^t Q_3^t) + 0.001W_1^t \\ Q_2^{t+1} &= 0.5Q_2^t + 0.1W_2^t \\ Q_3^{t+1} &= 0.5Q_3^t + 0.1W_3^t \end{aligned} \right\} \quad (4)$$

The mean SHAP values presented in Figure \ref{fig:barshap5} indicate that Q_1 is equally influenced by Q_2 and Q_3 , while Q_2 and Q_3 depend solely on their own previous value.

Supplementary Figure 4: SHAP evaluation of the collider system. Mean SHAP value of the input variables Q_1^t , Q_2^t and Q_3^t for the model defined in equation (4), to predict their evolution Q_1^{t+1} in blue, Q_2^{t+1} in orange and Q_3^{t+1} in green.

Finally, the redundant collider system is presented. In this case, two variables are identical, with Q_2 and Q_3 representing the same variable. The variable Q_1 depends on both Q_2 and Q_3 , despite them being identical. As shown in Figure \ref{fig:barshap6}, the SHAP values cannot differentiate them.

$$\left. \begin{aligned} Q_1^{t+1} &= 0.3Q_1^t + \sin(Q_2^t Q_3^t) + 0.001W_1^t \\ Q_2^{t+1} &= 0.5Q_2^t + 0.1W_2^t \\ Q_3^{t+1} &= Q_2^{t+1} \end{aligned} \right\} \quad (5)$$

Additionally, the SHAP values reveal that the influence of the sinus on the prediction of Q_1 is stronger than the effect of its previous state. The influence of the sinus in the evolution of the variable Q_1 is visualized in Figure \ref{fig:evo6}, where its variation follows Q_2 and Q_3 .

Supplementary Figure 5: SHAP evaluation of the redundant collider system. Mean SHAP value of the input variables Q_1^t , Q_2^t and Q_3^t for the model defined in equation (5), to predict their evolution Q_1^{t+1} in blue, Q_2^{t+1} in orange and Q_3^{t+1} in green.

Supplementary Figure 6: Temporal evolution of the redundant collider variables. The temporal evolution is presented for 30 consecutive time steps for the variables Q_1 in blue, Q_2 in orange and Q_3 in green. The time sampling is visualized by the position of the markers.

SHAP values effectively capture the causal contribution of input variables in a dynamic system. Deep-learning models used to predict the evolution of a dynamic system establish causal relationships between inputs and outputs. Applying SHAP values to these models reveals these relationships by identifying the variables that have the greatest influence on the evolution of the system's state. In the turbulent channel case, grid points with higher SHAP values are those that more strongly influence the prediction of the flow's next state—in other words, they play a greater causal role in its evolution.

- The paper might be too technical for a general audience. For example, the abstract describes Q-events, which, despite their importance in the field, are not familiar to many experts outside the wall-bounded turbulence community.

We thank the reviewer for pointing this out. To make the concepts more accessible, we have modified the abstract as follows:

- **Previous research has focused on analyzing the so-called coherent structures of the flow, or in other words, those regions of the flow intense enough in terms of energy production and transport, turbulent kinetic energy or rotation, to evolve on their own. However, the connection between these classically studied structures and the flow development is still uncertain. In a previous analysis, the importance of the different intense Reynolds stress structures was quantified through a data-driven methodology, showing that the calculated importance did not perfectly agree with the definition of the structures.**

In addition, in the main text we have also added explanations of the streaks and the vortices to clarify these concepts for a wider audience:

- **These are regions of high turbulent kinetic energy.**
- **In general, the vortices define those regions of the flow where rotation is larger than shear. Vortices appear close to the ejections \cite{loz14time} and contribute to the proposed near-wall cycle to sustain turbulence involving the streaks. In fact, vortices are a result of the instabilities of the streaks and can be considered dissipative as they carry high levels of enstrophy.**

- What is the role of the time scale in the U-Net used to predict the flow? The value chosen is 5-plus units, which is comparable to the shortest physical time scale in the flow. For such a short time scale, a model based on linearized Navier–Stokes equations might perform quite well. Either way, there is no guarantee that the important structures identified for 5-plus units are relevant for longer time scales. The work would be strengthened by an analysis of longer time lags.

Following the reviewer's recommendation, we have extended the analysis to a time interval of $\Delta t^+ = 10$. As can be shown in the modifications below, the results are essentially the same as the ones with $\Delta t^+ = 5$, a fact that indicates that these importance-based structures exhibit robust properties and have high importance for the near-wall dynamics. The following text was added to the main paper:

- The Supplementary Material also presents results for a time interval $\Delta t^+ = 10$, showing high levels of agreement with the analysis for $\Delta t^+ = 5$.

And a new section was added to the Supplementary Material:

SHAP structures for longer time horizons

In this section, SHAP structures are analyzed at $Re_{\tau} = 125$ for a longer time interval between input and output: $\Delta t^+ = 10$. The corresponding joint probability density functions are shown in Supplementary Figure \ref{fig:sup_fig_dt10_1}. These distributions for $\Delta t^+ = 10$ exhibit a strong similarity with those obtained for $\Delta t^+ = 5$, as illustrated in Supplementary Figure \ref{fig:sup_fig_1}. As with the shorter time step, the SHAP analysis at $\Delta t^+ = 10$ identifies regions near the wall with high velocity, and regions farther from the wall with low velocity, as most influential. Notably, for $\Delta t^+ = 10$, high-importance ejections are absent from the channel center, indicating that increasing the time interval shifts the importance toward regions with maximal velocity fluctuations, specifically around $y^+ \approx 15$.

Supplementary Figure 18: Joint probability density function of the three velocity components of the SHAP structures for the different wall-normal distances for a time horizon $\Delta t^+ = 10$. The figure shows the distribution of the three velocity fluctuations, namely the streamwise, wall-normal, and spanwise components from left to right for $Re_{\tau} = 125$.

Regarding the overlap between SHAP structures and traditional coherent structures, the results are very similar to the ones obtained for $\Delta t^+ = 5$, with a small reduction in agreement, particularly with the streaks, which account for a smaller proportion of the SHAP values. This observation reinforces the interpretation that SHAP values isolate the most influential regions of the flow rather than simply reproducing known structures.

Supplementary Figure 19: Coincidence of the coherent structures for $\Delta t^+ = 10$. Percentage of coincidence of pairs of the following structures: SHAP, Q events, streaks and vortices, relative to the volume of each type of the pair for a turbulent channel at $Re_\tau = 125$.

Furthermore, one could consider longer time horizons and even other output functions (such as the skin friction), and these would probably yield different SHAP structures. This framework allows us to identify importance-based structures given a particular objective function. In this work this objective is the near-wall dynamics, but many other options are possible, and they will surely motivate future studies. The following text was added to the main text to reflect this point:

- **Other possible models can be developed, e.g. for predictions with much longer time horizons (characteristic of the outer region) or of the wall-shear stress (where different SHAP structures may be identified). The different questions that can be formulated with various models will be addressed in future studies.**

- I found the choice of Reynolds number ($Re_\tau = 125$) curious, as the smallest Re_τ typically considered in the literature is $Re_\tau = 180$. I assume this is due to computational cost, but I am surprised that the flow does not laminarize at such a low value.

This is a very good point by the reviewer. We initially studied the flow at a low Reynolds number due to computational constraints, and the flow always remained turbulent partly thanks to the very large computational box we employed. We have significantly improved the efficiency of the algorithm, and have now been able to repeat the analysis at a significantly higher $Re_\tau = 550$. The conclusions are very similar to the ones at $Re_\tau = 125$, a fact that further supports the robustness and scalability of the analysis presented here.

The following text was added to the main paper:

- **To fill this gap, here we show a data-driven methodology for objectively identifying high-importance regions in a turbulent flow up to a friction Reynolds number of 550.**
- **Based on the gradient-SHAP methodology we determine an importance score for each grid point of two turbulent channels, the first one with a friction**

Reynolds number $Re_\tau = u_\tau h / \nu = 125$ and the second one with $Re_\tau = 550$.

- Figure 2 has been modified:

Figure 2: Instantaneous visualization of the various coherent structures in the channel flow. The figure shows four types of coherent structures (SHAP-based structures, Reynolds-stress structures or Q events, streaks and vortices from top to bottom) in half of the channel colored by wall-normal distance (purple near the wall and yellow in the mid-plane of the channel) for $Re_\tau = 125$ (left) and $Re_\tau = 550$ (right). In the figure, the wall is located at $y^+ = 0$ and the mid-plane of the channel is $y^+ = 125$ or $y^+ = 550$ depending on the case. In addition, we use periodic boundary conditions in x and z for both cases. Note that the same flow field is used for the four panels of each channel.

- The structures are presented in the second row of Figure 2 and are defined as:
- where α is the percolation parameter. The streamwise streaks are shown in the third row of Figure 2.
- The visualization of the vortices is provided in the bottom row of Figure 2.
- Figure 3 has been modified:

Figure 3: Joint probability density function of the streamwise velocity fluctuation and the inner-scaled wall-normal distance for the different types of structures. The figure presents the SHAP-based structures, Reynolds-stress structures, streaks and vortices from top to bottom for $Re_\tau = 125$ (left) and $Re_\tau = 550$ (right). The histogram distribution contours are defined in the logarithmic scale, representing the most likely areas with yellow colors and the most unlikely ones with purple regions.

- Figure \ref{fig:fig_3} shows that the SHAP-based structures exhibit several regions of high importance for both friction Reynolds numbers.
- As presented in Figure \ref{fig:fig_3}, the high-importance high-velocity region close to the wall is similar for both friction Reynolds numbers, despite the

higher separation of scales of $\text{Re}_\tau=550$ with respect to $\text{Re}_\tau=125$. In fact, our results show that the sweep events close to the wall constitute high-importance regions independently of the friction Reynolds number, as they transport the high-energy regions towards the wall, where the shear stress is stronger.

- Although the high-importance regions are similar for SHAP structures and Q events at $\text{Re}_\tau = 125$, larger differences are observed for $\text{Re}_\tau = 550$ due to the separation of scales. While the SHAP structures only detect a region of high importance for low-velocity events, the Q events increase the probability of sweeps.
- Note that sweeps and ejections have been widely studied in the literature as very important structures, and therefore the agreement, as well as the differences, with SHAP structures are a very important result.
- Therefore, the regions involving energy transfer between the wall-attached and wall-detached regions have been identified as SHAP structures with high importance for both friction Reynolds numbers.
- Therefore, SHAP values detect complex nonlinear patterns that were not identified by the other definitions and approaches, as could be clearly observed with the Q events far from the wall for $\text{Re}_\tau=550$.
- Figure 4 has been modified:

Figure 4: Coincidence between the various types of structures. Percentage of coincidence of pairs of the following structures: SHAP, Q events, streaks and vortices, relative to the volume of each type of the pair for $\text{Re}_\tau = 125$ (top) and $\text{Re}_\tau = 550$ (bottom).

- This co-occurrence represents around 60% of the volume of the SHAP structures, and around 40% of the volume of the intense Q events for $\text{Re}_\tau = 125$ and around 50% and 30% respectively for $\text{Re}_\tau = 550$. This coexistence can be mainly observed in the blue and green regions inside the black contours for the various wall-normal locations shown in Figure

\ref{fig:fig_5}. Thus, a significant fraction of the SHAP structures coincides with Q events throughout the channel for both friction Reynolds numbers, but specially for $\text{Re}_\tau = 125$. However, there is a non-negligible percentage, 40% and 50% for each friction Reynolds number respectively of the SHAP structures, which is not correlated with the Reynolds stress.

- Regarding the comparison between SHAP and streaks, for $10 \lesssim y^+ \lesssim 50$ the SHAP structures are contained inside the streaks, with almost 90% agreement in both cases, as can be observed for $y^+ \lesssim 13$, where the black contours of the SHAP structures remain inside the large orange or green streaks. Finally, the agreement between SHAP and vortices is lower and remains below 20% in both cases. Moreover, part of this coincidence is shared with the streaks and the Q events: as indicated by the purple, brown and gray colors inside the black contours.
- In the present manuscript, we have presented the capabilities of explainable artificial intelligence (XAI) to estimate the importance of each grid point in a turbulent channel \revo{up to a friction Reynolds number $\text{Re}_\tau = 550$ } for the prediction of its future states.
- Finally, farther from the wall, the presence of the streaks is lower and the SHAP structures mainly correlate with the ejections, although for $\text{Re}_\tau = 550$ the separation of scales is higher and sweeps are also present.
- In addition, the methodology has been proved to be scalable with the friction Reynolds number, producing results directly related to the wall-normal distance independently of the Reynolds number.
- Constant grid spacings of approximately 8.2 and 4.1 viscous units are used in the streamwise and spanwise directions, respectively for the $\text{Re}_\tau = 125$ case, and 8.9 and 4.3 for the $\text{Re}_\tau = 550$ case.
- A turbulent channel with dimensions $8\pi h \times 2h \times 3\pi h$ in the streamwise, wall-normal and spanwise dimensions is simulated for $\text{Re}_\tau = 125$ and $2\pi h \times 2h \times \pi h$ for $\text{Re}_\tau = 550$, where h is half the distance between the channel walls.
- The previous grid spacing and channel size generate a mesh containing $384 \times 201 \times 288$ points in the streamwise, wall-normal and spanwise directions for $\text{Re}_\tau = 125$ and $384 \times 251 \times 384$ for $\text{Re}_\tau = 550$, which matches the initial and final sizes of the data used in the U-net presented in Figure~\ref{fig:fig_1}.
- The architecture comprises four layers in the case of $\text{Re}_\tau = 125$, where the first layer has a size of $201 \times 288 \times 384$ points, which is padded into the shape $201 \times 318 \times 414$ (blue arrow) in the first operation, exploiting the periodicity of the channel to avoid any possible error in the edges of the convolution, and cropped back (light blue arrows) in the last operation.
- For the $\text{Re}_\tau = 550$ case the number of filters is adjusted to 24, 48, 96 and 192 depending on the layer and the sizes of the tensors are modified according to the size of the input field.

- The U-net is trained on a database comprising a total of 10000 instantaneous flow fields of a turbulent channel at a friction Reynolds number of $\text{Re}_\tau = 125$ and $\text{Re}_\tau = 550$ respectively, using 80% of them for training and 20% for validation until the relative error between the predicted output and the ground truth is approximately 1% in the three velocity components (1.15% in the streamwise, 0.98% in the wall-normal, and 1.08% in the spanwise velocity fluctuations for $\text{Re}_\tau = 125$ and 0.74%, 1.15% and 0.88% for $\text{Re}_\tau = 550$).

And the following section was added to the Supplementary Material:

- Supplementary Figure 9 has been modified:

Supplementary Figure 9: Joint probability density function of the three velocity components of the SHAP structures for the different wall-normal distances. The figure shows the distribution of the three velocity fluctuations, namely the streamwise, wall-normal, and spanwise components from left to right for $\text{Re}_\tau = 125$ (top) and $\text{Re}_\tau = 550$ (bottom).

- Note that the SHAP values exhibit similar distributions for both friction Reynolds numbers.
- Supplementary Figure 10 has been modified:

Supplementary Figure 10: Joint probability density function of the three velocity components of the intense Reynolds stress structures for the different wall-normal distances. The figure shows the distribution of the three velocity fluctuations, namely the streamwise, wall-normal, and spanwise components from left to right for $Re_\tau = 125$ (top) and $Re_\tau = 550$ (bottom).

- However, although in the case of $Re_\tau = 550$ a high probability of sweeps far from the wall is observed, these do not correspond to high-importance regions, which remain similar for both friction Reynolds numbers.
- The percentage of agreement between the SHAP structures and the intense Reynolds-stress structures, streaks and vortices was presented with a visualization of the structures at three different wall-normal distances in Figure 5 from the main paper. This section focuses on extending the visualization of the structures at different wall-normal distances for the whole channel size, Supplementary Figures 4, to Figure 4e.
- Supplementary Figure 11 has been modified:

Supplementary Figure 11: Joint probability density function of the three velocity components of the streaks for the different wall-normal distances. The figure shows the distribution of the three velocity fluctuations, namely the stream-wise, wall-normal, and spanwise components from left to right for $Re_\tau = 125$ (top) and $Re_\tau = 550$ (bottom).

- Supplementary Figure \ref{fig:sup_fig_1} shows that the important structures are located near $y^+ \approx 15$. This wall-normal distance matches the distribution of the streaks for both friction Reynolds numbers.
- Figure 12 has been modified:

Supplementary Figure 12: Joint probability density function of the three velocity components of the vortices for the different wall-normal distances. The figure shows the distribution of the three velocity fluctuations, namely the stream-wise, wall-normal, and spanwise components from left to right for $Re_\tau = 125$ (top) and $Re_\tau = 550$ (bottom).

- This fact agrees with the region of the joint probability density function of the SHAP structures (Supplementary Figure \ref{fig:sup_fig_1}), where they exhibit a non-negligible probability of structures in regions of low-velocity fluctuations for both friction Reynolds numbers.

- For a friction Reynolds number of $\text{Re}_\tau=125$, at $y^+\approx 3$ most of the SHAP structures are composed by intense Reynolds-stress structures or Q events, with a small presence of streaks and vortices, see Supplementary Figures \ref{fig:sup_fig_4} and \ref{fig:sup_fig_4b}.
- As the wall-normal distance is increased to $y^+\approx 6$, the streaks gain importance and there is a strong agreement between the SHAP structures and the regions in which Q events and streaks collide; note that this trend can also be observed for $y^+\approx 35$. Then, for a wall-normal distance $y^+\approx 13$ the SHAP structures are located inside the streaks, mostly where they match the Q events. For larger wall-normal distances, the SHAP structures are mostly located in regions of intense Reynolds stress, $y^+\approx 81$ and $y^+\approx 110$, and the vortices gain importance as the wall-normal distance increases.
- For a friction Reynolds number of $\text{Re}_\tau=550$, the coincidence between the coherent structures for different wall-normal distances is presented in Supplementary Figures \ref{fig:sup_fig_4c}, \ref{fig:sup_fig_4d} and \ref{fig:sup_fig_4e}. Near the wall, the intense Q events coincide with the SHAP structures. This visualization is consistent with the high-importance sweeps in Figure~3 of the main paper. As the wall-normal distance is increased, the Q events and the streaks collide and match the SHAP structures. Then, for a wall-normal distance $y^+\approx 13$, the streaks increase and the SHAP structures are included within them. After this point, the streaks become weaker and the SHAP structures are located in the regions in which the Q events and streaks are coincident. For $y^+>100$ the presence of Q events and vortices increases, matching the SHAP structures part of the Q events, in agreement with the ideas presented on the main text: the SHAP values are in better agreement with the ejections far from the wall despite the increased presence of sweeps.
- Supplementary Figure 13 has been modified:

Supplementary Figure 13: Instantaneous coincidence between Q events, streaks, and vortices for six wall-normal locations below $y^+ = 15$ for $Re_\tau = 125$. The colors used for the coincidence between structures follow this code: $Q_s \setminus (\text{streaks} \cup \text{vortices})$, $\text{streaks} \setminus (Q_s \cup \text{vortices})$, $(Q_s \cup \text{streaks}) \setminus \text{vortices}$, $\text{vortices} \setminus (Q_s \cup \text{streaks})$, $(Q_s \cup \text{vortices}) \setminus \text{streaks}$, $(\text{streaks} \cup \text{vortices}) \setminus Q_s$, $Q_s \cup \text{streaks} \cup \text{vortices}$. The SHAP structures are represented by the black solid lines. Note that the dashed lines indicate the domain used in Figure 5 of the main article.

- Supplementary Figure 14 has been modified:

Supplementary Figure 14: Instantaneous coincidence between Q events, streaks, and vortices for three wall-normal locations above $y^+ = 15$ for $Re_\tau = 125$. The colors used for the coincidence between structures follow this code: \blacksquare $Qs \setminus (\text{streaks} \cup \text{vortices})$, \blacksquare $\text{streaks} \setminus (Qs \cup \text{vortices})$, \blacksquare $(Qs \cup \text{streaks}) \setminus \text{vortices}$, \blacksquare $\text{vortices} \setminus (Qs \cup \text{streaks})$, \blacksquare $(Qs \cup \text{vortices}) \setminus \text{streaks}$, \blacksquare $(\text{streaks} \cup \text{vortices}) \setminus Qs$, \blacksquare $Qs \cup \text{streaks} \cup \text{vortices}$. The SHAP structures are represented by the black solid lines. Note that the dashed lines indicate the domain used in Figure 5 of the main article.

- Supplementary Figure 15 has been modified:

Supplementary Figure 15: Instantaneous coincidence between Q events, streaks, and vortices for six wall-normal locations below $y^+ = 15$ for $Re_\tau = 550$. The colors used for the coincidence between structures follow this code: $\blacksquare Qs \setminus (\text{streaks} \cup \text{vortices})$, $\blacksquare \text{streaks} \setminus (Qs \cup \text{vortices})$, $\blacksquare (Qs \cup \text{streaks}) \setminus \text{vortices}$, $\blacksquare \text{vortices} \setminus (Qs \cup \text{streaks})$, $\blacksquare (Qs \cup \text{vortices}) \setminus \text{streaks}$, $\blacksquare (\text{streaks} \cup \text{vortices}) \setminus Qs$, $\blacksquare Qs \cup \text{streaks} \cup \text{vortices}$. The SHAP structures are represented by the black solid lines. Note that the dashed lines indicate the domain used in Figure 5 of the main article.

- Supplementary Figure 16 has been modified:

Supplementary Figure 16: Instantaneous coincidence between Q events, streaks, and vortices for three wall-normal locations above $y^+ = 15$ for $Re_\tau = 550$. The colors used for the coincidence between structures follow this code: $\blacksquare Q_s \setminus (\text{streaks} \cup \text{vortices})$, $\blacksquare \text{streaks} \setminus (Q_s \cup \text{vortices})$, $\blacksquare (Q_s \cup \text{streaks}) \setminus \text{vortices}$, $\blacksquare \text{vortices} \setminus (Q_s \cup \text{streaks})$, $\blacksquare (Q_s \cup \text{vortices}) \setminus \text{streaks}$, $\blacksquare (\text{streaks} \cup \text{vortices}) \setminus Q_s$, $\blacksquare Q_s \cup \text{streaks} \cup \text{vortices}$. The SHAP structures are represented by the black solid lines. Note that the dashed lines indicate the domain used in Figure 5 of the main article.

- Supplementary Figure 17 has been modified:

Supplementary Figure 17: Instantaneous coincidence between Q events, streaks, and vortices for three wall-normal locations above $y^+ = 15$ for $Re_\tau = 550$. The colors used for the coincidence between structures follow this code: $Qs \setminus (\text{streaks} \cup \text{vortices})$, $\text{streaks} \setminus (Qs \cup \text{vortices})$, $(Qs \cup \text{streaks}) \setminus \text{vortices}$, $\text{vortices} \setminus (Qs \cup \text{streaks})$, $(Qs \cup \text{vortices}) \setminus \text{streaks}$, $(\text{streaks} \cup \text{vortices}) \setminus Qs$, $Qs \cup \text{streaks} \cup \text{vortices}$. The SHAP structures are represented by the black solid lines. Note that the dashed lines indicate the domain used in Figure 5 of the main article.

In fact, the following figure provides a direct comparison between both Reynolds numbers, showing the high degree of agreement between the SHAP distributions in both cases.

Figure: Joint probability density function of SHAP structures as a function of the wall-normal distance and the streamwise velocity fluctuation for the (dashed) $Re_\tau = 125$ and (solid) $Re_\tau = 550$ turbulent channel flows.

- The indirect connection between turbulence and climate change in the introduction feels tenuous. While turbulence is a key component, there are too many factors involved to make a decisive connection. This seems like a bit of a stretch. The topic of turbulence is important enough without the need for overselling it.

We understand the reviewer's point and have made the following modification to the introduction:

A complete understanding of turbulence may have worldwide implications, since 15% of the energy consumed worldwide is spent near the surface of vehicles due to turbulent effects~\cite{Jimenez2013}.

Furthermore, we implemented the following modification in the abstract to address this point:

The dissipation of energy due to turbulence is significant, and understanding turbulence physics has wide implications for energy efficiency.

Overall, I see merit in the paper and believe it is a meaningful contribution to the field of turbulence. Nonetheless, it is not entirely clear that it belongs in Nat. Comm. To be honest, I think it would have more impact in a specialized journal, where researchers can better appreciate the technicalities and relevance of the work.

We appreciate the reviewer's kind words and positive assessment of our work.

The decision of submitting the paper to Nature Communications was based on multiple reasons. The first reason is the fact that the present paper is the continuation of a study already published in Nature Communications:

Cremades, A., Hoyas, S., Deshpande, R., Quintero, P., Lellep, M., Lee, W. J., ... & Vinuesa, R. (2024). Identifying regions of importance in wall-bounded turbulence through explainable deep learning. *Nature Communications*, 15(1), 3864.

This article has had a large impact in 1 year: over 11,000 accesses, 35 citations and great online attention (97th percentile or all tracked articles of a similar age in all journals and 88th percentile in Nature Communications). Therefore, we believe that the readers of Nature Communications will be interested in the topic. Furthermore, Nature Communications can reach a broad audience, including researchers working on other complex and non-linear problems that could benefit from the advances presented in this paper. In order to reflect this idea in the main text of the paper, the following sentence has been added:

The definition of causally important regions is a change of paradigm in the analysis of physical problems, and can be replicated in many other fields such as thermal fields or cavitation. Furthermore, it can help to develop much more efficient control strategies~\cite{beneitez2025improving}.

The article by Beneitez et al. (2025) (also mentioned in a previous response) shows that using the methodology presented here to identify the most important structures in the flow can guide the most efficient control strategies to reduce turbulent drag. This may lead to a completely new paradigm in analysis and optimization of high-dimensional chaotic systems, with broad implications in a wide range of areas, further justifying the relevance of Nature Communications for this study.

Beneitez, M., Cremades, A., Guastoni, L., & Vinuesa, R. (2025). Improving turbulence control through explainable deep learning. *arXiv preprint arXiv:2504.02354*.

REVIEWER COMMENTS

Reviewer #3 (Remarks to the Author):

This paper introduces data-driven turbulence structure detection using the SHAP algorithm. They detected the highest importance in channel turbulence. However, since the channel flow dataset has a very low Reynolds number, $Re_\tau = 125$, the length scale separation between the large-scale motion and the smallest (vertical) one is insufficient.

First and foremost, we would like to thank the reviewer for their valuable feedback. This is a very good point by the reviewer. We initially studied the flow at a low Reynolds number due to computational constraints, and the flow always remained turbulent partly thanks to the very large computational box we employed. We have significantly improved the efficiency of the algorithm, and have now been able to repeat the analysis at a significantly higher $Re_\tau = 550$. The conclusions are very similar to the ones at $Re_\tau = 125$, a fact that further supports the robustness and scalability of the analysis presented here.

The following text was added to the main paper:

- To fill this gap, here we show a data-driven methodology for objectively identifying high-importance regions in a turbulent flow up to a friction Reynolds number of 550.
- Based on the gradient-SHAP methodology we determine an importance score for each grid point of two turbulent channels, the first one with a friction Reynolds number $Re_\tau = 125$ and the second one with $Re_\tau = 550$.
- Figure 2 has been modified:

Figure 2: Instantaneous visualization of the various coherent structures in the channel flow. The figure shows four types of coherent structures (SHAP-based structures, Reynolds-stress structures or Q events, streaks and vortices from top to bottom) in half of the channel colored by wall-normal distance (purple near the wall and yellow in the mid-plane of the channel) for $Re_\tau = 125$ (left) and $Re_\tau = 550$ (right). In the figure, the wall is located at $y^+ = 0$ and the mid-plane of the channel is $y^+ = 125$ or $y^+ = 550$ depending on the case. In addition, we use periodic boundary conditions in x and z for both cases. Note that the same flow field is used for the four panels of each channel.

- The structures are presented in the second row of Figure 2 and are defined as:
- where α is the percolation parameter. The streamwise streaks are shown in the third row of Figure 2.
- The visualization of the vortices is provided in the bottom row of Figure 2.
- Figure 3 has been modified:

Figure 3: Joint probability density function of the streamwise velocity fluctuation and the inner-scaled wall-normal distance for the different types of structures. The figure presents the SHAP-based structures, Reynolds-stress structures, streaks and vortices from top to bottom for $Re_\tau = 125$ (left) and $Re_\tau = 550$ (right). The histogram distribution contours are defined in the logarithmic scale, representing the most likely areas with yellow colors and the most unlikely ones with purple regions.

- Figure \ref{fig:fig_3} shows that the SHAP-based structures exhibit several regions of high importance for both friction Reynolds numbers.
- As presented in Figure \ref{fig:fig_3}, the high-importance high-velocity region close to the wall is similar for both friction Reynolds numbers, despite the

higher separation of scales of $\text{Re}_\tau=550$ with respect to $\text{Re}_\tau=125$. In fact, our results show that the sweep events close to the wall constitute high-importance regions independently of the friction Reynolds number, as they transport the high-energy regions towards the wall, where the shear stress is stronger.

- Although the high-importance regions are similar for SHAP structures and Q events at $\text{Re}_\tau = 125$, larger differences are observed for $\text{Re}_\tau = 550$ due to the separation of scales. While the SHAP structures only detect a region of high importance for low-velocity events, the Q events increase the probability of sweeps.
- Note that sweeps and ejections have been widely studied in the literature as very important structures, and therefore the agreement, as well as the differences, with SHAP structures are a very important result.
- Therefore, the regions involving energy transfer between the wall-attached and wall-detached regions have been identified as SHAP structures with high importance for both friction Reynolds numbers.
- Therefore, SHAP values detect complex nonlinear patterns that were not identified by the other definitions and approaches, as could be clearly observed with the Q events far from the wall for $\text{Re}_\tau=550$.
- Figure 4 has been modified:

Figure 4: Coincidence between the various types of structures. Percentage of coincidence of pairs of the following structures: SHAP, Q events, streaks and vortices, relative to the volume of each type of the pair for $\text{Re}_\tau = 125$ (top) and $\text{Re}_\tau = 550$ (bottom).

- This co-occurrence represents around 60% of the volume of the SHAP structures, and around 40% of the volume of the intense Q events for $\text{Re}_\tau = 125$ and around 50% and 30% respectively for $\text{Re}_\tau = 550$. This coexistence can be mainly observed in the blue and green regions inside the black contours for the various wall-normal locations shown in Figure

\ref{fig:fig_5}. Thus, a significant fraction of the SHAP structures coincides with Q events throughout the channel for both friction Reynolds numbers, but specially for $\text{Re}_\tau = 125$. However, there is a non-negligible percentage, 40% and 50% for each friction Reynolds number respectively of the SHAP structures, which is not correlated with the Reynolds stress.

- Regarding the comparison between SHAP and streaks, for $10 \lesssim y^+ \lesssim 50$ the SHAP structures are contained inside the streaks, with almost 90% agreement in both cases, as can be observed for $y^+ \lesssim 13$, where the black contours of the SHAP structures remain inside the large orange or green streaks. Finally, the agreement between SHAP and vortices is lower and remains below 20% in both cases. Moreover, part of this coincidence is shared with the streaks and the Q events: as indicated by the purple, brown and gray colors inside the black contours.
- In the present manuscript, we have presented the capabilities of explainable artificial intelligence (XAI) to estimate the importance of each grid point in a turbulent channel \revo{up to a friction Reynolds number $\text{Re}_\tau = 550$ } for the prediction of its future states.
- Finally, farther from the wall, the presence of the streaks is lower and the SHAP structures mainly correlate with the ejections, although for $\text{Re}_\tau = 550$ the separation of scales is higher and sweeps are also present.
- In addition, the methodology has been proved to be scalable with the friction Reynolds number, producing results directly related to the wall-normal distance independently of the Reynolds number.
- Constant grid spacings of approximately 8.2 and 4.1 viscous units are used in the streamwise and spanwise directions, respectively for the $\text{Re}_\tau = 125$ case, and 8.9 and 4.3 for the $\text{Re}_\tau = 550$ case.
- A turbulent channel with dimensions $8\pi h \times 2h \times 3\pi h$ in the streamwise, wall-normal and spanwise dimensions is simulated for $\text{Re}_\tau = 125$ and $2\pi h \times 2h \times \pi h$ for $\text{Re}_\tau = 550$, where h is half the distance between the channel walls.
- The previous grid spacing and channel size generate a mesh containing $384 \times 201 \times 288$ points in the streamwise, wall-normal and spanwise directions for $\text{Re}_\tau = 125$ and $384 \times 251 \times 384$ for $\text{Re}_\tau = 550$, which matches the initial and final sizes of the data used in the U-net presented in Figure~\ref{fig:fig_1}.
- The architecture comprises four layers in the case of $\text{Re}_\tau = 125$, where the first layer has a size of $201 \times 288 \times 384$ points, which is padded into the shape $201 \times 318 \times 414$ (blue arrow) in the first operation, exploiting the periodicity of the channel to avoid any possible error in the edges of the convolution, and cropped back (light blue arrows) in the last operation.
- For the $\text{Re}_\tau = 550$ case the number of filters is adjusted to 24, 48, 96 and 192 depending on the layer and the sizes of the tensors are modified according to the size of the input field.

- The U-net is trained on a database comprising a total of 10000 instantaneous flow fields of a turbulent channel at a friction Reynolds number of $\text{Re}_\tau = 125$ and $\text{Re}_\tau = 550$ respectively, using 80% of them for training and 20% for validation until the relative error between the predicted output and the ground truth is approximately 1% in the three velocity components (1.15% in the streamwise, 0.98% in the wall-normal, and 1.08% in the spanwise velocity fluctuations for $\text{Re}_\tau = 125$ and 0.74%, 1.15% and 0.88% for $\text{Re}_\tau = 550$).

And the following modifications were implemented in the Supplementary Material:

- Supplementary Figure 9 has been modified:

Supplementary Figure 9: Joint probability density function of the three velocity components of the SHAP structures for the different wall-normal distances. The figure shows the distribution of the three velocity fluctuations, namely the streamwise, wall-normal, and spanwise components from left to right for $\text{Re}_\tau = 125$ (top) and $\text{Re}_\tau = 550$ (bottom).

- Note that the SHAP values exhibit similar distributions for both friction Reynolds numbers.
- Supplementary Figure 10 has been modified:

Supplementary Figure 10: Joint probability density function of the three velocity components of the intense Reynolds stress structures for the different wall-normal distances. The figure shows the distribution of the three velocity fluctuations, namely the streamwise, wall-normal, and spanwise components from left to right for $Re_\tau = 125$ (top) and $Re_\tau = 550$ (bottom).

- However, although in the case of $Re_\tau = 550$ a high probability of sweeps far from the wall is observed, these do not correspond to high-importance regions, which remain similar for both friction Reynolds numbers.
- The percentage of agreement between the SHAP structures and the intense Reynolds-stress structures, streaks and vortices was presented with a visualization of the structures at three different wall-normal distances in Figure 5 from the main paper. This section focuses on extending the visualization of the structures at different wall-normal distances for the whole channel size, Supplementary Figures 4, to Figure 4e.
- Supplementary Figure 11 has been modified:

Supplementary Figure 11: Joint probability density function of the three velocity components of the streaks for the different wall-normal distances. The figure shows the distribution of the three velocity fluctuations, namely the stream-wise, wall-normal, and spanwise components from left to right for $Re_\tau = 125$ (top) and $Re_\tau = 550$ (bottom).

- Supplementary Figure \ref{fig:sup_fig_1} shows that the important structures are located near $y^+ \approx 15$. This wall-normal distance matches the distribution of the streaks for both friction Reynolds numbers.
- Figure 12 has been modified:

Supplementary Figure 12: Joint probability density function of the three velocity components of the vortices for the different wall-normal distances. The figure shows the distribution of the three velocity fluctuations, namely the stream-wise, wall-normal, and spanwise components from left to right for $Re_\tau = 125$ (top) and $Re_\tau = 550$ (bottom).

- This fact agrees with the region of the joint probability density function of the SHAP structures (Supplementary Figure \ref{fig:sup_fig_1}), where they exhibit a non-negligible probability of structures in regions of low-velocity fluctuations for both friction Reynolds numbers.

- For a friction Reynolds number of $\text{Re}_\tau=125$, at $y^+\approx 3$ most of the SHAP structures are composed by intense Reynolds-stress structures or Q events, with a small presence of streaks and vortices, see Supplementary Figures \ref{fig:sup_fig_4} and \ref{fig:sup_fig_4b}.
- As the wall-normal distance is increased to $y^+\approx 6$, the streaks gain importance and there is a strong agreement between the SHAP structures and the regions in which Q events and streaks collide; note that this trend can also be observed for $y^+\approx 35$. Then, for a wall-normal distance $y^+\approx 13$ the SHAP structures are located inside the streaks, mostly where they match the Q events. For larger wall-normal distances, the SHAP structures are mostly located in regions of intense Reynolds stress, $y^+\approx 81$ and $y^+\approx 110$, and the vortices gain importance as the wall-normal distance increases.
- For a friction Reynolds number of $\text{Re}_\tau=550$, the coincidence between the coherent structures for different wall-normal distances is presented in Supplementary Figures \ref{fig:sup_fig_4c}, \ref{fig:sup_fig_4d} and \ref{fig:sup_fig_4e}. Near the wall, the intense Q events coincide with the SHAP structures. This visualization is consistent with the high-importance sweeps in Figure~3 of the main paper. As the wall-normal distance is increased, the Q events and the streaks collide and match the SHAP structures. Then, for a wall-normal distance $y^+\approx 13$, the streaks increase and the SHAP structures are included within them. After this point, the streaks become weaker and the SHAP structures are located in the regions in which the Q events and streaks are coincident. For $y^+>100$ the presence of Q events and vortices increases, matching the SHAP structures part of the Q events, in agreement with the ideas presented on the main text: the SHAP values are in better agreement with the ejections far from the wall despite the increased presence of sweeps.
- Supplementary Figure 13 has been modified:

Supplementary Figure 13: Instantaneous coincidence between Q events, streaks, and vortices for six wall-normal locations below $y^+ = 15$ for $Re_\tau = 125$. The colors used for the coincidence between structures follow this code: $Q_s \setminus (\text{streaks} \cup \text{vortices})$, $\text{streaks} \setminus (Q_s \cup \text{vortices})$, $(Q_s \cup \text{streaks}) \setminus \text{vortices}$, $\text{vortices} \setminus (Q_s \cup \text{streaks})$, $(Q_s \cup \text{vortices}) \setminus \text{streaks}$, $(\text{streaks} \cup \text{vortices}) \setminus Q_s$, $Q_s \cup \text{streaks} \cup \text{vortices}$. The SHAP structures are represented by the black solid lines. Note that the dashed lines indicate the domain used in Figure 5 of the main article.

- Supplementary Figure 14 has been modified:

Supplementary Figure 14: Instantaneous coincidence between Q events, streaks, and vortices for three wall-normal locations above $y^+ = 15$ for $Re_\tau = 125$. The colors used for the coincidence between structures follow this code: $\blacksquare Q_s \setminus (\text{streaks} \cup \text{vortices})$, $\blacksquare \text{streaks} \setminus (Q_s \cup \text{vortices})$, $\blacksquare (Q_s \cup \text{streaks}) \setminus \text{vortices}$, $\blacksquare \text{vortices} \setminus (Q_s \cup \text{streaks})$, $\blacksquare (Q_s \cup \text{vortices}) \setminus \text{streaks}$, $\blacksquare (\text{streaks} \cup \text{vortices}) \setminus Q_s$, $\blacksquare Q_s \cup \text{streaks} \cup \text{vortices}$. The SHAP structures are represented by the black solid lines. Note that the dashed lines indicate the domain used in Figure 5 of the main article.

- Supplementary Figure 15 has been modified:

Supplementary Figure 15: Instantaneous coincidence between Q events, streaks, and vortices for six wall-normal locations below $y^+ = 15$ for $Re_\tau = 550$. The colors used for the coincidence between structures follow this code: $\blacksquare Qs \setminus (\text{streaks} \cup \text{vortices})$, $\blacksquare \text{streaks} \setminus (Qs \cup \text{vortices})$, $\blacksquare (Qs \cup \text{streaks}) \setminus \text{vortices}$, $\blacksquare \text{vortices} \setminus (Qs \cup \text{streaks})$, $\blacksquare (Qs \cup \text{vortices}) \setminus \text{streaks}$, $\blacksquare (\text{streaks} \cup \text{vortices}) \setminus Qs$, $\blacksquare Qs \cup \text{streaks} \cup \text{vortices}$. The SHAP structures are represented by the black solid lines. Note that the dashed lines indicate the domain used in Figure 5 of the main article.

- Supplementary Figure 16 has been modified:

Supplementary Figure 16: Instantaneous coincidence between Q events, streaks, and vortices for three wall-normal locations above $y^+ = 15$ for $Re_\tau = 550$. The colors used for the coincidence between structures follow this code: \blacksquare $Q_s \setminus (\text{streaks} \cup \text{vortices})$, \blacksquare $\text{streaks} \setminus (Q_s \cup \text{vortices})$, \blacksquare $(Q_s \cup \text{streaks}) \setminus \text{vortices}$, \blacksquare $\text{vortices} \setminus (Q_s \cup \text{streaks})$, \blacksquare $(Q_s \cup \text{vortices}) \setminus \text{streaks}$, \blacksquare $(\text{streaks} \cup \text{vortices}) \setminus Q_s$, \blacksquare $Q_s \cup \text{streaks} \cup \text{vortices}$. The SHAP structures are represented by the black solid lines. Note that the dashed lines indicate the domain used in Figure 5 of the main article.

- Supplementary Figure 17 has been modified:

Supplementary Figure 17: Instantaneous coincidence between Q events, streaks, and vortices for three wall-normal locations above $y^+ = 15$ for $Re_\tau = 550$. The colors used for the coincidence between structures follow this code: $Qs \setminus (\text{streaks} \cup \text{vortices})$, $\text{streaks} \setminus (Qs \cup \text{vortices})$, $(Qs \cup \text{streaks}) \setminus \text{vortices}$, $\text{vortices} \setminus (Qs \cup \text{streaks})$, $(Qs \cup \text{vortices}) \setminus \text{streaks}$, $(\text{streaks} \cup \text{vortices}) \setminus Qs$, $Qs \cup \text{streaks} \cup \text{vortices}$. The SHAP structures are represented by the black solid lines. Note that the dashed lines indicate the domain used in Figure 5 of the main article.

In fact, the following figure provides a direct comparison between both Reynolds numbers, showing the high degree of agreement between the SHAP distributions in both cases.

Figure: Joint probability density function of SHAP structures as a function of the wall-normal distance and the streamwise velocity fluctuation for the (dashed) $Re_\tau = 125$ and (solid) $Re_\tau = 550$ turbulent channel flows.

I cannot find the novelty of using the machine learning (ML) technique, which costs a lot for the learning process and is difficult to apply for higher-Re simulation datasets. Also, the main result of Figure 3 represents the SHAP algorithm, which, in my interpretation, shows the velocity fluctuation scale (i.e. large-scale fluid motion). Since this paper only uses low-Re data,

Regarding the novelty of the machine learning (ML) technique, we would like to note that this paper does not focus on the architecture used for predicting flow evolution or the accuracy of the predictions. Instead, we employ an explainable framework to uncover the complex and non-linear causal relationships within a turbulent flow. The key novelty of this work lies in defining the regions of the flow (SHAP structures) having the greatest influence on the causal correlations within the velocity field. To justify this idea, we have added a new section in the Supplementary Material to explain the causal implications of the explainable-deep-learning methodology:

Validation of the causal nature of the SHAP values

SHAP values are applied to the evolution of turbulent flows, identifying key regions that influence their development. The causal implications of the SHAP values are exploited, unveiling the cause and effect relationships inside the flow. In this section, we present a series of causal validation tests, adapted from \citet{martinez2024}, to assess the effectiveness of the explainable deep learning methodology. Each system consists of three variables, Q_1 , Q_2 and Q_3 , whose values depend on their

states at the previous time steps and a noise level W_1 , W_2 and W_3 respectively:

$$\left. \begin{aligned} Q_1^{t+1} &= f_{Q_1}(Q_1^t, Q_2^t, Q_3^t) + g_{W_1}(W_1^t) \\ Q_2^{t+1} &= f_{Q_2}(Q_1^t, Q_2^t, Q_3^t) + g_{W_2}(W_2^t) \\ Q_3^{t+1} &= f_{Q_3}(Q_1^t, Q_2^t, Q_3^t) + g_{W_3}(W_3^t) \end{aligned} \right\} \quad (1)$$

Next, a deep-learning model, f , is trained to predict the evolution of the variables Q_1 , Q_2 and Q_3 : $\left[Q_1^{t+1}, Q_2^{t+1}, Q_3^{t+1} \right] = f \left(\left[Q_1^t, Q_2^t, Q_3^t \right] \right)$. The model f is a fully connected neural network with 9 hidden layers of 8 neurons each and every an output layer with 3 neurons. The model is trained on a database of 200,000 samples, with 80% used for training and 20% for testing. Finally, the SHAP values are applied to assess the influence of the variables Q_1 , Q_2 and Q_3 at time t on their evolution at time $t+1$. These values are computed over the test database and averaged to identify which input variable has the most significant impact on the predictions of the models.

The first case is a system with a mediator variable, where an intermediate variable transmits the information between the other two variables. In this system, the variable Q_1 depends on Q_2 and Q_2 depends on Q_3 . The model is defined as follows:

$$\left. \begin{aligned} Q_1^{t+1} &= \sin(Q_2^t) + 0.001W_1^t \\ Q_2^{t+1} &= \cos(Q_3^t) + 0.01W_2^t \\ Q_3^{t+1} &= 0.5Q_3^t + 0.1W_3^t \end{aligned} \right\} \quad (2)$$

The mean SHAP values averaged over the test data, are presented in Supplementary Figure \ref{fig:barshap3}. In the figure, each color represents the contribution of the input variables Q_1^t , Q_2^t and Q_3^t to the evolution of the output variables: Q_1^{t+1} (blue), Q_2^{t+1} (orange) and Q_3^{t+1} (green). The SHAP values confirm that Q_2^t is the only variable influencing Q_1^{t+1} , while Q_3^t plays the most significant role in predicting both Q_2^{t+1} and Q_3^{t+1} , as defined by equation (\ref{eq:mediator}).

Supplementary Figure 1: SHAP evaluation of the mediator system. Mean SHAP value of the input variables Q_1^t , Q_2^t and Q_3^t for the model defined in equation (2), to predict their evolution Q_1^{t+1} in blue, Q_2^{t+1} in orange and Q_3^{t+1} in green.

The second case is a system with a cofounder variable. In this system, a single variable generates the other two, meaning that both variables Q_1 and Q_2 depend on Q_3 . The model is defined as follows:

$$\left. \begin{aligned} Q_1^{t+1} &= \sin(Q_1^t + Q_3^t) + 0.001W_1^t \\ Q_2^{t+1} &= \cos(Q_2^t - Q_3^t) + 0.001W_2^t \\ Q_3^{t+1} &= 0.5Q_3^t + 0.1W_3^t \end{aligned} \right\} \quad (3)$$

The results for the cofounder system, shown in Figure \ref{fig:barshap4}, demonstrate that the SHAP values can effectively capture the shared influence of Q_1 and Q_3 on the prediction of Q_1 . They also reflect the influence of Q_2 and Q_3 on the prediction of Q_2 . The predictions for Q_1 primarily depend on its own previous value, with a smaller contribution from Q_3 . In contrast, the cofounder variable Q_3 has a stronger influence in the prediction of Q_2 as indicated by the orange bars. This idea can be justified by analyzing the evolution of the temporal signals, Figure \ref{fig:evo4}. In this figure, the strong self-dependency of the variable Q_1 is evidenced as the high frequency of the signal Q_3 produces relatively small perturbations on the previous state of Q_1 . However, the signal Q_2 presents a higher frequency which is mostly condition by the previous state of Q_3 . This idea also evidences the capacity of the SHAP values not only to determine the causality between variables but also the intensity of the cause-effect relationships. Finally, Q_3 only depends on its previous value.

Supplementary Figure 2: SHAP evaluation of the cofounder system. Mean SHAP value of the input variables Q_1^t , Q_2^t and Q_3^t for the model defined in equation (3), to predict their evolution Q_1^{t+1} in blue, Q_2^{t+1} in orange and Q_3^{t+1} in green.

Supplementary Figure 3: Temporal evolution of the cofounder variables. The temporal evolution is presented for 30 consecutive time steps for the variables Q_1 in blue, Q_2 in orange and Q_3 in green. The time sampling is visualized by the position of the markers.

In the collider system, one variable depends on the other two. Specifically, the value of the variable Q_1^{t+1} is determined by the states of Q_2 and Q_3 . The model is defined as follows:

$$\left. \begin{aligned} Q_1^{t+1} &= \sin(Q_2^t Q_3^t) + 0.001W_1^t \\ Q_2^{t+1} &= 0.5Q_2^t + 0.1W_2^t \\ Q_3^{t+1} &= 0.5Q_3^t + 0.1W_3^t \end{aligned} \right\} \quad (4)$$

The mean SHAP values presented in Figure \ref{fig:barshap5} indicate that Q_1 is equally influenced by Q_2 and Q_3 , while Q_2 and Q_3 depend solely on their own previous value.

Supplementary Figure 4: SHAP evaluation of the collider system. Mean SHAP value of the input variables Q_1^t , Q_2^t and Q_3^t for the model defined in equation (4), to predict their evolution Q_1^{t+1} in blue, Q_2^{t+1} in orange and Q_3^{t+1} in green.

Finally, the redundant collider system is presented. In this case, two variables are identical, with Q_2 and Q_3 representing the same variable. The variable Q_1 depends on both Q_2 and Q_3 , despite them being identical. As shown in Figure \ref{fig:barshap6}, the SHAP values cannot differentiate them.

$$\left. \begin{aligned} Q_1^{t+1} &= 0.3Q_1^t + \sin(Q_2^t Q_3^t) + 0.001W_1^t \\ Q_2^{t+1} &= 0.5Q_2^t + 0.1W_2^t \\ Q_3^{t+1} &= Q_2^{t+1} \end{aligned} \right\} \quad (5)$$

Additionally, the SHAP values reveal that the influence of the sinus on the prediction of Q_1 is stronger than the effect of its previous state. The influence of the sinus in the evolution of the variable Q_1 is visualized in Figure \ref{fig:evo6}, where its variation follows Q_2 and Q_3 .

Supplementary Figure 5: SHAP evaluation of the redundant collider system. Mean SHAP value of the input variables Q_1^t , Q_2^t and Q_3^t for the model defined in equation (5), to predict their evolution Q_1^{t+1} in blue, Q_2^{t+1} in orange and Q_3^{t+1} in green.

Supplementary Figure 6: Temporal evolution of the redundant collider variables. The temporal evolution is presented for 30 consecutive time steps for the variables Q_1 in blue, Q_2 in orange and Q_3 in green. The time sampling is visualized by the position of the markers.

SHAP values effectively capture the causal contribution of input variables in a dynamic system. Deep-learning models used to predict the evolution of a dynamic system establish causal relationships between inputs and outputs. Applying SHAP

values to these models reveals these relationships by identifying the variables that have the greatest influence on the evolution of the system's state. In the turbulent channel case, grid points with higher SHAP values are those that more strongly influence the prediction of the flow's next state—in other words, they play a greater causal role in its evolution.

Furthermore, the following modifications have been implemented to further justify the relevance of the proposed method:

- The SHAP values quantify the causal importance of each individual grid point in the evolution of the flow (refer to the supplementary material for a detailed discussion and a comparison with the causal system examples presented by \cite{martinez2024}). As previously mentioned, due to the evolution of the model, the SHAP values determine which grid points are the most influential for the evolution of the flow. Accordingly, the SHAP structures identify the regions of the flow that should be targeted to effectively control its evolution~\cite{beneitez2025improving}.
- Thus, the SHAP structures can identify the Q events with higher impact for the evolution of the flow, or in other words, the most causal Q events.
- Since the SHAP structures objectively identify the most important regions of the flow, analyzing the differences between SHAP and other structures can provide an excellent venue to deepen our insight into wall-bounded turbulence, uncovering the most important causal relations in the system.
- The definition of causally important regions is a change of paradigm in the analysis of physical problems, and can be replicated in many other fields such as thermal fields or cavitation. Furthermore, it can help to develop much more efficient control strategies~\cite{beneitez2025improving}.

Additionally, the friction Reynolds number has been increased up to a value of 550, ensuring the separation of scales.

Regarding Figure 3, we present the probability of encountering a streamwise velocity fluctuation across different types of coherent structures, conditioned on the wall-normal distance. This plot encompasses all scales of velocity fluctuations detected within the structures, thereby capturing the full range of turbulent flow scales. Although we understand the doubts that might have arisen in the case of $Re_\tau = 125$ where the SHAP structures mainly appeared for high values of the streamwise velocity, the agreement with the results of $Re_\tau = 550$ demonstrate the capacity of the methodology to detect high-importance regions. This idea can be inferred from the modification of Figure 3. As shown below, although the Q events exhibit high probability of containing high streamwise velocity regions, the SHAP structures do not highlight these regions as important. Therefore, SHAP values not only detect velocity fluctuation scales, but they highlight the regions with a higher causal contribution for the future evolution of the flow. Our answer to the previous comment provides more details on the results at $Re_\tau = 550$.

Figure: Joint probability density function of the SHAP structures (left) and Q events (right) for a $Re_\tau = 550$.

I cannot address whether the new algorithm is superior to the existing structure detection method.

The methodology proposed in this paper offers several advantages over existing structure-detection methods. Explainable-machine-learning models inherently capture causal correlations. Therefore, when analyzing SHAP values, we are not merely examining an instantaneous flow magnitude (such as Reynolds stress in Q events) but rather identifying the regions that have the greatest influence on the future evolution of the velocity field.

This approach is crucial for turbulent-flow control, as effective control strategies require anticipating the flow's evolution to maximize the impact of control actions. In fact, we would like to add here the reference of a preprint paper in which we used a SHAP-based control strategy which outperformed the reduction of skin friction with other traditional methods:

Beneitez, M., Cremades, A., Guastoni, L., & Vinuesa, R. (2025). Improving turbulence control through explainable deep learning. *arXiv preprint arXiv:2504.02354*.

In order to highlight this idea in the main text, we have added the following sentence:

The definition of causally important regions is a change of paradigm in the analysis of physical problems, and can be replicated in many other fields such as thermal fields or cavitation. Furthermore, it can help to develop much more efficient control strategies~\cite{beneitez2025improving}.

Consequently, the SHAP framework not only identifies completely new structures purely based on importance, but can also guide control and optimization strategies by using the SHAP field as a reward to minimize/maximize.

At least Figure 3(a) represents just the sum of the u-streak structure near the wall ($y^+ \approx 12$), and above the Reynolds stress ($u'v'$) structures, then it is hardly said that the ML technique uncovers new turbulence structures even at this low rate of Reynolds number.

In fact, while the SHAP structures in Figure 3 may appear statistically similar to the sum of Q events and streaks, a closer examination of their instantaneous overlap in Figure 4 reveals significant differences. This analysis demonstrates that SHAP structures are not merely a combination of Q events and streaks, although some resemblance exists. In addition, the new case of application, at a higher Reynolds number, evidences larger differences between the SHAP structures, the Q events and the streaks. In the figures below, the joint probability density function of the streamwise velocity and the wall-normal distance for the SHAP structures (left), the Q events (center) and the streaks (right) for a turbulent channel at a friction Reynolds number $Re_\tau = 550$ is presented. In the figures the Q events show a high probability of high streamwise velocity regions at a wall-normal distance above 100 wall units. Nevertheless, this probability does not match the SHAP structures at the same wall-normal distance. Additionally, streaks and SHAP structures only match around $y^+ \approx 15$. Both ideas evidence that SHAP values do not represent Q events, streaks or both at the same time, but they represent those regions of the flow with higher causal implications, which might or might not match any other classical structure, as it has been presented in the supplementary material and it would be discussed later in this document. The modifications concerning the higher Reynolds number and the causal implication of the SHAP values were presented in the previous answers.

Figure: Joint probability density function of the streamwise velocity and the wall-normal distance for SHAP structures (left), Q events (center) and streaks (right).

In addition, an additional case of application is added to the paper, showing the identification of high-importance structures in the flow around a square wall-mounted obstacle. The following sentences have been added to the main text:

- An example of application of the present SHAP framework to another flow case, namely the flow around a wall-mounted square obstacle~\cite{martinez2023,yousif2023deep}, can be found in the Supplementary Material. In this case, the SHAP values highlight the influence of the wall and the obstacle, reducing the importance of the streaks $\backslash\text{rev}Rv\{$, identifying a similar importance of the Q events and a low importance of the vortices.

And the following section to the Supplementary Material:

SHAP structures in the flow around a square wall-mounted obstacle

In the present study, the SHAP structures have been calculated for a turbulent channel flow at two different friction Reynolds numbers: $\text{Re}_{\tau}=125$ and $\text{Re}_{\tau}=550$. However, in order to illustrate the adaptability of the present methodology to any type of flow, the SHAP structures in the flow around a square wall-mounted obstacle are also analyzed. The analyzed database, described in detail in Refs.~\cite{martinez2023,yousif2023deep}, was obtained by means of a direct numerical simulation (DNS) at a Reynolds number $\text{Re}_h = u_0 h / \nu = 2000$, where h is the height of the obstacle, u_0 the freestream velocity and ν the kinematic viscosity. We analyze a region of the domain on the leeward side of after the obstacle with size $2.86 h \times 2 h \times 1.24 h$. The obstacle has a cross-section $0.25h \times 0.25h$. The methodology of the main paper is reproduced for the obstacle flow, detecting the various coherent structures and evaluating their coincidence. It can be observed that, up to the obstacle height (for $y < h$), the agreement between SHAP structures and Q events is slightly lower than that at $\text{Re}_{\tau}=125$ (with a moderate increase in the shear layers at $y > h$). Regarding the streaks, the coincidence is much lower than in the case of the channel, since the main dynamics are within the wake, with the wall having a less important role. Finally, the vortices exhibit a low level of agreement with the SHAP structures as in the channel case. This example highlights that the SHAP framework can identify the most important features in different flows, regardless of the roles played by the classical structures.

Supplementary Figure 20: Coincidence of the various coherent structures in the flow around a square wall-mounted obstacle [10, 11]. Percentage of coincidence of pairs of the following structures: SHAP, Q events, streaks and vortices, relative to the volume of each type of the pair.

Below are specific minor comments: 1. How did the authors choose the time interval of $\Delta t^+ = 5$ for the high-importance region of SHAP, and what kind of physics is focused on within the short time interval? Considering the chaotic nature of turbulence, the highly important turbulence region should not be limited to such a short (viscous) time-interval projection, which may relate to the dissipation scale.

Following the reviewer's recommendation, we have extended the analysis to a time interval of $\Delta t^+ = 10$. As can be shown in the modifications below, the results are essentially the same as the ones with $\Delta t^+ = 5$, a fact that indicates that these importance-based structures exhibit robust properties and have high importance for the near-wall dynamics. The following text was added to the main paper:

- The Supplementary Material also presents results for a time interval $\Delta t^+ = 10$, showing high levels of agreement with the analysis for $\Delta t^+ = 5$.

And the following section was added to the Supplementary Material:

SHAP structures for longer time horizons

In this section, SHAP structures are analyzed at $Re_{\tau} = 125$ for a longer time interval between input and output: $\Delta t^+ = 10$. The corresponding joint probability density functions are shown in Supplementary Figure \ref{fig:sup_fig_dt10_1}. These distributions for $\Delta t^+ = 10$ exhibit a strong similarity with those obtained for $\Delta t^+ = 5$, as illustrated in Supplementary Figure \ref{fig:sup_fig_1}. As with the shorter time step, the SHAP analysis at $\Delta t^+ = 10$ identifies regions near the wall with high velocity, and regions farther from the wall with low velocity, as most influential. Notably, for $\Delta t^+ = 10$, high-importance ejections are absent from the channel center, indicating that increasing the time interval shifts the importance toward regions with maximal velocity fluctuations, specifically around $y^+ \approx 15$.

Supplementary Figure 18: Joint probability density function of the three velocity components of the SHAP structures for the different wall-normal distances for a time horizon $\Delta t^+ = 10$. The figure shows the distribution of the three velocity fluctuations, namely the streamwise, wall-normal, and spanwise components from left to right for $Re_\tau = 125$.

Regarding the overlap between SHAP structures and traditional coherent structures, the results are very similar to the ones obtained for $\Delta t^+ = 5$, with a small reduction in agreement, particularly with the streaks, which account for a smaller proportion of the SHAP values. This observation reinforces the interpretation that SHAP values isolate the most influential regions of the flow rather than simply reproducing known structures.

Supplementary Figure 19: Coincidence of the coherent structures for $\Delta t^+ = 10$. Percentage of coincidence of pairs of the following structures: SHAP, Q events, streaks and vortices, relative to the volume of each type of the pair for a turbulent channel at $Re_\tau = 125$.

Furthermore, one could consider longer time horizons and even other output functions (such as the skin friction), and these would probably yield different SHAP structures. This framework allows us to identify importance-based structures given a particular objective function. In this work this objective is the near-wall dynamics, but many other options are possible, and they will surely motivate future studies. The following text was added to the main text to reflect this point:

- Other possible models can be developed, e.g. for predictions with much longer time horizons (characteristic of the outer region) or of the wall-shear stress (where different SHAP structures may be identified). The different questions that can be formulated with various models will be addressed in future studies.

2. Another critical factor might be the choice of the U-net input in the SHAP algorithm. What will happen when we use the vorticity field as the input?

This is an interesting suggestion by the Reviewer. In fact, the proposed explainability framework enables answering questions regarding the most important underlying mechanisms for any input and output, as long as one can train a model yielding sufficiently good predictions. In this case, we created a model where the input and the output are the vorticity fluctuations, with the same time interval as that of the velocity fluctuations. The analysis reveals completely different physical phenomena, as discussed below. This can open the door to a wide range of possible analysis to identify novel mechanisms in high-order chaotic systems.

The following sentence was added to the discussion of the main paper:

In fact, the SHAP structures for the evolution of the vorticity are presented in the Supplementary Material.

Furthermore, the following modifications were added to the Supplementary Material:

The present analysis focuses on the evolution of vorticity in a turbulent channel flow at $\text{Re}_\tau = 125$. Here, vorticity fluctuations are temporally evolved using a U-net model, following the approach shown in Figure 1 of the main paper. SHAP values are then employed to quantify the importance of each grid point in the predictions of the model. Supplementary Figure \ref{fig:sup_fig_vor} presents the normalized joint probability density function (PDF) of the velocity fluctuation components and wall-normal distance within the SHAP-identified structures. These joint PDFs reveal that vorticity transport is primarily concentrated in ejection-like structures near the wall and sweep-like structures farther from the wall. The former are associated with vorticity generation due to wall friction and lift-off effects, while the latter contribute through the downward rotational motion of the flow toward the wall.

Supplementary Figure 20: Joint probability density function of the three velocity components of the SHAP structures for the evolution of the vorticity as a function of the wall-normal distance. The figure shows the distribution of the three velocity fluctuations, namely the streamwise, wall-normal, and spanwise components from left to right for $Re_\tau = 125$.

To investigate the instantaneous overlap between SHAP structures associated with vorticity and other coherent structures, a spatio-temporal coincidence analysis is performed. Supplementary Figure \ref{fig:sup_fig_coincvor} shows the volumetric overlap between the SHAP-identified vorticity structures and the other flow structures, including Q events, streaks, vortices, and SHAP structures for velocity. The SHAP structures for vorticity display a low level of agreement, with less than 20% volumetric coincidence with the other coherent structures. This result highlights the power of the explainable-deep-learning framework to identify different types of relevant phenomena in high-dimensional chaotic systems.

Supplementary Figure 21: Coincidence of the coherent structures. Percentage of coincidence of pairs of the following structures: SHAP for the vorticity, Q events, streaks, vortices and SHAP for the velocity, relative to the volume of each type of the pair for a turbulent channel at $Re_\tau = 125$.

3. What is the physical meaning of the SHAP vector, and how can we relate the squared-averaged vectors (eq.(1)) to the Reynolds-averaged Navier-Stokes equations?

The physical meaning of the SHAP vector depends on the specific question posed to the model. In this study, the model output is the mean-squared error of the velocity reconstruction at a future time step. Therefore, the SHAP values represent the error introduced when the information at a specific grid point is removed. In other words, they identify the most influential points in the causal relationships detected by the machine-learning model for the evolution of the velocity fluctuations. In order to clarify their meaning the following sentences has been added to the text:

- **These SHAP values identify the most influential points in the causal relationships detected by the deep-learning model. For this reason, the physical meaning of the SHAP values is conditioned by the output of the model. The SHAP values answer the question formulated mathematically for the deep-learning model. In the case presented in this work, the model calculates the mean-squared error of the reconstruction of the flow. Therefore, the SHAP values identify the most important regions of the flow for the evolution of the velocity fluctuation field.**
- **As previously mentioned, due to the evolution of the model, the SHAP values determine which grid points are the most influential for the evolution of the flow.**
- **The SHAP values are used to identify these high-importance regions by analyzing the most sensitive input features for a mathematically defined question: which regions minimize the error of the model predictions?**

4. I guess, authors means eq.(3): $|u| > \text{aur}$.

Since we are working with a three-dimensional problem, we aim to account for all the turbulent kinetic energy parallel to the wall. For this reason, we define the streaks by also incorporating the spanwise component.

Reviewer #1 (Remarks to the Author):

The authors answered the few open questions and also added another application of the method to study the flow around a square wall-mounted obstacle. Everything is well detailed and convincing

We would like to thank the kind words of the reviewer.

Reviewer #1 (Remarks on code availability):

The code is working and well commented, presented with a docker that eases the installation.

We also appreciate the words of the reviewer regarding the code.

Reviewer #2 (Remarks to the Author):

I'm satisfied with the modifications in the manuscript and I recommend it for publication.

We would love to thank the decision of the reviewer.

Review of the manuscript, Classically studied coherent structures only paint a partial picture of wall-bounded turbulence, NCOMMS-24-80608

This paper introduces data-driven turbulence structure detection using the SHAP algorithm. They detected the highest importance in channel turbulence. However, since the channel flow dataset has a very low Reynolds number, $Re_\tau = 125$, the length scale separation between the large-scale motion and the smallest (vertical) one is insufficient. I cannot find the novelty of using the machine learning (ML) technique, which costs a lot for the learning process and is difficult to apply for higher-Re simulation datasets. Also, the main result of Figure 3 represents the SHAP algorithm, which, in my interpretation, shows the velocity fluctuation scale (i.e. large-scale fluid motion). Since this paper only uses low-Re data, I cannot address whether the new algorithm is superior to the existing structure detection method. At least Figure 3(a) represents just the sum of the u-streak structure near the wall ($y^+ \approx 12$), and above the Reynolds stress ($u'v'$) structures, then it is hardly said that the ML technique uncovers new turbulence structures even at this low rate of Reynolds number. Below are specific minor comments:

1. How did the authors choose the time interval of delta $t^+ = 5$ for the high-importance region of SHAP, and what kind of physics is focused on within the short time interval? Considering the chaotic nature of turbulence, the highly important turbulence region should not be limited to such a short (viscous) time-interval projection, which may relate to the dissipation scale.
2. Another critical factor might be the choice of the U-net input in the SHAP algorithm. What will happen when we use the vorticity field as the input?
3. What is the physical meaning of the SHAP vector, and how can we relate the squared-averaged vectors (eq.(1)) to the Reynolds-averaged Navier-Stokes equations?
4. I guess, authors means eq.(3): $|u| > \alpha u_\tau$.